# Single-cell profiling screen identifies microtubule-dependent reduction of variability in signaling

C Gustavo Pesce[1], Stefan Zdraljevic[2] (iD), William J Peria[3] (iD), Alan Bush[4], María Victoria Repetto[4], Daniel Rockwell[1], Richard C Yu[1], Alejandro Colman-Lerner[4,†] (iD) & Roger Brent[3,†,*] (iD)

## Abstract

Populations of isogenic cells often respond coherently to signals, despite differences in protein abundance and cell state. Previously, we uncovered processes in the *Saccharomyces cerevisiae* pheromone response system (PRS) that reduced cell-to-cell variability in signal strength and cellular response. Here, we screened 1,141 non-essential genes to identify 50 "variability genes". Most had distinct, separable effects on strength and variability of the PRS, defining these quantities as genetically distinct "axes" of system behavior. Three genes affected cytoplasmic microtubule function: *BIM1*, *GIM2*, and *GIM4*. We used genetic and chemical perturbations to show that, without microtubules, PRS output is reduced but variability is unaffected, while, when microtubules are present but their function is perturbed, output is sometimes lowered, but its variability is always high. The increased variability caused by microtubule perturbations required the PRS MAP kinase Fus3 and a process at or upstream of Ste5, the membrane-localized scaffold to which Fus3 must bind to be activated. Visualization of Ste5 localization dynamics demonstrated that perturbing microtubules destabilized Ste5 at the membrane signaling site. The fact that such microtubule perturbations cause aberrant fate and polarity decisions in mammals suggests that microtubule-dependent signal stabilization might also operate throughout metazoans.

**Keywords** cell-to-cell variability; genetic screen; MAP kinase; microtubules; noise
**Subject Categories** Cell Adhesion, Polarity & Cytoskeleton; Quantitative Biology & Dynamical Systems; Signal Transduction
**Mol Syst Biol. (2018) 14: e7390**

## Introduction

Cell signaling systems transmit information about the external environment, enabling cells to respond to extracellular signals. Accurate signal transmission and response of individual cells, and coherence in cell population response, are critical for the choreographed sequence of signal and response during embryonic development, and for regulated cell division and differentiation during tissue maintenance in the adult. Variability in cell responses is well recognized and widespread, from *Escherichia coli* infected with phages (Delbrück, 1945), to mammalian cells subjected to pro-apoptotic signals (Spencer *et al*, 2009 and Appendix). However, the means by which cells transmit and respond to signals accurately, and so manifest coherent population responses, remain largely unknown.

We and others have studied cell-to-cell variability, using the cell fate decision system that controls mating in *Saccharomyces cerevisiae*, the pheromone response system (PRS) (Colman-Lerner *et al*, 2005; Yu *et al*, 2006; Paliwal *et al*, 2007; Ricicova *et al*, 2013). The PRS has elements prototypic for many other signaling systems: It uses a GPCR, which, when bound by pheromone, couples via a G-protein to a scaffold-dependent MAPK cascade (Dohlman & Thorner, 2001; Fig 1). In this cascade, there are two partially redundant MAP kinases, Fus3 and Kss1, each able to activate downstream steps. After activation, receptors, G-proteins, and the scaffold concentrate into a membrane patch (Suchkov *et al*, 2010; Ventura *et al*, 2014; Ismael *et al*, 2016) here called the signaling site. The cell converts extracellular ligand concentration into an occupancy measurement (Brent, 2009) by determining the ratio of ligand-occupied to unoccupied receptors (Bush *et al*, 2016) and transmitting that information accurately, via negative feedback (Yu *et al*, 2008) and "push–pull" mechanisms (Andrews *et al*, 2016). Signaling causes outputs including induction of genes at appropriate levels (here called "system output") that depend on a set of proteins that constitute the signaling arm of the PRS. Determination of the direction of a gradient of pheromone concentration, and subsequent growth toward a mating partner, depends on a partly overlapping set of proteins, the polarity determination arm of the system.

1   Abalone Bio, Inc., Richmond, CA, USA
2   Department of Molecular Biosciences, Northwestern University, Evanston, IL, USA
3   Fred Hutchinson Cancer Research Center, Seattle, WA, USA
4   IFIBYNE-UBA-CONICET and Departamento de Fisiología, Biología Molecular y Celular, Facultad de Ciencias Exactas y Naturales, Universidad de Buenos Aires, Buenos Aires, Argentina
    *Corresponding author. Tel: +1 206 667 1482; E-mail: rbrent@fhcrc.org
    †These authors contributed equally to this work

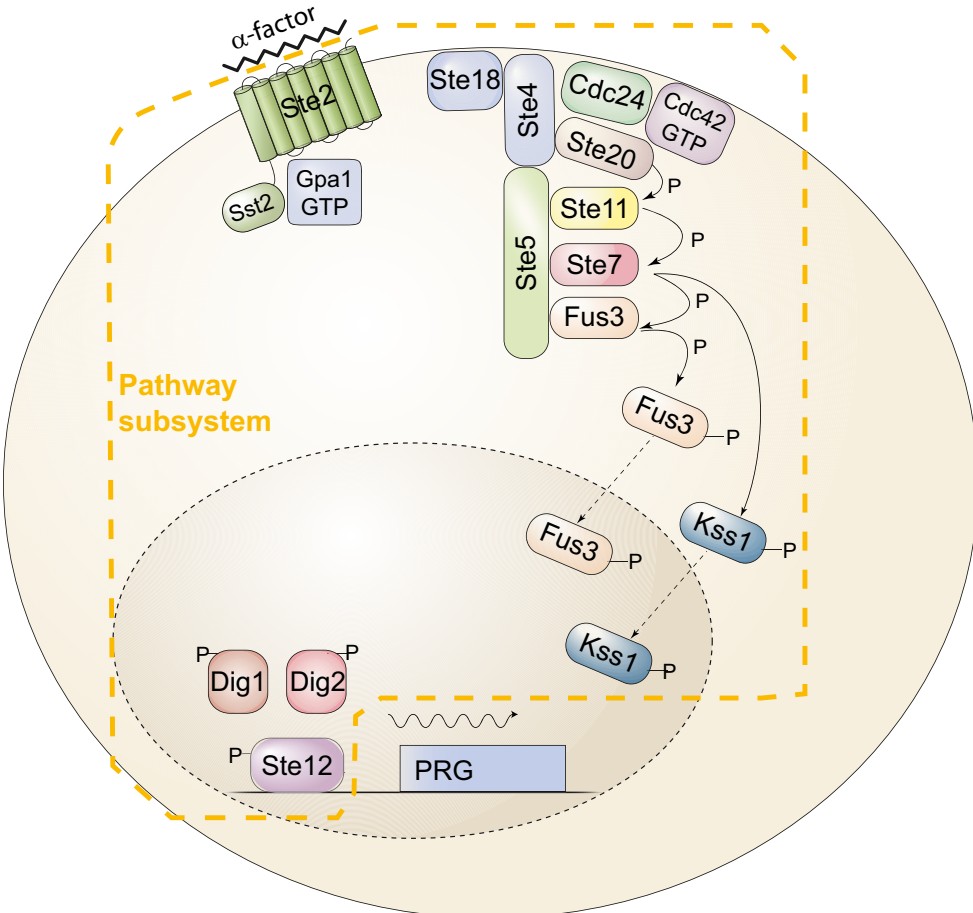

**Figure 1.  The signaling arm of the yeast pheromone response system (PRS).**

Binding of the ligand, α-factor, to a seven-helix transmembrane receptor, Ste2, in the MATa cell depicted, causes the dissociation of the α subunit of a trimeric G-protein, Gpa1, from the βγ dimer, Ste4/Ste18. This event causes the recruitment to the plasma membrane of the scaffold protein Ste5, leading to the assembly and activation of the MAP kinase cascade (MAPKKK Ste11, MAPKK Ste7) and the detachment from the scaffold of the Erk1/2-like MAPKs Fus3 and Kss1 (Dohlman & Thorner, 2001). In the cytoplasm, activated Fus3 and Kss1 phosphorylate targets including Ste5 (Bhattacharyya *et al*, 2006; Malleshaiah *et al*, 2010), and in the nucleus, they phosphorylate Dig1, Dig2, and Ste12 (Tedford *et al*, 1997). These events comprise the pathway subsystem, P; that is, the subsystem that transmits the signal to the promoters of inducible genes. Activation of Ste12 leads to the induction of approximately 100 pheromone-responsive genes (PRGs) (Roberts *et al*, 2000) and their expression via the expression subsystem G (defined in the text).

Our previous work quantified system output by expression from PRS-responsive and control reporter genes. It separated the cell-to-cell variability in output into two contributions. The first of these was from cell-to-cell variability in the pathway subsystem, **P** (includes all events upstream of the promoter of the reporter gene), quantified as $\eta^2(P)$ ($\eta^2$ = variance/mean$^2$), and here called "pathway variability". The second contribution was from variability in events related to reporter gene expression, either due to (i) preexisting differences in the general capacity of cells to express genes into proteins, **G**, quantified as $\eta^2(G)$, and here called "variability in gene expression" or (ii) rapid-acting changes in gene expression due to "intrinsic noise", which we quantified as $\eta^2(\gamma)$. In this previous work, we made the assumption that cell-to-cell differences in (P) were composed of $\eta^2(L)$, (differences in L, the capacity component of the signal transmission subsystem at the start of the experiment) and $\eta^2(\lambda)$, (rapid-acting changes in signal during the measurement) but we could not separate $\eta^2(L)$ and $\eta^2(\lambda)$ experimentally.

Four lines of evidence show that cell-to-cell variation and pathway variability, $\eta^2(P)$, is under active control. First, pathway subsystem output P correlates negatively with gene expression capacity G, indicating a compensatory mechanism that reduces variability in system output (Colman-Lerner *et al*, 2005). Second, mutations in either of the PRS MAPKs Kss1 and Fus3 affect $\eta^2(P)$, and do so differently (Colman-Lerner *et al*, 2005). Third, maintenance of the matching dose-response relationship between system output and system activity, which reduces the amplification of stochastic noise $\eta^2(\lambda)$ during signal transmission at an intermediate point requires the action of negative feedback from Fus3 (Yu *et al*, 2008). Fourth, we showed recently (Bush *et al*, 2016) that a push–pull mechanism suppresses cell-to-cell differences in signal-dependent gene expression caused by changes in the abundance of the receptor. Here, we hypothesized that there might be additional mechanisms that regulate (or suppress) variability in transmitted signal.

# Results

## Large-scale screen identifies genes whose products affect pathway variability

To identify genes that affected cell-to-cell variability, we first constructed a whole genome collection of yeast carrying the necessary reporters and mutations in each non-essential gene. To do this, we extended established methods facilitating genetic crosses of arrayed collections (Tong *et al* 2004, see Appendix).

During our initial characterization of cell-to-cell variability phenotypes in our collection we found that, for many gene deletions, the patches of post-sporulation segregants contained varying numbers of colonies of genetically variant haploids, likely arising from chromosomal mis-segregation during meiosis (Hughes *et al* 2000), same sex diploid formation (Giaever & Nislow, 2014) and possibly also from mutations present in some cells in the starting collection. While such heterogeneity, if present, might have not had a large impact on phenotypes studied before in similar collections (Jonikas *et al*, 2009; Neklesa & Davis, 2009; Wolinski *et al*, 2009; Ayer *et al*, 2012), our measures of cell-to-cell variability were very sensitive to it. We thus generated our collection from clonal cultures derived from single colonies.

To screen the mutant collection, we optimized a flow cytometry adaptation of our microscopic methods to measure single-cell responses. In this screen, we arrested cell cycle progression by inhibition of the Cdc28-as2 mutant protein with the inhibitor 1NM-PP1-NM, incubated cells with pheromone in well plates, stopped the response by the addition of cycloheximide, and allowed time for the fluorophores to fully mature. We first tested this method on the reference strain. We extended our previous characterization of the dose response of pathway variability $\eta^2(P)$ to a broader range of different pheromone concentrations (0.1–30 nM). Consistent with previous microscopic measurements at just two doses (Colman-Lerner *et al*, 2005), the fine-grained dose response showed that $\eta^2(P)$ decreased monotonically with increasing pheromone (Fig EV1). We thus were reassured about using this approach to screen the arrayed mutant collection.

For the primary screen, we assayed 1,141 strains from the collection (996 randomly selected and 145 bearing a deletion in a non-essential kinase or phosphatase (Appendix Table S2). We screened these for expression related variables (Table 1) that allowed us to compute pathway output (P) and/or variability in it ($\eta^2(P)$), system output (O) and variability in it, and a proxy for gene expression capacity, G. Screened strains corresponded to more than 1/4 of the non-essential yeast genes. Appendix Table S2 shows the numerical results.

From these screened strains, we selected gene deletions for follow-up "secondary screen" studies, based on their $\eta^2(P)$ and average output (O) phenotypes (Fig 2A–C). We chose selection thresholds that lay in the tails of the distributions of values measured for the 52 separate cultures of the reference strain included in the screen. From the low dose (0.6 nM pheromone) data, we selected mutants with high or low median pheromone system output (O) (Fig 2A) or high or low $\eta^2(P)$ (Fig 2B). From the high dose (20 nM pheromone) data, we only selected mutants that showed high $\eta^2(P)$ (Fig 2C). Figure 2D and E shows $\eta^2(P)$ vs. P, at 0.6 nM (D) and 20 nM (E) pheromone doses, for all measured strains.

For the secondary screen, we isolated three fresh independent haploid segregants and assayed them by flow cytometry as above (see Appendix for a complete description of primary and secondary screens). These screens identified 50 deletion strains (Table 2) that reproducibly showed changes in O or $\eta^2(P)$.

We did an additional follow-up "tertiary screen" on duplicate independent isolates of 44 of the haploid deletion strains (Appendix Table S6). For this screen, we used microscope-based quantification of the fluorescent protein reporters. Although we did not seek to gain biological insight from observation of effects of these gene deletions on cell morphology, this microscope-based quantification of fluorescence signal had two advantages. First, it allowed us to rule out the possibility that putative single-cell values were actually derived from clumps of several cells. None of the mutant cultures we imaged was affected by these problems. Second, it allowed us to measure another variable, gene expression noise, $\eta^2(\gamma)$, by simultaneous quantitation of the two fluorescent protein reporters (CFP and mRFP) driven by $P_{PRM1}$ (accurate CFP measurements were not possible in the flow cytometer). The tested mutants showed values of $\eta^2(\gamma)$ that were typical of the reference strain. The only significant differences were in O, $\eta^2(O)$, and $\eta^2(P)$.

## Mutant genes define different axes of quantitative system behavior

To gain insight into the different phenotypes caused by these gene deletions, we grouped the mutant strains in the secondary screen

**Table 1.  Variables measured in isogenic cell populations.**

| Variable name | Short expression | Calculated as |
|---|---|---|
| Median pheromone response system (PRS) output | **O**, $<P_{PRM1}$-mRFP$>$ or $<P_{PRM1}$-mCherry$>$ | Median (inducible RFP) |
| Median constitutive or control output | **G**, $<P_{ACT1}$-XFP$>$ or $<P_{BMH2}$-XFP$>$ | Median (constitutive XFP) |
| Cell-to-cell variability in PRS output | $\boldsymbol{\eta^2(O)}$, $\eta^2(P_{PRM1}$-mRFP$)$ or $\eta^2$(mCherry) | $\sigma^2_{XFP}/\mu^2_{XFP}$ |
| Cell-to-cell variability in constitutive or control output (representing general gene expression capacity) | $\boldsymbol{\eta^2(G)}$, $\eta^2(P_{ACT1}$-YFP$)$ or $\eta^2(P_{BMH2}$-YFP$)$ | $\sigma^2_{YFP}/\mu^2_{YFP}$ |
| Cell-to-cell variability in signal transmission | $\boldsymbol{\eta^2(P)}$, or $\eta^2(L + \lambda)$ | $\sigma^2(mRFP_i/<mRFP> - YFP_i/<YFP>)$ |
| Signal strength | **P** | **O/G** |

$\sigma^2$ is variance, $\mu$ is average, $<B>$ means the average of B. Quantities in bold type are those used for selection and/or clustering analysis below.

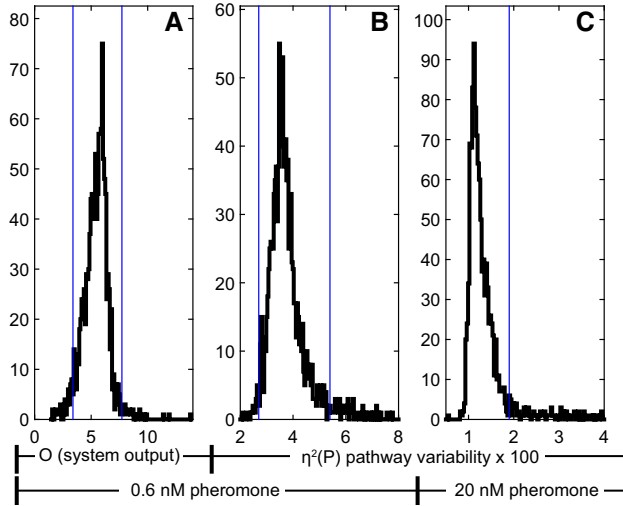

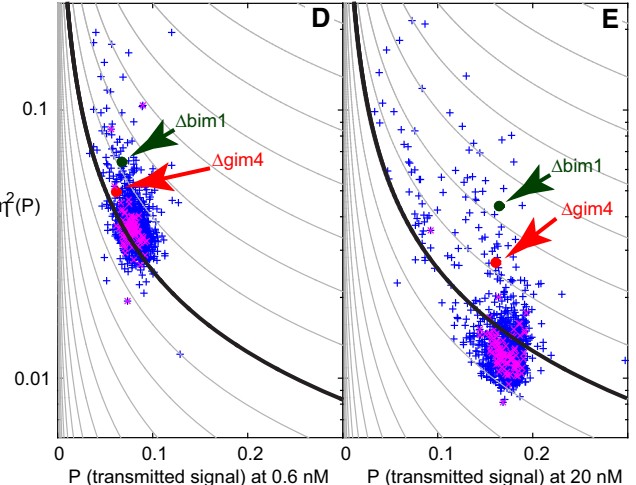

**Figure 2. Selection of mutants for follow-up studies.**

Plots show distributions of values for 991 randomly selected non-essential deletion strains, and 102 additional strains with deletions of a non-essential kinase or phosphatase, and two wild-type strains. Values were derived from flow cytometry data obtained after 3 h of stimulation with pheromone. Blue vertical bars indicate the thresholds used to select mutants for secondary screens (see Appendix).

A    PRS output, O (median mRFP signal), in 0.6 nM pheromone.

B    Estimated pathway variability $\eta^2(P)$ in 0.6 nM pheromone

C    Estimated $\eta^2(P)$ in 20 nM pheromone (see Appendix Table S1).

D, E    Signaling variability vs. transmitted signal P (median mRFP/median YFP) for all 1,093 strains screened. Plots show an estimate of $\eta^2(P)$ vs. P for the same dataset displayed in Fig 4A–C. The contour lines show the expected dependence of variability on output for outputs proportional to a Poisson random variable (lower noise at higher outputs), with proportionality constants logarithmically spaced from $10^{-5}$ to 1. Purple Xs are independent replicates of the reference SGA 85 strain. Their spread gives an indication of the limits of this primary screen. The SGA 85 swarm lies below the 0.158 contour at 20 nM but above it at 0.6 nM, indicating that variability at the low dose is higher than expected from the same Poisson processes taking place at 20 nM. At 0.6 nM, $\Delta bim1$ and $\Delta gim4$ showed somewhat greater, and at 20 nM substantially greater, pathway variability than reference cells. See Appendix Table S2 for a list of all strains and their corresponding raw output and variability values.

Source data are available online for this figure.

using a hierarchical clustering approach based on the five variables we measured by flow cytometry, at low and high pheromone dose (Fig 3 and Appendix Table S2). Fourteen of the 19 cultures of the reference strain grouped together in one cluster (cluster I), one in cluster IIa, two in cluster IIIa, one in cluster IIIb, and one in cluster Vc. With a few exceptions (for example $\Delta ckb1$, $\Delta his1$, and $\Delta sky1$) either all or all but one of the independent segregants bearing each gene deletion grouped in the same subcluster. Taken together with the results of the tertiary screen, these results show that differences in variability in strains with different gene deletions were due to the mutations. Since all 19 cultures of the SGA85 reference cells were isogenic, that five of these cultures grouped into different clusters highlight the fact that these high-throughput flow cytometric assays sometimes perform inconsistently. Similarly, since our independent haploid segregants came from crosses with an otherwise isogenic MATα strain, we believe that the observed infrequent grouping of any single deletion's isolates into multiple clusters most likely reflects measurement anomalies rather than uncharacterized genetic differences between the MATa and MATα parents of the strains.

We noted that the pathway and gene expression output variables (O and G) were often affected by different genes than the "cell-to-cell variability" variables ($\eta^2(O)$, $\eta^2(G)$, $\eta^2(P)$). For example, cluster II was comprised of all the entries with low pathway output (O) and low gene expression output (G). Within this cluster, there were three subclusters: strains with high or unchanged $\eta^2(O)$ and $\eta^2(P)$ (IIa), strains with low or unchanged $\eta^2(O)$ (IIb), and strains with high or unchanged $\eta^2(O)$ and $\eta^2(P)$ at the 0.6 nM dose (IIc). Another example was cluster III, which contained strains with predominantly high pathway output O and gene expression output G, but low or unchanged $\eta^2(O)$ and $\eta^2(G)$. Within cluster III, subclusters IIIa and IIIb were defined, respectively, by low $\eta^2(P)$ and high $\eta^2(P)$, both at the low pheromone dose. Such genetic independence strongly suggests the existence of distinct, independent mechanisms affecting the two types of quantitative phenotypes (mean output and variability) and disfavor an interpretation in which variability is inextricably linked to output strength. From this screen, mean and variability emerged as independent axes of system behavior, subject to independent regulation, in the sense that they were often independently affected by genetic changes (see Discussion).

This analysis also revealed that processes that affected variability appeared to be different at low and high doses of pheromone. This was evidenced in the subclusters within cluster V. Cluster V grouped strains with high cell-to-cell variability in both system output and gene expression output. Subcluster Va contained strains with high $\eta^2(P)$ at high pheromone and low $\eta^2(P)$ at low pheromone. In contrast, strains in subcluster Vb showed high, unchanged, or low $\eta^2(P)$ at high or low pheromone. Strains in subcluster Vc had moderately high or unchanged $\eta^2(P)$ at low pheromone and low $\eta^2(P)$ at high pheromone.

Another result of the clustering analysis was that mutations in related genes showed similar patterns of change in their set of quantitative measurements. This was expected and yet reassuring. For example, deletions of duplicated paralogs of ribosomal protein genes were grouped in subclusters IIa and IIb (distinguished by their different variability phenotypes) and those for the two PRS MAPKs, FUS3 and KSS1, grouped together in cluster IV. By analogy with dataset clustering studies based on gene expression data and other

**Table 2. Genes found in the screen.**

| Gene name | Screen, criteria | Important for | Description |
|---|---|---|---|
| ARG82 | U,2 | AA metabolism | Inositol polyphosphate multikinase |
| ERV46 | U,1,4,5 | Cargo transport | ER vesicle protein, component of COPII complex; required for membrane fusion |
| HIS1 | U,1,4,5 | AA metabolism | ATP phosphoribosyltransferase |
| UGA1 | U,5 | AA metabolism | Gamma-aminobutyrate (GABA) transaminase |
| SLA1 | U,2,3 | Actin binding | Cytoskeletal protein binding protein; required for assembly of the cortical actin cytoskeleton |
| SAP155 | K,2,3,5 | Cell cycle | Protein required for function of the Sit4 protein phosphatase |
| YER068C-A | U,5 | Dubious open reading frame | Dubious open reading frame/overlaps with ARG5, ARG6 acetylglutamate kinase and N-acetyl-gamma-glutamyl-phosphate reductase |
| YIL032C | U,5 | Dubious open reading frame | Dubious open reading frame/next to BCY1 |
| ERG3 | U,1,5 | Ergosterol biosynthesis | C-5 sterol desaturase |
| GAL83 | K,1 | Glucose repression | One of three possible beta-subunits of the Snf1 kinase complex |
| GUP1 | U,2,3 | Glycerol metabolism, protein folding | Plasma membrane protein involved in remodeling GPI anchors |
| PPZ1 | K,5 | Ion homeostasis | Serine/threonine protein phosphatase Z, isoform of Ppz2p; involved in regulation of potassium transport, which affects osmotic stability, cell cycle progression, and halotolerance |
| FUS1 | U,5 | Mating | Membrane protein localized to the shmoo tip |
| AAT2 | U,5 | Metabolism | Cytosolic aspartate aminotransferase involved in nitrogen metabolism |
| PTC6 | K,2,3 | Metabolism | Mitochondrial type 2C protein phosphatase (PP2C) |
| GIM4 | U,5 | Microtubule chaperone/Protein folding | Subunit of the heterohexameric cochaperone prefoldin complex |
| PAC10 GIM2 | U,5 | Microtubule chaperone/Protein folding | Subunit of the heterohexameric cochaperone prefoldin complex |
| BIM1 | U,4,5 | Microtubule end binding | Microtubule plus-end-binding protein |
| MSH1 | U,5 | Mitochondrial homeostasis | *Escherichia coli* MutS homolog, binds DNA mismatches, required for mitochondrial function |
| BUB1 | K,4,5 | Mitosis | Protein kinase required for cell cycle checkpoint, delays entry into anaphase until kinetochores bound by opposing microtubules |
| ELM1 | K,2 | Morphogenesis | Serine/threonine protein kinase that regulates cellular morphogenesis |
| HSL1 | K,2 | Morphogenesis | Nim1-related protein kinase; regulates the morphogenesis and septin checkpoints |
| NUP60 | U,5 | Nuclear transport | FG-nucleoporin component of central core of the nuclear pore complex |
| SXM1 | U,4,5 | Nuclear transport | Nuclear transport factor (karyopherin) |
| CBR1 | U,5 | Respiration | Microsomal cytochrome β reductase |
| RTC3 | U,4,5 | RNA metabolism | Protein of unknown function involved in RNA metabolism |
| CKA1 | U,1,4,5 | Signaling | Alpha catalytic subunit of casein kinase 2 (CK2) |
| CKB1 | U,5 | Signaling | Beta regulatory subunit of casein kinase 2 (CK2) |
| CKB2 | K,1 | Signaling | Beta' regulatory subunit of casein kinase 2 (CK2) |
| FUS3 | K,3,5 | Signaling | Mitogen-activated serine/threonine protein kinase (MAPK), part of PRS |
| HOG1 | K,3 | Signaling | Mitogen-activated protein kinase involved in High Osmolarity (HOG) pathway |
| KSS1 | K,3 | Signaling | Mitogen-activated protein kinase (MAPK); functions in PRS and signal transduction pathways that control filamentous growth and pheromone response |

**Table 2**  (continued)

| Gene name | Screen, criteria | Important for | Description |
|---|---|---|---|
| PBS2 | K,5 | Signaling | MAP kinase kinase of the HOG signaling pathway |
| SSK2 | K,3 | Signaling | MAP kinase kinase kinase of HOG1 signaling pathway |
| FAR1 | U,1 | Signaling/cell cycle/polarization | CDK inhibitor, nuclear anchor, recruited by Ste18-Ste4 at polarity patch |
| SIP1 | U,4,5 | Signaling/glucose repression | Alternate beta-subunit of the Snf1 protein kinase complex |
| KAR4 | U,5 | Signaling/mating | Transcription factor required for activation of some pheromone-responsive genes |
| STE50 | U,1,4 | Signaling/mating | Adaptor protein, in PRS helps connect Ste20 MAPKKKK to Ste11 MAPKKK |
| SKY1 | K,3 | Splicing | SR protein kinase (SRPK); varied functions, regulates proteins involved in mRNA metabolism and cation homeostasis, helps some LexA fusion proteins bind operator |
| KIN3 | K,5 | Stress | Non-essential serine/threonine protein kinase; possible role in DNA damage response |
| OCA1 | K,1 | Stress | Protein tyrosine phosphatase; required for cell cycle arrest in response to oxidative damage of DNA |
| CTK1 | K,4 | Transcription regulation | Catalytic (alpha) subunit of C-terminal domain kinase I (CTDK-I); phosphorylates RNA pol II |
| DEP1 | U,5 | Transcription regulation | Component of the Rpd3L histone deacetylase complex, variously needed for activation and repression, regulates DNA replication origin timing |
| SUM1 | U,2,4,5 | Transcription regulation | Transcriptional repressor that regulates middle-sporulation genes; required for mitotic repression of middle-sporulation-specific genes; also acts as general replication initiation factor; involved in telomere maintenance, chromatin silencing |
| SWI5 | U,5 | Transcription regulation | Part of Mediator and Swi/Snf nucleosome remodeling complexes |
| UME6 | U,4,5 | Transcription regulation | Meiotic transcription regulator, DNA binding, recruits variously Sin3/Rpd3 repressor (HDAC) and Ime1 activator. |
| RPL12A | U,1,4 | Translation | Ribosomal 60S subunit protein L12A |
| RPL19B | U,1,4,5 | Translation | Ribosomal 60S subunit protein L19B |
| RPL34A | U,4,5 | Translation | Ribosomal 60S subunit protein L34A |
| ECM15 | U,1,4,5 | Unknown | Possibly tetrameric, non-essential protein, unknown function |
| VPS64 | U,5 | Vacuole metabolism | Required for cytoplasmic proteins to enter vacuole |

Selection criteria codes (see Fig 2A–C): (1) O(0.6 nM) < 3.39; (2) O(0.6 nM) > 7.72; (3) $\eta^2$(P(0.6 nM)) < 0.027; (4) $\eta^2$(P(0.6 nM)) > 0.054; (5) $\eta^2$(P(20 nM)) > 0.019. Genes from the strains in the unbiased (U) screen and the strains in the non-essential kinase and phosphatase screen (K) screen that showed altered PRS system output (O), low or high pathway variability ($\eta^2$(P)) at low pheromone (0.6 nM), or high variability ($\eta^2$(P)) at high dose (20 nM). Table shows gene name, screen from which it was selected and selection criteria, overall functional class, and a brief description of its molecular role or activity.

phenotypes (including our own, Colman-Lerner *et al* 2001; Wolfe *et al* 2005), we expect that the gene deletions that shared cluster or subcluster membership might function in the same processes.

**Three gene deletions with higher signaling variability affect microtubule function**

Of the 50 "variability genes" so identified, six (*FUS1, FUS3, KSS1, FAR1, KAR4,* and *STE50*) were known components of the pheromone response system or induced by it, three of which (*FUS3, KSS1,* and *STE50*) we had previously shown to affect pathway variability (Colman-Lerner *et al*, 2005; Pincus *et al*, 2013). The other 44 included genes involved in cell cycle regulation (11), gene expression—transcription, RNA processing, and nuclear pore transport (12 genes, including *NUP60*)—metabolism—amino acid synthesis and

mitochondrial function (10)—morphogenesis—actin, tubulin, and their regulation (7)—other (2) and unassigned function (3).

Three genes that affected $\eta^2$(P) were known to affect microtubule function. These were *GIM4* and *PAC10/GIM2*, whose products form part of a six-protein prefoldin complex needed for tubulin supply, and *BIM1*, whose product mediates attachment of cytoplasmic microtubule plus ends to the signaling site. We selected two of these genes, *BIM1* (two out of three in cluster IIIa) and *GIM4* (*GIM4* two out of three in cluster IIa) as candidate genes to explore a possible relationship between microtubule function and signal variability. Although deletions of both *BIM1* and *GIM4* caused elevated $\eta^2$(P) in the primary screen at both low and high doses, Δ*bim1* did not show elevated $\eta^2$(P) at low doses in the secondary screen, but showed elevation at both doses in the tertiary screen. We again took these differences in

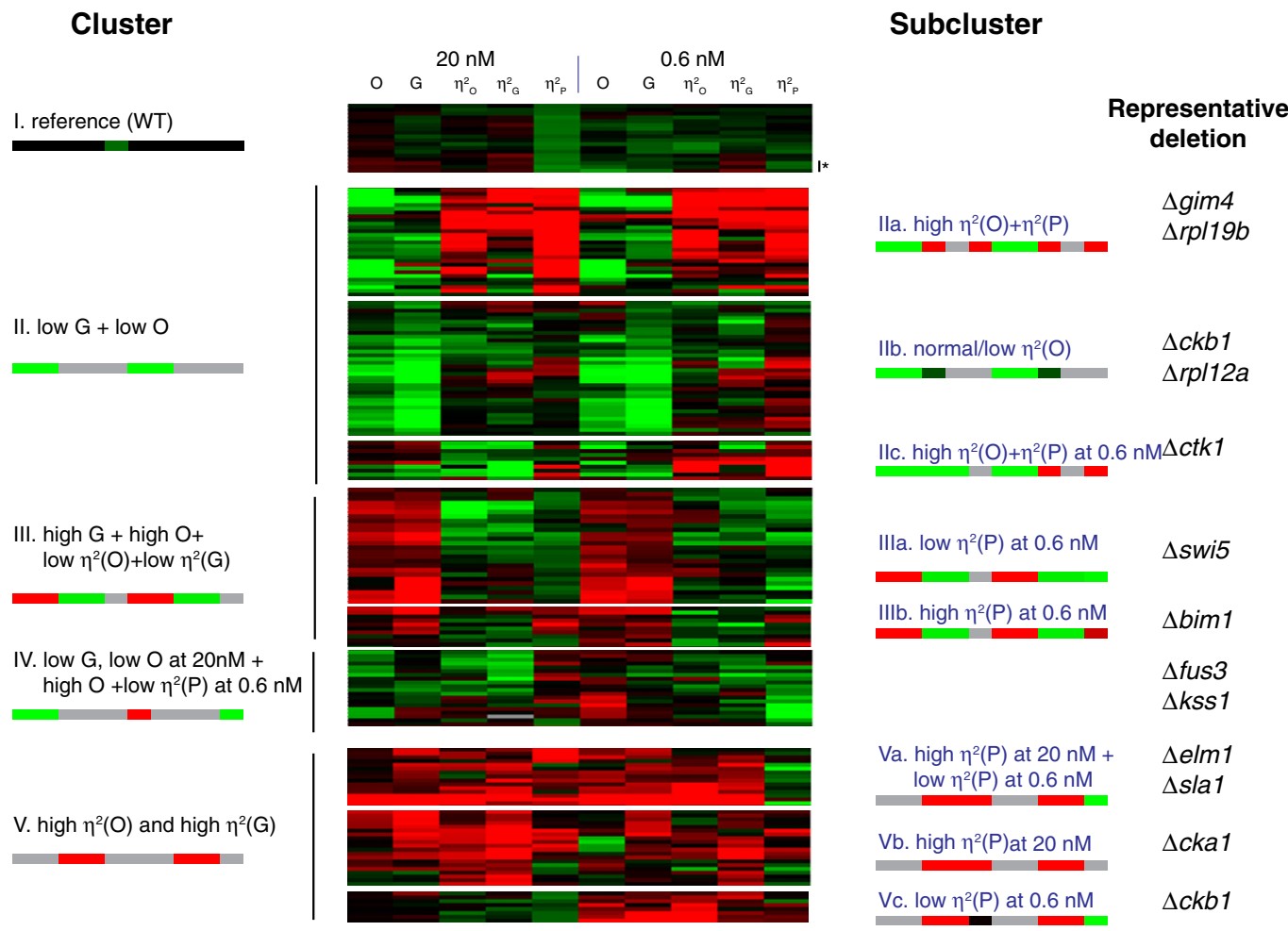

**Figure 3.   Cluster analysis of 50 genes identified as affecting variability and or pheromone response output.**

Hierarchical clustering of values derived from flow cytometry measurements from 198 cell populations (19 replicates for reference strain SGA85, four independent segregants each for 17 deletions from the kinases or phosphatase set and three independent segregants each for 37 deletions from the unbiased set). We used the Pearson correlation metric to assess distance between strains and the average linkage method to form clusters. Before clustering, we first log-transformed the data and then median centered each row (each strain). Each strain had the following 10 measurements (five after induction with 20 nM pheromone and five after induction with 0.6 nM pheromone): O (pheromone system output), G (gene expression output), and $\eta^2(O)$, $\eta^2(G)$ and $\eta^2(P)$, the three cell-to-cell variability measurements. The panel shows these values as a "heat map", from red (higher than the median) to black (equal to the median) to green (lower than the median). The signature pattern for each cluster or subcluster is represented with a color bar with 10 blocks, one for each measurement (gray indicates that that the measurement may take any value). Rightmost column shows representative deletion strains for each subcluster. The asterisk next to the last row of the reference cluster indicates the data are from $\Delta fus1$, which did not differ from reference in this re-assay. Appendix Table S3 shows the raw data and Appendix Table S4 lists the clustered, log-transformed and median-centered dataset. Appendix Table S6 shows microscope data that complement these flow cytometer data for 44 of these 50 mutant strains selected for clustering.

Source data are available online for this figure.

measured $\eta^2(P)$ values as likely indicating the limitations of such measurements via the relatively high-throughput culture in multi-well plate/flow cytometry assays rather than arising from otherwise cryptic genetic variability among isolates.

However, to address the above possibility, and to get around any possible effect of uncharacterized genetic heterogeneity among independent haploid deletion strains resulting from independent meioses, we remade these strains without meiosis, in a clean genetic background, an independently constructed BY4741 derivative, equivalent to the reference strain (Appendix). From this strain (GPY4000), we constructed $\Delta bim1$ and $\Delta gim4$ single deletion strains, and a $\Delta bim1\ \Delta gim4$ double mutant. We characterized the behavior of these newly generated mutant strains in repeated fine-grained dose-response flow cytometry assays. Figure 4 shows the results. Pathway variability as a function of transmitted signal, $\eta^2(P)$ vs. P, was increased similarly in both deletion strains, across all pheromone doses (Fig 4A). In contrast, transmitted signal P as a function of pheromone dose was relatively unaffected in $\Delta bim1$ but reduced by ~30% in $\Delta gim4$ (Fig 4B). Reductions in P at a given dose merely indicate that signal transmission is less efficient in the mutant strain; it might still occur by the same process. In $\Delta bim1$ $\Delta gim4$ cells, the increase in pathway variability $\eta^2(P)$ was more than twice as large as the measured effect of the two individual deletions. This synergistic genetic interaction suggested that the two

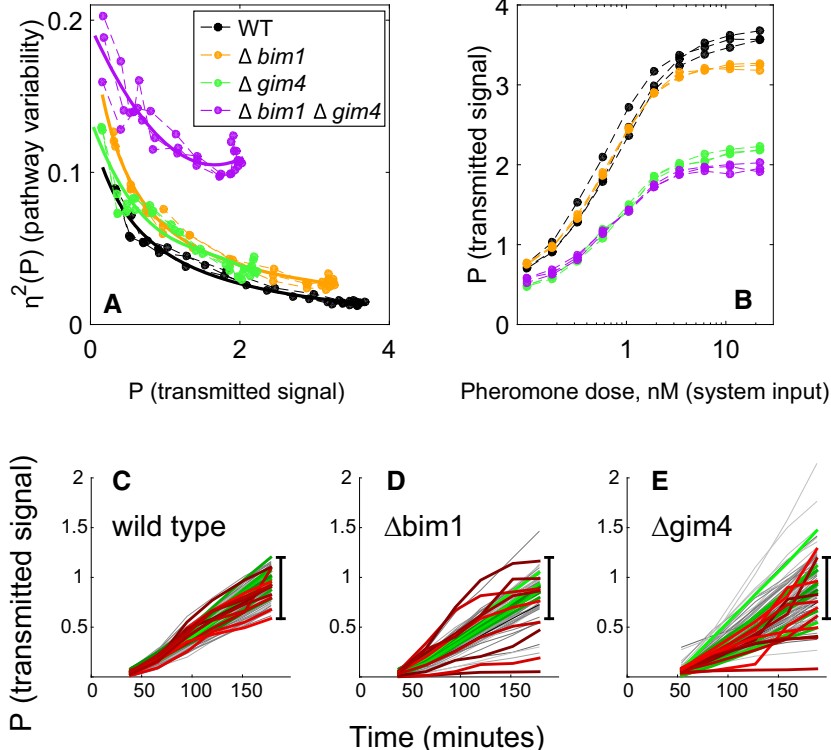

**Figure 4.  Increased cell-to-cell variability and distinct time-dependent trajectories in Δ*bim1* and Δ*gim4* mutants.**

A, B  Deletions of *BIM1* and *GIM4* in clean genetic background increase signaling variability at all outputs. Data were collected from dose-response flow cytometry measurements of reference (GPY4000), Δ*bim1* (GPY4001), Δ*gim4* (GPY4031), and *bim1 gim4* (GPY4036) (bearing the $P_{PRM1}$-*mCherry* and $P_{BMH2}$-*YFP* reporters) stimulated for 3 h with the indicated pheromone doses. (A) $\eta^2(P)$ as a function of P. Solid curves are best fits of a rational polynomial model to the measurements from each strain. These models tend toward infinity as transmitted signal tends toward zero, where we expect very large relative variability. (B) P as a function of pheromone dose. Data correspond to measurements at different doses measured on the same day (three replicates).

C–E  Accumulated signal P vs. time in reference and microtubule perturbed cells. Transmitted signal P vs. time in individual cells of the reference (GPY123, "wild type"), Δ*bim1* (GPY4144) and Δ*gim4* (GPY4150) strains (N = 103, 102, 118, respectively). We induced the PRS by addition of 20 nM pheromone to the medium and imaged cells every 30 min as previously done (Colman-Lerner *et al*, 2005 and Gordon *et al*, 2007). In all populations, about 5% of the cells did not respond to pheromone induction. Traces correspond to pathway output (inducible $P_{PRM1}$-*mCherry* signal/constitutive $P_{ACT1}$-*CFP* signal) from individual cells followed over time. For each strain, we colored the 10 most stable trajectories green, and the 10 least stable (crooked) trajectories red. The black bar shows the full range of WT responses, for comparison.

Source data are available online for this figure.

gene products acted through distinct mechanisms to affect $\eta^2(P)$ (see for example Fisher 1918; Boone *et al* 2007).

To better characterize the signaling phenotype of the Δ*bim1* and Δ*gim4* mutants, we used time-lapse microscopy to measure total system output (O) and transmitted signal (P) in single cells tracked over time. Figure 4C–E shows plots of P in single cells over time ("trajectories") after stimulation with 20 nM pheromone. The reference strain ("wild-type") cell trajectories were clustered relatively tightly (Fig 4C); in contrast, the trajectories in the Δ*bim1* (Fig 4D) and Δ*gim4* (Fig 4E) populations were loosely spread. To the naked eye, the Δ*bim1* and Δ*gim4* cells differed from each other in terms of how common trajectories with extremely high or low Ps were. In Δ*gim4*, the trajectories presented a broader distribution of trajectories around the mean. In contrast, in Δ*bim1* populations, most cells had trajectories matching the reference strain, but a number of "outlier" cells showed P very far from the mean.

To quantify these differences, we developed a distributional measure called the median progressive spread (MPS) based on the progressive spread distribution (PSD, see Appendix; For a symmetric distribution, the MPS is essentially the interquartile range). The MPS for the reference strain population was 0.22 (95% CI 0.19–0.23); for Δ*bim1* cells, it was 0.24 (95% CI 0.17–0.31, not distinguishable from reference), while for Δ*gim4* cells, it was 0.36 (95% CI 0.32–0.37). The MPS thus shows that the distribution of trajectories in Δ*gim4* cells is in fact significantly broader overall than the reference, while, for Δ*bim1* cells, the breadth of the core distribution is indistinguishable from the reference.

We then analyzed the stability of the trajectories over time. In all populations, a few cells showed unstable increases in P visually evidenced as erratic or "crooked" trajectories [red traces in Fig 4C–E]. To assess changes in the occurrence of crooked trajectories we defined a crookedness index ($I_C$, see Appendix). We set an $I_C$ threshold of 0.3, above which we consider trajectories to be crooked. By

this measure, 11% of the trajectories in the Δ*bim1* set and 16% in the Δ*gim4* set were crooked, in contrast to 6% in the reference strain population.

Both reference and Δ*bim1* cells showed crooked trajectories above and below the mean value of P. In contrast, in the Δ*gim4* cells, 13 out of 17 (76.5%) cells showing crooked trajectories had values of P below the mean. Since Δ*gim4* cells also showed a reduction in system output (O), this result suggested that cells that have their signal lowered by loss of prefoldin are prone to exhibiting unstable trajectories. Thus, for Δ*gim4* cells, a substantial fraction of the increased $\eta^2(P)$ might be a secondary consequence of a primary defect in signal transmission strength, while for Δ*bim1* cells, $I_C$ and $\eta^2(P)$ are independent of changes in signal transmission strength.

### Impairment of the microtubule bridge between the nucleus and the signaling site causes increased signaling variability, while its elimination diminishes pathway output

When the PRS is activated, receptors, G-proteins, the Ste5 scaffold, and other membrane proteins localize to a membrane signaling site. Polarized growth from this site causes the cells to form a mating projection, and the cells to adopt an overall morphology known as a "shmoo". Cytoplasmic microtubules form a bridge connecting the Spindle Pole Body (SPB) on the nuclear membrane to the site at the

shmoo tip (Maddox *et al*, 1999, 2000, 2003). To better understand the effect of the Δ*bim1* and Δ*gim4* mutations, we used genetic and chemical means to perturb specific aspects of microtubule function (Fig 5). We first tested the effect of mutations known to affect microtubule-generated pulling forces. At the plasma membrane, the microtubule bridge connecting the signaling site to the nucleus alternately grows and shrinks. This is due to the fact that microtubule plus ends at the plasma membrane alternate between binding membrane-attached Bim1 and the Kar3-Cik1 complex. Bim1 binds plus ends of polymerizing microtubules, thus lengthening the microtubule bridge and generating a "push" force on the nucleus. The Δ*bim1* mutation disrupts pushing, and, as shown in Fig 4, increases $\eta^2(P)$ without greatly diminishing average P. Kar3/Cik1 binds plus ends and actively depolymerizes them, thus generating a "pull" force on the nucleus. Figure 5 shows the effects of deletions in *KAR3* or *CIK1* (Fig 5A–D). For comparable values of pathway output (P), both perturbations markedly increased variability in transmitted signal, $\eta^2(P)$ (Fig 5A and C). Δ*kar3* did not affect the average transmitted signal (P) (Fig 5B). By contrast, Δ*cik1* caused a strong reduction in P (Fig 5D).

We next disrupted microtubule polymerization and de-polymerization by expression of a dominant negative variant of α-tubulin, encoded by the *TUB1-828* allele. Tub1-828 generates "frozen" plus ends, unable to depolymerize or polymerize (Anders & Botstein, 2001). To do so, we constructed a strain carrying an

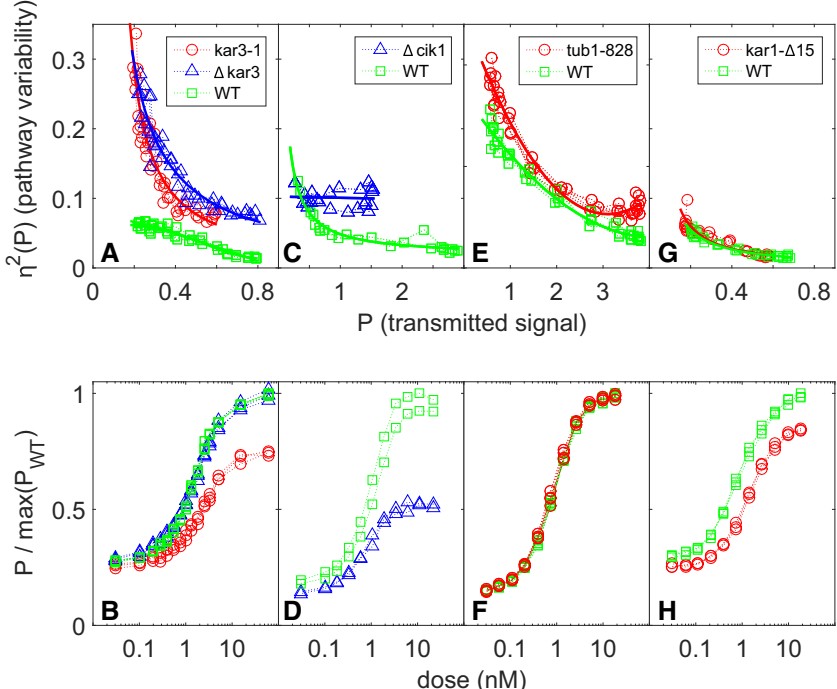

**Figure 5. Cell-to-cell variability in signal transmission in cells with mutations affecting microtubule end function.**

A–H    $\eta^2(P)$ vs. P and dose response in: (A, B) *kar3-1* (SGA108, the "rigor mutant"), Δ*kar3* (SGA2015), and WT cells (SGA103) (three replicates each); (C, D) Δ*cik1* (GPY4123) and WT cells (GPY4000) (two replicates); (E, F) tub1-828 expressing and WT cells (GPY1858) (four replicates); and (G, H) *kar3-Δ15* (SGA109) and WT cells (SGA103) (three replicates). In (A, C, E, G), the x-axis values are the transmitted signal P. In (B, D, F, H), the y-axis values are P divided by the maximum value of P observed for the WT strain in each.

Source data are available online for this figure.

estradiol-inducible *TUB1-828* construct driven by the *GAL1* promoter (see Materials and Methods and Louvion *et al*, 1993). As shown in Fig 5E and F, the very low baseline *Tub1-828* expression in the absence of estradiol increased $\eta^2(P)$ at all pheromone doses tested. Notably, this low level, *Tub1-828-expression* caused no changes in transmitted signal (P) (Fig 5F). These results showed that altering microtubule function can increase variability in transmitted signal without affecting mean signal strength.

In Δ*bim1* and Δ*kar3* cells, the microtubule bridge was frequently detached from the plasma membrane (Maddox *et al*, 1999, 2000). In light of this behavior, we considered the hypothesis that the existence of the bridge alone would be sufficient to maintain normal levels of $\eta^2(P)$, and that generation of pulling and pushing forces might be irrelevant. To test this idea, we constructed a *kar3-1* variant of the reference strain. Kar3-1 localizes to the signaling site and binds microtubule plus ends, but can neither actively depolymerize nor release them (because these microtubules can neither lengthen nor contract, Kar3-1 is referred to as a "rigor" mutant). The rigor mutation increased $\eta^2(P)$ by at least a factor of three, as much as the full Δ*kar3* lesion; it also caused a ~30% reduction in transmitted signal (Fig 5A and B). These results show that, to reduce $\eta^2(P)$, microtubules not only have to be attached to the plasma membrane, but must also be able to alternatively push and pull on the nucleus and exert force.

Next, we tested the effects of preventing the attachment of microtubules to the SPB. To do that, we replaced *KAR1* with the *kar1-Δ15* allele (Appendix Table S1). *KAR1* encodes a component of the SPB. The *kar1-Δ15* allele expresses a C-terminally truncated variant of Kar1. During the pheromone response, the Kar1 C terminus is the main site of minus-end microtubule anchoring to the MTOC (Pereira *et al*, 1999); its absence leads to detachment of cytoplasmic microtubules from the nucleus and from the signaling site (Erlemann *et al*, 2012). The *kar1-Δ15* mutation was without effect on $\eta^2(P)$ (Fig 5G), although it caused a reduction in pathway output (P) (Fig 5H).

Like *kar1-Δ15*, some of the microtubule perturbations above (Δ*gim4*, Δ*cik1, kar3-1*) also caused decreases in P. We thus hypothesized that a microtubule bridge is required for pathway output to reach its maximal levels. In this view, even a malfunctioning bridge allows P to reach high values (as seen with the Δ*kar3*, Δ*bim1,* and *Tub1-828* perturbations), albeit with a concomitant increase in $\eta^2(P)$. Conversely, the absence of the bridge constrains P without impacting $\eta^2(P)$ (as in *kar1-Δ15*).

To test the hypothesis above, we used a chemical treatment that causes the disappearance of visible microtubule structures in pheromone-treated cells. We used a combination of the microtubule-polymerization inhibitors benomyl and nocodazole, each at close-to-saturation concentrations (see Materials and Methods and Palframan *et al*, 2006). Chemical disruption of microtubules caused a decrease in P at doses of pheromone higher than the EC50 (Fig EV2A) and did not affect $\eta^2(P)$ (Fig EV2B). These results mirrored the *kar1-Δ15* results. Since both perturbations caused the disappearance of the microtubule bridge, these results support the view that the absence of the bridge diminishes the strength of the transmitted signal P vs. dose, but causes no changes in signal variability $\eta^2(P)$ vs. P, while presence of a bridge that cannot exert push and pull forces results in normal P but increased variability in it.

## Nucleus-to-signaling-site distance does not correlate with pathway output

Yeast cells actively position both the cell nucleus and the pheromone-inducible genes within it in relation to the signaling site. During the pheromone response, the nucleus remains seemingly "anchored" in a cytoplasmic volume at the base of the mating projection (often referred to as the "base of the shmoo tip"). Casolari *et al* (2005) showed that 49 pheromone-inducible nuclear genes became associated with the nuclear pore complex after pheromone stimulation and that one of them, *FIG2*, translocated to the region of the nuclear periphery closest to the shmoo tip. More recently, Randise-Hinchliff *et al* (2016) showed that *PRM1* and other pheromone-inducible genes translocate to the same spot, as long as they are bound by the activated pheromone-responsive transcription factor Ste12. The relationship between this localization and signal transmission is unclear. Maeder *et al* (2007) showed that, in cells exposed to saturating isotropic pheromone, the concentration of active Fus3 (phosphorylated or "P-Fus3", see Fig 1), was maximal at the signaling site and decreased toward the base of the shmoo tip. We had observed that at doses of pheromone near the EC50 the nuclei are localized closer to the signaling site than they are at saturating doses (Appendix Fig S6).

Given the above and that a variability gene found in our screen, *NUP60*, encodes a nuclear pore protein required for pheromone-responsive gene recruitment to the nuclear periphery, we wondered whether the position of the Ste12 bound genes within the nucleus and within the gradient might affect or even regulate signal transmission. Since nuclear positioning depends on the integrity and proper function of the microtubule bridge, we hypothesized that microtubule perturbations that increased signal variability might do so by altering the position of the nucleus within the P-Fus3 gradient in the cell, which in turn might affect the average strength and/or the variability of the transmitted signal received by the pheromone-responsive genes.

We therefore tested whether in mutant cells in which the nucleus was unattached to the signaling site, pathway output (P) would be weaker when the nucleus wandered further from the signaling site, and stronger when closer. To do so, we measured nuclear positioning and its relation to pathway output (P) in Δ*bim1*, Δ*gim4, kar3-1,* and reference cells. Our measurements showed clear and distinct effects of the three perturbations on nuclear positioning. However, we did not see a correlation between the position of the nucleus and the strength of transmitted signal (see Appendix Fig S6). These results argued against the idea that the signal received at pheromone-induced genes depended on the distance between the nucleus and the signaling site (see Discussion).

## Genetic bypass of Ste5 recruitment suppresses the effect of microtubule perturbations on pathway variability

We next sought to identify the steps in the signaling pathway at which microtubule perturbation increased signal variability. To do so, we performed "bypass" experiments made possible by work by Pryciak and collaborators that demonstrated graded ectopic activation of the PRS at different downstream steps by expression of different artificial activators (Takahashi & Pryciak, 2008). In this system, expression of PRS activator proteins was driven by $P_{GAL1}$,

whose activity was controlled in turn by the above-described estrogen-responsive Gal4 derivative (Gal4-ER-VP16, Louvion *et al*, 1993). Figure 6 shows results of experiments using two artificial activators: native Ste4, whose expression mimics dissociated (i.e., active) Gβγ dimers, and Ste5-CTM, a fusion of the Ste5 scaffold with a transmembrane domain, whose expression mimics membrane-recruited (i.e., activated) Ste5 (Pryciak & Huntress, 1998). To prevent interference from basal activation of the native PRS, this last strain was Δste5. In the Ste4 activator strains, Δ*bim1* and Δ*gim4* caused increases in $\eta^2(P)$ (Fig 6A), the same effects they had when we stimulated the PRS with pheromone (see Fig 4). In Ste5-CTM cells, the Δ*bim1* perturbation did not increase $\eta^2(P)$, while the increase caused by Δ*gim4* was smaller than in the Ste4

strain. These results suggested that the microtubule-dependent process(es) affected by the Δ*bim1* perturbation, and possibly by the Δ*gim4* perturbation, increased pathway variability downstream of Ste4, but at or upstream of recruitment of Ste5 to the membrane by Ste4. Given that Ste5 activation by membrane recruitment is an early signaling-site event that is not immediately relatable to known microtubule roles, this was a surprising finding.

By contrast, the effects of Δ*bim1* and Δ*gim4* on transmitted signal strength, P, were not suppressed by Ste5-CTM (Fig 6B). Δ*bim1* caused a decrease in P in the artificial activator system that was greater than in the pheromone-activated PRS (see Fig 4), while Δ*gim4* caused a decrease of similar magnitude. These reductions in P were also present in the Ste5-CTM activator strains. These results suggested that Δ*bim1* and Δ*gim4* affected signal strength (P) by mechanisms distinct from those that mediated their effects in signal variability ($\eta^2(P)$).

### Induced signaling variability by microtubule perturbations requires Fus3

These bypass experiments were consistent with the idea that the increased pathway variability caused by the Δ*bim1* and Δ*gim4* perturbations are due to effects on membrane recruitment by Ste4 of the Ste5 MAP kinase cascade scaffold. To be activated, the Fus3 MAPK must be bound to the Ste5 scaffold, while the other MAPK, Kss1, does not require such association (Fig 1 and legend). We therefore suspected that the effect of microtubule perturbations might propagate preferentially via Fus3. Thus, we measured $\eta^2(P)$ in Δ*fus3* or Δ*kss1* cells. Notably, in Δ*bim1* and Δ*gim4* strains, the Δ*fus3* deletion suppressed, at all doses, the increased $\eta^2(P)$ (Fig 7A, top panels), while the Δ*kss1* deletion exacerbated the defect of Δ*bim1* (Fig 7A, bottom panels). The fact that deletion of *FUS3* eliminated the increase in pathway variability caused by the Δ*gim4* and Δ*bim1* mutations showed that the increased variability was not a secondary consequence of a generalized increase in variability in cells with disrupted microtubule function, but rather reflected an effect of these mutations on the operation of the PRS. These results demonstrate that microtubule perturbations increase pathway

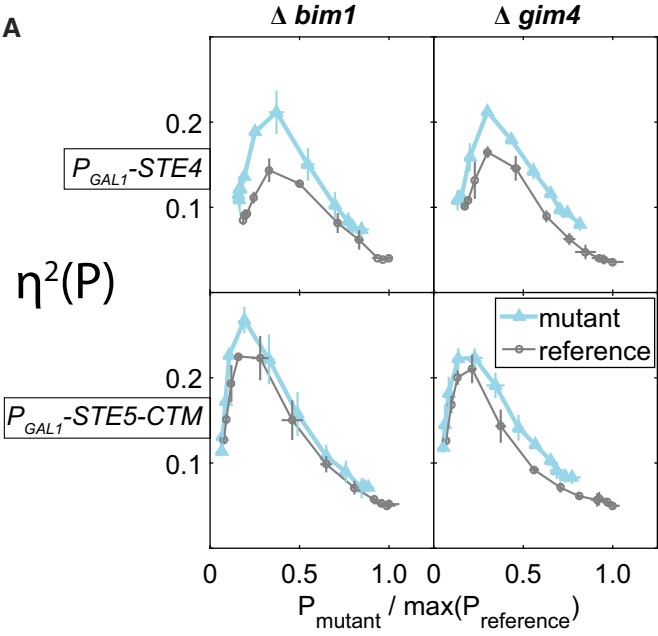

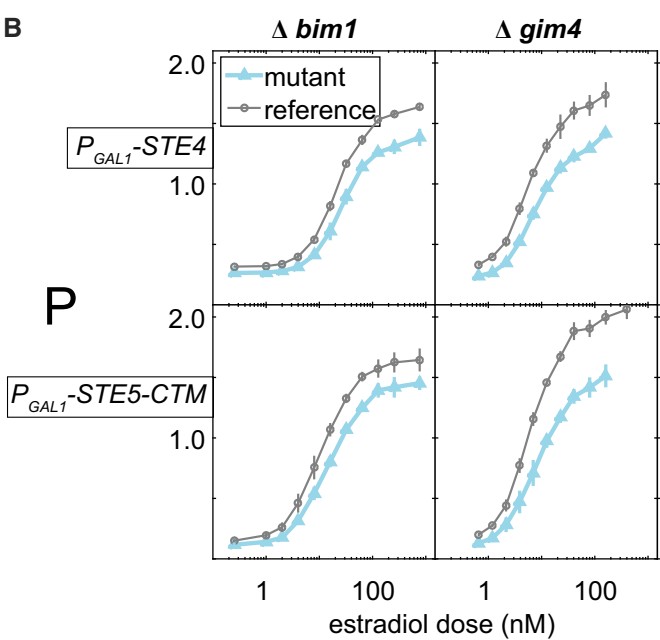

**Figure 6. Microtubule perturbations affect pathway variability $\eta^2(P)$ and transmitted signal P at or upstream of the Ste5 recruitment step.**

We exposed reference ("WT"), and Δ*bim1*, Δ*gim4* derivatives of GPY1810 (bearing the chimeric genes $P_{PRM1}$-*mCherry*, $P_{BMH2}$-*YFP*, and a gene constitutively expressing the chimeric transcription factor $P_{BMH2}$-*GAL4BD-hER-VP16*), to the indicated concentrations of estradiol for 180 min to induce expression of two ectopic activators of the pheromone response system, Ste4 and Ste5-CTM. Error bars show standard deviations computed over the three replicates.

A Y-axis shows pathway variability $\eta^2(P)$, x-axis shows signal strength P, normalized by the maximum P observed for each reference strain. This normalization allows comparison between strains with different activation points. There are three replicate cultures of each mutant and reference strain.

B X-axis values are estradiol dose, y-axis values are P (same measurements as in x-axis of panel A, but here un-normalized). Reductions in these values are thus also reflected in reduced ranges of P, relative to reference, for plots in (A). Figure 4B shows corresponding reduced response of Δ*bim1* and Δ*gim4* mutants to normal pheromone induction.

Source data are available online for this figure.

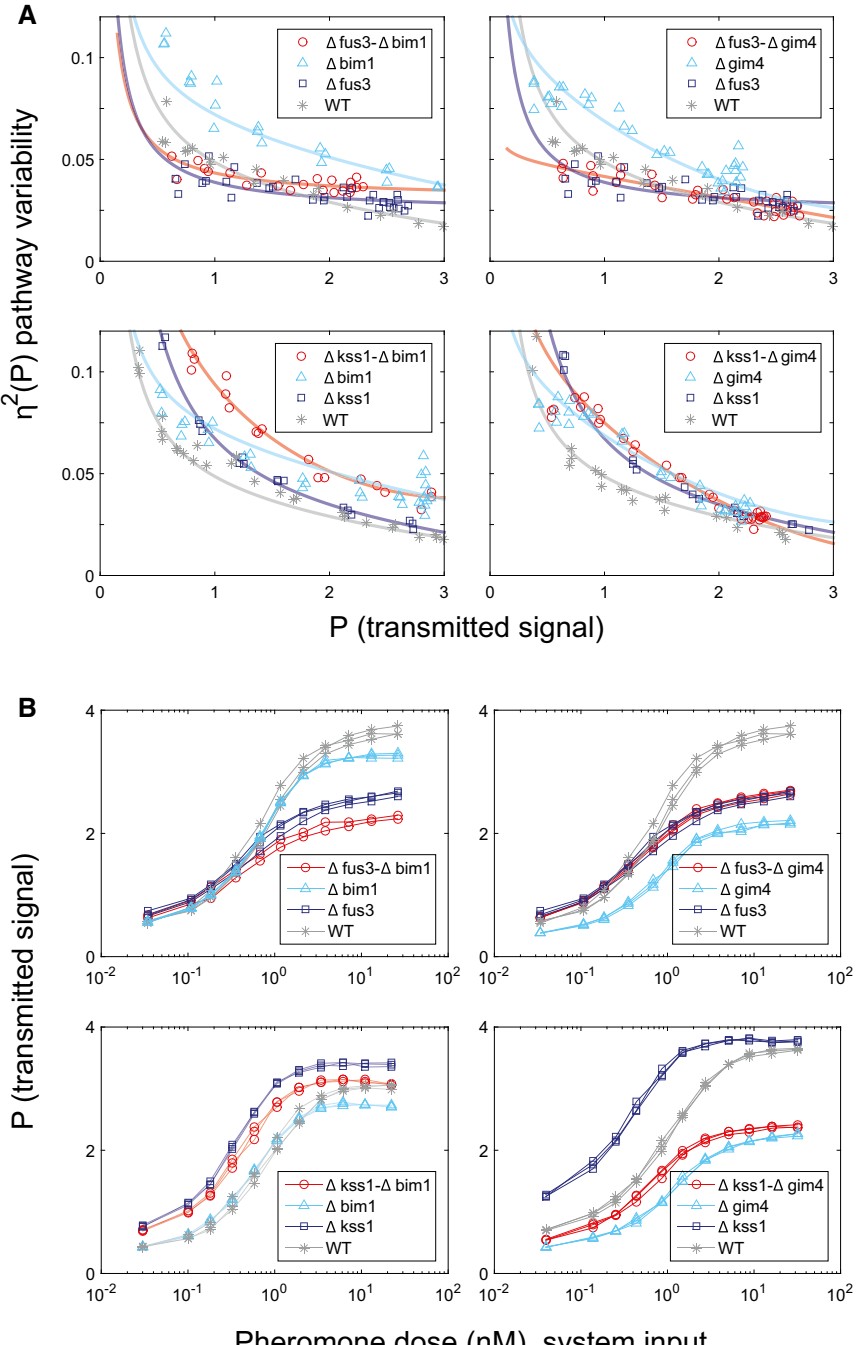

**Figure 7. Fus3 is required for deletions of GIM4 and BIM1 to increase cell-to-cell variability.**

A Deletions of FUS3 and KSS1 have distinct effects on the increased cell-to-cell variability caused by deletions of GIM4 and BIM1. We induced the PRS in the indicated strains by addition of 20 nM pheromone to the medium and measured reporter activity after 3 h. Y-axis in each panel shows $\eta^2(P)$ as a function of P. Deletion strains derive from the reference WT GPY4000. $\Delta bim1$ and $\Delta gim4$ strains show increased $\eta^2(P)$ relative to WT (P-values are both less than $10^{-5}$), while $\Delta fus3\ \Delta bim1$ and $\Delta fus3\ \Delta gim4$ strains show $\eta^2(P)$ not distinguishable from WT (P-values are 0.341 and 0.095, respectively). However, $\Delta bim1\ \Delta kss1$ and $\Delta gim4\ \Delta kss1$ cells show higher $\eta^2(P)$ relative to WT (P-values are both less than $10^{-5}$). The double mutant $\Delta bim1\ \Delta kss1$ had significantly higher $\eta^2(P)$ than either $\Delta kss1$ or $\Delta bim1$ alone (P-values are both $10^{-5}$ or less), while the double mutant $\Delta gim4\ \Delta kss1$ had essentially the same $\eta^2(P)$ as either $\Delta kss1$ or $\Delta gim4$ alone (P-values are 0.341 and 0.106, respectively). This shows that Fus3 protein is required for increased $\eta^2(P)$. (All strains have three replicates except for one of the double deletions, $\Delta fus3\ \Delta bim1$, which has two replicates.) To compare the piecewise-linear curves implied by the data from each pair of strains, we use an Area Between the Curves (ABC) metric, with each area estimated by trapezoidal integration. We then obtain the P-values by resampling over the replicates under the null hypothesis that the two strains are in fact the same; the P-value is the number of times that the resampled ABC was at least as far from the median as the realized ABC.

B P vs. dose for the datasets of panel (A).

Source data are available online for this figure.

   

variability $\eta^2(P)$ by specifically impacting signaling by the MAP kinase Fus3.

In its effects on P, diminution caused by the deletion $\Delta fus3$ was additive to the diminution caused by $\Delta bim1$. By contrast, the deletion $\Delta kss1$ counteracted the diminution caused by $\Delta bim1$, so that P for the double mutant $\Delta kss1$-$\Delta bim1$ was above the reference at all but the highest doses (Fig 7B, left panels). The reduction of signal strength due to $\Delta gim4$ (Fig 7B, top right) was smaller in cells that were also $\Delta fus3$, whose reduction in P was the same as in $\Delta fus3$ $GIM4^+$ cells. By contrast, in $\Delta gim4$ cells, the additional deletion $\Delta kss1$ does not enhance signal strength, but in fact reduces it (Fig 7B, bottom right). These results further support the idea that the mechanism(s) that affect pathway variability is (are) distinct from those affecting signal transmission strength.

### Perturbation of microtubule function increases variability in establishment and maintenance of the initial signaling site

The above experiments suggested that the $\Delta bim1$ and $\Delta gim4$ perturbations operated at the Ste5 membrane recruitment/retention step to cause variability in signal transmitted by Fus3. We thus sought to directly observe the dynamics of Ste5 membrane localization in the natural PRS and in the perturbed strains. We constructed strains that expressed Ste5 fused at its C terminus to three tandem copies of YFP (Ste5-YFP- YFP-YFP, here called Ste5-YFP). In previous work, we had used this construct to show that, in cells in the G1 phase of the cell cycle, cytosolic Ste5-YFP translocated to the plasma membrane isotropically within 2–3 min of exposure to isotropic pheromone, followed, after 2–30 min, by clustering of the Ste5-YFP signal into the signaling site (Fig 8A; Ventura et al, 2014). Here, we constructed reference cells that contained this construct, and otherwise isogenic derivatives carrying the $\Delta bim1$, $\Delta gim4$, and tub1-828 expression perturbations. We exposed these cells to a saturating concentration of isotropic pheromone and monitored them by fluorescence confocal microscopy for up to 3 h.

In cells of the reference strain, for the first 45–120 min, a Ste5-YFP patch was visible as a crescent at the tip of the growing shmoo. In most cells, the shmoo tip ceased growing after 45–120 min. In such cells, a second site of polarized growth formed later in a different location. At that location, before polarized growth was apparent, a premonitory Ste5-YFP patch appeared. This second site became progressively brighter while the first site faded (Fig 8A). Cells bearing all three microtubule perturbations showed several common changes. First, a greater number of cells failed to form a signaling site (Fig 8B). This result is consistent with the fact that about 5% of $\Delta bim1$ and $\Delta gim4$ cells did not induce the $P_{PRM1}$-mCherry PRS reporter. Second, in perturbed cells that did form a signaling site, the formation time was greater on average, with more variability between cells. Third, the first site in perturbed strains lasted a shorter time on average and varied more between cells. Fourth, the time from initiation of the first site until appearance of the second was on average shorter and more variable than in WT. By contrast, the perturbed strains showed no changes in the average and variability of duration of the second patch.

These observations showed that cells with microtubule perturbations had difficulties establishing and maintaining their first signaling sites, but not their second sites. This indicates that the three microtubule perturbations we used did not cause a generalized cellular effect that impacted signaling-site formation but rather that they specifically affected the formation of the first signaling site. These results suggest that formation and stability of the second site do not require the microtubule bridge. Overall, these results indicate that microtubule perturbations affect recruitment and retention of Ste5 and are consistent with the idea that their effects on signal are due to their effects on Ste5 function at the signaling site.

## Discussion

We present here the results of a hunt for mutants that affect variability in cell signaling responsive to the activation of a surface receptor by its cognate ligand (Fig 1). For this study, we created a whole genome collection of S. cerevisiae strains, each of which bore the reporter genes required to quantify signaling variables, as well as a deletion in a non-essential gene. We used this collection to screen more than 1,100 non-essential yeast genes, comprising more than 1/4 of the non-essential genome, including all non-essential protein kinases and phosphatases. We isolated and studied carefully 50 mutants. To our knowledge, this was the first large-scale genetic screen for genes affecting variability in receptor-responsive cell signaling phenotypes. Cluster analysis of the reporter gene phenotypes revealed groups of non-essential genes that affected receptor-responsive signaling in distinct ways, suggesting common mechanisms of action for genes within each group.

Some deletions specifically affected pathway variability (cell-to-cell variation in transmitted signal) and others signaling strength (mean transmitted signal), showing that these two aspects of the signal depended on different sets of genes (Fig 3). In this sense, our study defined two quantitative traits, signal strength and pathway variability, as independent axes of system behavior, affected independently by genes, and that also might be affected independently by chemical and environmental perturbations. The fact that mutations exist that can specifically affect the amount of pathway variability suggests that adaptive evolution, for example for increased signaling accuracy, may have shaped the value of this quantitative trait, in the yeast pheromone response and in other signaling systems.

Our screen extends previous work identifying genes that affect variability, as well as mean value of quantitative phenotypes. In S. cerevisiae, work by Raser & O'Shea (2004) showed that deletion of genes whose products participate in the Swi/Snf, Ino80, and SAGA chromatin remodeling complexes increased cell-to-cell variability in expression of a $P_{PHO5}$ reporter gene. Chromatin remodeling is required to induce PHO5 and this result suggests that proper chromatin remodeling is required to suppress variability. Our previous work (Colman-Lerner et al, 2005) showed that mutations in the two PRS MAP kinases differentially affected cell-to-cell variability in signal transmitted to pheromone-responsive $P_{PRM1}$ reporter genes. Our work here showed that the Fus3 kinase was required for the signal variability caused by microtubule perturbations, and suggested that proper assembly of microtubule plus ends at the signaling site is needed to suppress variability. Work by El-Samad, Madhani, and coworkers (McCullagh et al, 2010) showed that deletion of DIG1, but not its paralog DIG2, increased both mean

**A**

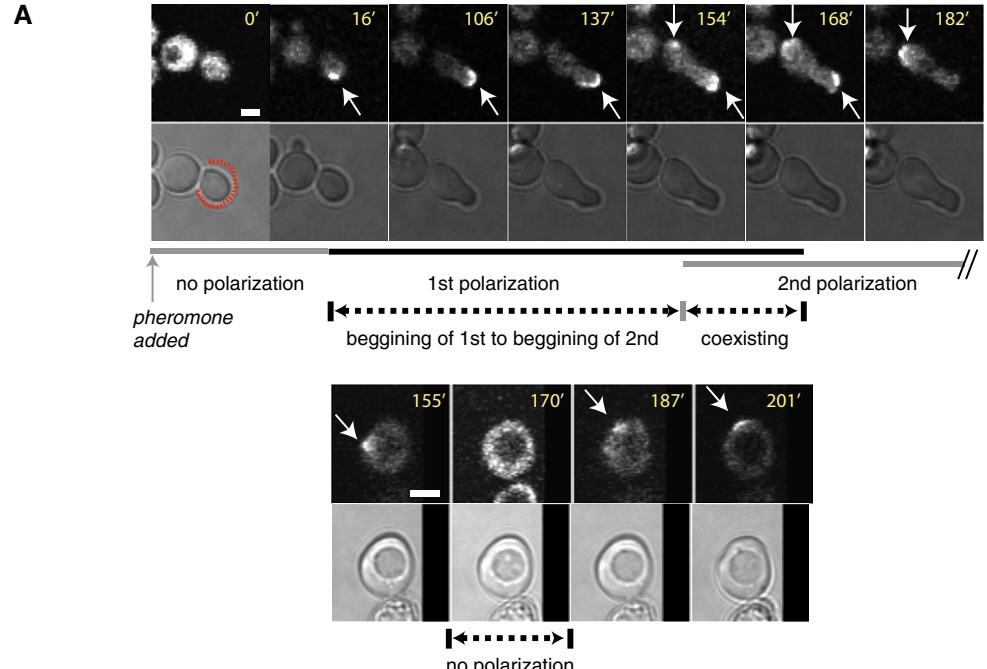

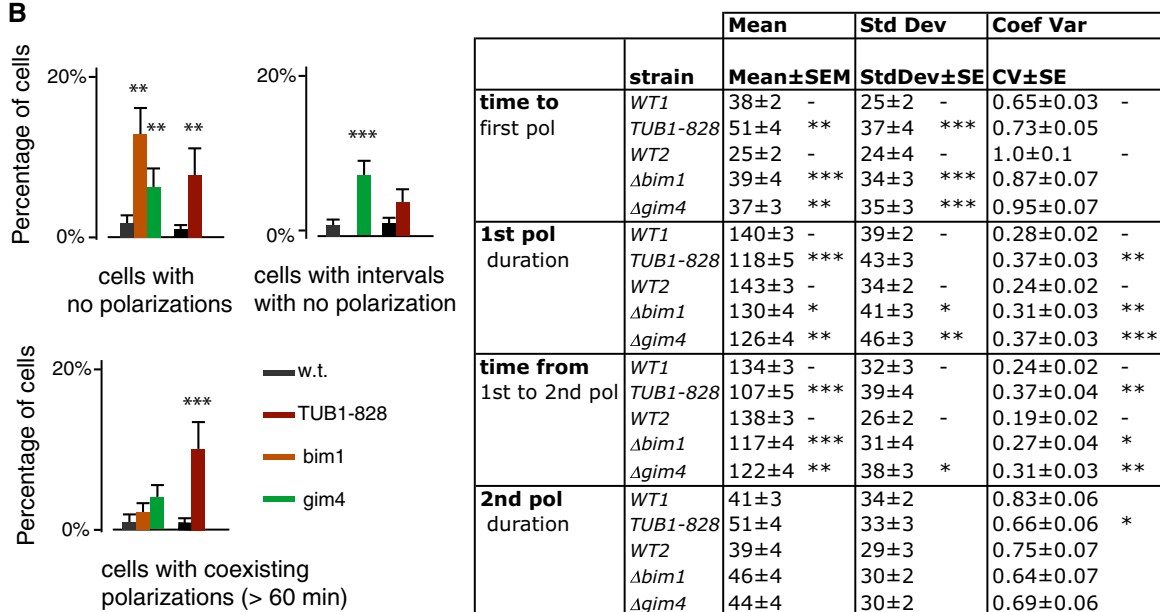

**B**

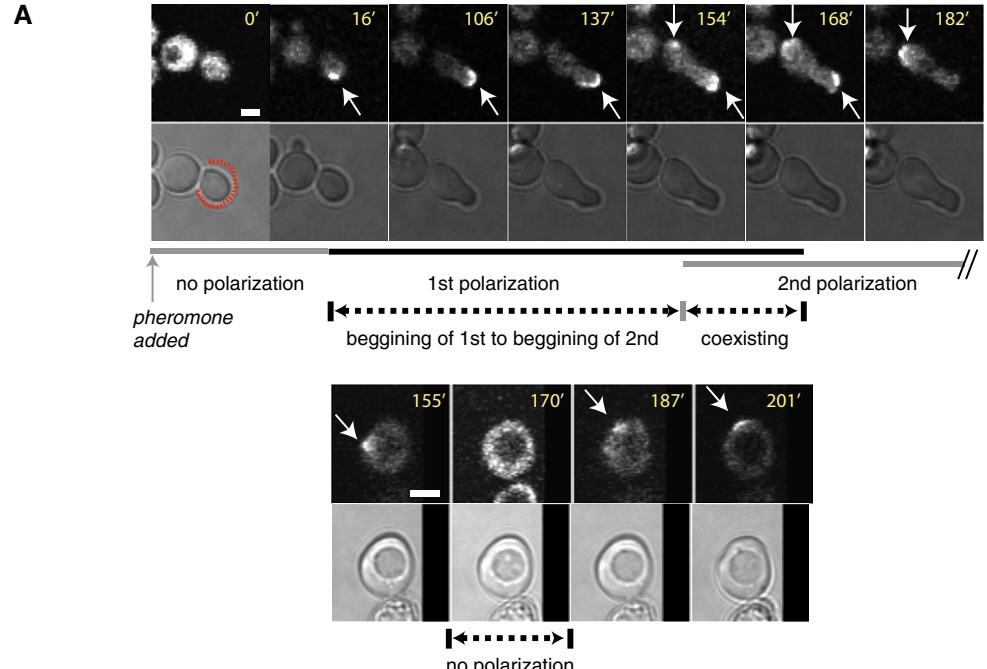

| | strain | Mean | | Std Dev | | Coef Var | |
|---|---|---|---|---|---|---|---|
| | | Mean±SEM | | StdDev±SE | | CV±SE | |
| **time to** first pol | WT1 | 38±2 | - | 25±2 | - | 0.65±0.03 | - |
| | TUB1-828 | 51±4 | ** | 37±4 | *** | 0.73±0.05 | |
| | WT2 | 25±2 | - | 24±4 | - | 1.0±0.1 | - |
| | Δbim1 | 39±4 | *** | 34±3 | *** | 0.87±0.07 | |
| | Δgim4 | 37±3 | ** | 35±3 | *** | 0.95±0.07 | |
| **1st pol** duration | WT1 | 140±3 | - | 39±2 | - | 0.28±0.02 | - |
| | TUB1-828 | 118±5 | *** | 43±3 | | 0.37±0.03 | ** |
| | WT2 | 143±3 | - | 34±2 | - | 0.24±0.02 | - |
| | Δbim1 | 130±4 | * | 41±3 | * | 0.31±0.03 | ** |
| | Δgim4 | 126±4 | ** | 46±3 | ** | 0.37±0.03 | *** |
| **time from** 1st to 2nd pol | WT1 | 134±3 | - | 32±3 | - | 0.24±0.02 | - |
| | TUB1-828 | 107±5 | *** | 39±4 | | 0.37±0.04 | ** |
| | WT2 | 138±3 | - | 26±2 | - | 0.19±0.02 | - |
| | Δbim1 | 117±4 | *** | 31±4 | | 0.27±0.04 | * |
| | Δgim4 | 122±4 | ** | 38±3 | * | 0.31±0.03 | ** |
| **2nd pol** duration | WT1 | 41±3 | | 34±2 | | 0.83±0.06 | |
| | TUB1-828 | 51±4 | | 33±3 | | 0.66±0.06 | * |
| | WT2 | 39±4 | | 29±3 | | 0.75±0.07 | |
| | Δbim1 | 46±4 | | 30±2 | | 0.64±0.07 | |
| | Δgim4 | 44±4 | | 30±2 | | 0.69±0.06 | |

**Figure 8. Microtubule perturbations cause Ste5 patches to form less reliably, delay patch formation, and cause patches to persist for less time.**

We stimulated reference ("WT"), Tub1-828-expressing, Δbim1 and Δgim4 derivatives of MW003 (bearing three copies of $P_{STE5}$-STE5-3xYFP, Ventura et al, 2014) with 1 μM pheromone and imaged them over time for up to 3.5 h.

A    Single-cell measurements of Ste5 patch dynamics. Arrows indicate first and second Ste5 patches. Images show examples of a cell with two co-existing Ste5 patches (top) and a cell with an interval without detectable Ste5 patches between the first and second patch (bottom). The lines below mark the dynamic features we quantified: time to 1st patch (polarization), duration of 1st and 2nd polarization, interval in which 1st and 2nd polarization overlap, or gap between them, and time from 1st to 2nd polarization. Scale bar: 2 μm.

B    Bar graph plots show qualitative defects observed: cells with no polarization, cells with gaps between first and second polarizations and cells with overlapping (coexisting) polarizations. Error bars represent the standard error as calculated by bootstrapping ($10^4$ resamples), and asterisks indicate significant difference from WT as calculated by Fisher's exact test for count data. Table shows quantitative defects. Data correspond to the mean ± SEM, the standard deviation ± SE and the coefficient of variation (standard deviation divided by the mean) ± SE. Standard errors were calculated by bootstrapping ($10^4$ resamples), and significant differences from WT were calculated by permutation tests ($10^4$ permutations). Values for the probability P of the observed data under the null hypothesis that each mutant strain is no different from WT (WT1 vs. TUB1-828, and WT2 vs. Δbim1 and Δgim4) are shown by asterisks: *P < 0.05; **P < 0.01; ***P < 0.001. Experiments were done in three biological replicates (N = 3). As no important differences were observed among replicates, cells were pooled for the analysis. At least 80 cells of each strain were quantified (N > 80).

expression and cell-to-cell variability in uninduced pheromone-responsive $P_{FUS1}$ and $P_{AGA1}$ reporters, suggesting that Dig2 may have lost the ability to decrease variability after the whole genome duplication early in the species history. Work by Yvert and coworkers (Ansel *et al*, 2008) in different strains mapped three loci that increased cell-to-cell variability in expression of a $P_{MET17}$ reporter; surprisingly, one of these loci, crossed into S288C, reduced variability, suggesting that this locus might regulate variability more directly, rather than functioning in a process that affects it. Numerous studies in plants and animals, including our work in *Caenorhabditis elegans*, have revealed genes and alleles that affect variability in gene expression (Fraser & Schadt, 2010; Jimenez-Gomez *et al*, 2011; Mendenhall *et al*, 2017) and variability in other quantitative traits from plant height, flowering, and leaf number, to the number of somatic cells present in fresh cow milk (Hall *et al*, 2007; Ansel *et al*, 2008; Ordas *et al*, 2008; Fraser & Schadt, 2010; Jimenez-Gomez *et al*, 2011; Makumburage & Stapleton, 2011; Landers & Stapleton, 2014). These results make the point that there is much to be learned about mechanisms that affect variability in quantitative traits, and about circumstances in which allelic differences that increase or decrease this variability have general or intelligible effects on organismic fitness.

This work uncovered an unexpected contribution of cytosolic microtubules to pathway variability and signaling strength. In normally signaling cells, a microtubule bridge connects the signaling site on the cell membrane with the SPB in the nuclear envelope. Cells with perturbations in microtubule function showed increased variability in transmitted signal and/or reduced mean signaling strength. In particular, cells with deletions in *BIM1* and in *GIM4* showed greater cell–cell variability in transmitted signal in populations of single cells at a single time point. Monitoring the

"trajectories" of accumulation of pathway output in these mutant cells over time revealed a broader distribution of transmitted signal and, in some cells, "jumps" in accumulated pathway output, evidenced by crooked trajectories, defining erratic operation of the signaling system.

The effect of microtubule perturbations on pathway variability was specific to the operation of the native pheromone response system. For example, pathway variability was not affected by some microtubule perturbations when the signal was triggered by artificially membrane-anchored Ste5 (Fig 6). Furthermore, direct microscopic observation of Ste5 at the signaling site showed that, in microtubule perturbed cells, the initial Ste5 patches at the signaling site were slower to form and more likely to disappear (Fig 8). Future experiments using faster maturing and shorter lived fluorescent protein derivatives might enable temporal correlation of transmitted signal increase and decrease with formation and loss of Ste5 patches. In the absence of such data, our results showed that perturbations that increased pathway variability impaired Ste5 accumulation at the signaling site and that their effects on variability were suppressed by artificial recruitment of Ste5 to the membrane. Moreover, variability in signaling caused by the perturbations required the MAP Kinase Fus3: in cells that lacked Fus3, the Δ*bim1* and Δ*gim4* perturbations did not increase pathway variability (Fig 7). These two facts suggest that microtubule perturbations cause variable signaling by Fus3, when activated by the MAPKK Ste7 and the MAPKKK Ste11 in complex with membrane-recruited Ste5.

We propose that small irregularities in Ste5 recruitment or Fus3 signaling caused by microtubule perturbations might become amplified into larger differences in Ste5/Fus3 dependent signaling by positive feedback. Figure 9 shows some of the stimulatory reactions by

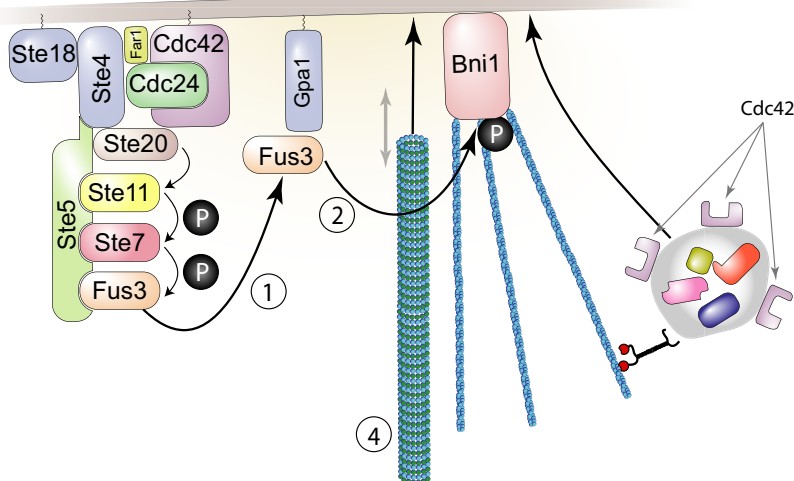

**Figure 9.  Model for origin of pathway variability.**

Our work suggests that disruption of microtubule plus-end function near the signaling site by perturbations such as the Δ*bim1* mutation causes variability in Ste5 recruitment or Fus3 signaling. Figure shows stimulatory reactions at the signaling site by which irregularities in Ste5 recruitment or Fus3 signaling could be amplified. (1) Fus3 is phosphorylated and activated due to the operation of the PRS. Some phosphorylated Fus3 binds to the Gpa1 subunit of the dissociated G-protein. (2) Gpa1-bound Fus3 activates a protein, Bni1, which nucleates formation of actin cables. (3) Additional proteins involved in signaling, cell polarization, and cell fusion, including Cdc42 and Fus2 (Paterson *et al*, 2008, not shown) are then trafficked to the membrane as cargo carried along the actin cables. Activated Cdc42 stimulates its own activation (Kozubowski *et al*, 2008; Johnson *et al*, 2011). (4) Microtubule plus ends are captured by Kar3/Cik1 associated with Gpa1 (not shown) and Kar9/Bim1-complexed plus ends can be walked down the actin cables to the signaling site by motor proteins including Myo2 (not shown). Microtubules, actin cables, and cargo proteins are larger than the approximate scale used here would suggest. Appendix and Appendix Fig S7 describe additional stimulatory reactions and positive feedbacks that might contribute to irregular signaling at the site.

which irregularities in Fus3 signaling can be amplified, including the stimulated membrane recruitment of Cdc42, which has previously been shown to stimulate its own activity. (Kozubowski *et al*, 2008; Johnson *et al*, 2011). In this view, disruption of plus-end function of microtubules near the site might cause differences in delivery of signaling components; such differences would then become amplified. In controlled systems, including servo systems (Hanzen, 1934) and electronic amplifiers (Armstrong, 1914; Franklin, 1914; Meissner, 1919), positive feedbacks can improve performance, but without damping or negative feedback, such systems are prone to instability (Bennett, 1979). In the PRS, impact of positive feedbacks on system performance is further affected by the fact that system components are in limited supply [for example there are only ~2,000 molecules of the Gβ, Ste4, per cell (Thomson *et al*, 2011)], so that the autostimulatory molecular events that generate a robust second signaling site can only occur after a first site falls apart.

The mechanisms by which microtubules increase transmitted signal strength and reduce signal variability may be distinct. Some perturbations that impaired (but did not eliminate) the microtubule bridge increased pathway variability but did not affect the strength of the transmitted signal (Δ*bim1*, Δ*kar3*, *TUB1-828-expression*), while two perturbations that eliminated the microtubule bridge (*kar1-Δ15* and microtubule depolymerizing drugs) did not affect pathway variability but reduced transmitted signal strength. This finding is consistent with the initial results of the genetic screen, in which transmitted signal strength and pathway variability can be affected differently by different mutations (Fig 3). These effects on pathway variability are not apparent at low doses. This fact suggests that cells may have a microtubule-independent way to transmit signal that is weaker than the normal mode, but that functions with low pathway variability as well. Such an alternative mode of signaling might be helpful if attempts to signal by the microtubule-dependent mode had failed. The Ste5 dynamics results (Fig 8) are consistent with this idea: cells form first a Ste5 patch that is impaired by microtubule perturbations and then a second patch that is impervious to them. This hypothesis could be tested: it predicts that the Ste5 patches should behave normally under perturbations that eliminate the microtubule bridge (*kar1-Δ15* and microtubule depolymerizing drugs). In any case, a weaker transmitted signal in the absence of microtubules might enable cells to delay cell fusion until the nucleus is properly tethered to the membrane fusion site, a required step for zygotic nuclear fusion.

How might microtubules strengthen signal transmission? Just as they facilitate transport to other membrane sites (Maekawa *et al*, 2003; Cavalli *et al*, 2005; Foe & von Dassow, 2008), microtubules might facilitate transport of signaling molecules to the signaling site. Here, we considered a non-exclusive alternative idea, that the microtubule bridge might position the nucleus (Maddox *et al*, 2003) at an optimal location with respect to the signaling site and a gradient of activated MAP kinase emanating from the site (Maeder *et al*, 2007). We found a lack of correlation between distance to the signaling site and pathway output in cells with destabilized or frozen nuclear positioning (Appendix Fig S6). However, it is possible that some of the perturbations we used might have interfered with a MAP kinase gradient. For example, the frequent forays of the unattached nucleus in Δ*bim1* (Maddox *et al*, 2003) might stir the cytosol and disrupt a gradient. It thus remains possible that in

unperturbed cells nuclear positioning strengthens signal by positioning the nucleus higher up a signal gradient.

Our current means to perturb the system and monitor its operation are not sufficient to elucidate how normal operation of cytoplasmic microtubules helps the cell transmit signal of constant strength. Too much remains unknown. At the signaling site, there are too many different proteins operating, too many positive cross-regulatory interactions, too many simultaneously occurring mechanical processes like cargo delivery and membrane fusion that are now insufficiently understood. It is as if we had tried to understand the smooth function of an electric motor by monitoring frequency and timing of sounds it made after disrupting the operation of particular bearings, bushings, and shafts. In this light, analysis of microtubule effects on signal transmission is a classic "inverse problem", for which inferences from doable experiments are limited and insufficient to fully describe the system under investigation (Brenner, 2010). Within these limits, however, our genetics-powered quantitative physiological experimentation enabled us to identify the proteins involved, and this in turn helped us constrain models for their function. Moreover, as in the motor analogy above, different kinds of noises may well identify aspects or axes of system function dependent on different proteins and molecular events, and so perhaps contribute to future insight.

It is possible that cytoplasmic microtubule function may affect pathway variability in metazoans, for which coherent population fate and polarity decisions are needed for forming and maintaining correct tissue architecture. For example, vertebrate orthologs of yeast *BIM1*, MAPRE1-3/EB1-3 (Su & Qi, 2001) interact with APC (Adenoma Polyposis Coli), which is required for radial glial cells polarization, and for them to support birth and migration of cortical neurons (Yokota *et al*, 2009). APC is a tumor suppressor, frequently inactivated in colorectal and other epithelial cancers (Kinzler & Vogelstein, 1996; Vogelstein *et al*, 2013). At the minus end, lesions in proteins that connect microtubules to the microtubule organizing center (in particular those affecting the Nesprin-1 and Nesprin-2 isoforms encoded by SYNE1 and SYNE2 (reviewed by Gundersen & Worman, 2013) contribute to formation of solid tumors in which tumor development requires incorrect polarity decisions [e.g., squamous cell carcinomas of the head and neck (Stransky *et al*, 2011)]. Exome sequencing reveals considerable coding sequence polymorphism in genes encoding the *BIM1* ortholog MAPRE/EB1, and in APC, and other microtubule end-interacting proteins in the human population (Fu *et al*, 2013), raising the possibility that different allelic forms of microtubule end proteins might have different quantitative effects on variability in cell decisions in response to signals, and so affect cancer incidence. In this light, the existence of alleles affecting pathway variability may help motivate development of genetic and pharmacological interventions aimed at reducing it.

## Materials and Methods

We performed DNA manipulations including PCR and subcloning as described (Ausubel *et al*, 1987–2017). We cultured and manipulated yeast as described (Ausubel *et al*, 1987–2017; Guthrie & Fink, 1991). Unless otherwise noted, we grew cells in synthetic dextrose complete (SDC) media consisting of Brent Supplemental Media (MP Biomedicals, Solon, OH), yeast nitrogen base without amino acids

and ammonium sulfate (BD, Franklin Lakes, NJ), and dextrose (Sigma-Aldrich, St Louis, MO).

## Analysis of cell-to-cell variability

We performed the analysis as in Colman-Lerner et al (2005). Briefly, we considered the system output for any given cell $O_i$, determined by the abundance of a fluorescent protein inducible by the pheromone response system, to be the product of (i) the average pathway subsystem output per unit time, $P_i$ (which varies with input pheromone dose $\alpha F$), (ii) the expression subsystem output $E_i$, and (iii) the duration of stimulation $\Delta T$ (Colman-Lerner et al, 2005), as follows:

$$O_i = P_i(\alpha F) \times E_i \Delta T$$

We considered $P_i$ and $E_i$ to be the sum of the capacity of the subsystem in each cell ($L_i$ and $G_i$) plus stochastic fluctuations in the operation of each subsystem during the course of an experiment ($\lambda_i$ and $\gamma_i$). Thus,

$$O_i = (L_i(\alpha F) + \lambda_i) \times (G_i + \gamma_i)\Delta T$$

We defined the cell-to-cell variability in system output as the normalized variance of $O_i$, $\eta^2(O)$, decomposable into the sum of individual sources and a correlation term (Colman-Lerner et al, 2005), as follows,

$$\eta^2(O) = \eta^2(L) + \eta^2(\lambda) + \eta^2(G) + \eta^2(\gamma) + 2\rho\eta(L)\eta(G)$$

In the WT, in the deletion strains, and mutants used in the manuscript, we measured output and cell-to-cell variability of each reporter; $\eta^2(\gamma)$, gene expression noise ["intrinsic noise" (Elowitz et al, 2002)]; as well as $\eta^2(P)$ ($\eta^2(L) + \eta^2(\lambda)$), cell-to-cell variability of the pheromone response system. We measured $\eta^2(\gamma)$ as the variance in the difference of the normalized abundance of the two fluorescent proteins driven by identical copies of $P_{PRM1}$. We estimated cumulative signal transmitted P in individual cells as the normalized signal from the pheromone-inducible $P_{PRM1}$ reporter (O) divided by the signal from the constitutive control promoter, ($P_{ACT1}$ or $P_{BMH2}$, depending on the strain) (O/G). We estimated $\eta^2(P)$ as the variance in the difference between the normalized abundances of two fluorescent proteins, one driven by the $P_{PRM1}$ and the other by the constitutive, pheromone-independent promoter ($P_{ACT1}$ or $P_{BMH2}$, depending on the strain) ($\sigma^2(mRFP_i/<mRFP> - YFP_i/<YFP>)$). This variance is actually equal to $\eta^2(P) + \eta^2(\gamma)$ (see Appendix), but $\eta^2(\gamma)$ was low enough in the WT and the mutants in which we measured it to assume that it may be neglected (Colman-Lerner et al, 2005).

## Construction of Heterozygous Diploid Deletion Variability Collection and its use to generate sets of haploid deletion strains for screening

We constructed a MATα strain, SGA88, which carried two pheromone-inducible reporter genes, one constitutive reporter gene, a bar1- mutation which blocked a protease that removed pheromone from the extracellular medium, and a cdc28-as2 mutation which allowed us to block the inhibition of the pheromone response by the

cell cycle machinery by adding to the cells a chemical inhibitor of the mutant protein kinase. In SGA88, all of these genetic elements and the *MATa* marker were linked to individually selectable recessive (nutritional auxotrophy) or dominant (antibiotic resistance) markers. We mated SGA88 to a fresh instance of the original ("1.0") haploid deletion collection (Chu and Davis, 2008, a gift of Amy Chu) to create the Pesce Heterozygous Deletion Diploid Variability collection (PHDDV collection), comprised of more than 4,100 diploid strains. In these diploid strains, three dominant resistance markers: $hygB^R$, $G418^R$, $nat^R$, and two recessive markers, *his3* and *leu2*, allowed selection of genetic elements, while two dominant sensitivity markers: canavanine$^s$ (due to the *CAN1* allele) and thialysine$^S$ (due to the *LYP1* allele) allowed selection against unsporulated diploids. We then sporulated different members of the HDDV collection on appropriate selective media to generate haploids that bore the deletion and the other genetic markers needed for the screen. We picked these as individual small colonies on selective plates and assayed individual cultures grown from these colonies.

To screen for mutants that affected cell-to-cell variability in pathway output, we grew cells in log phase ($< 3$–$10^6$ cells/ml) for at least 14 h. This step is in contrast to the standard practice of diluting carbon-exhausted cultures 4–6 h prior to measuring them. By relying on exponential phase cultures we minimized undesired variability in PRS output arising from strain-to-strain and day-to-day differences in time to enter the exponential growth phase. We exposed our cultures for 3 h to two different pheromone concentrations (0.6 nM or 20 nM) and 10 μM *cdc28-as2* inhibitor 1-NM-PP1. We then added 50 μg/ml cycloheximide to inhibit protein synthesis and allowed for existing translated fluorescent protein molecules to mature (Colman-Lerner et al, 2005; Gordon et al, 2007). To aid the mutant screen and follow-up experiments, we measured the maturation times of mRFP (strain collection) and mCherry (follow-up experiments) after blocking protein synthesis with cyclohexamide as in Gordon et al (Gordon et al, 2007). Measured 1/2 time to maturation was 120 min (mRFP) and 45 min (mCherry).

We measured fluorescence signal from the $P_{PRM1}$-*mRFP* and $P_{ACT1}$-*YFP* reporters by cytometry (BD LSRII with HTS auto-sampling attachment) and calculated or estimated parameters of interest, such as system output $O_i$ and cell-to-cell variability in signal transmission, $\eta^2(P)$, as described above. We then verified (by cytometry) altered behaviors in three additional clonal isolates from the same mating, as described above. We confirmed by PCR in a random strain from the set of four for the presence of the expected deletion and the absence of the wild-type coding sequence. We checked this strain by image cytometric fluorescence microscopy at the two different doses to confirm lack of aggregation and to measure $P_{PRM1}$-*CFP* signal. Measurement of CFP signal allowed us to determine if the mutants affected $\eta^2(\gamma)$. As described, $\eta^2(\gamma)$ was a small contributor to cell-to-cell differences in gene expression and no mutant affected it.

## Data availability

Datasets sufficient to reproduce all plots in this paper are provided as Source Data files or Appendix tables. The flow cytometry data for individual strains in the screen are available at the Dryad Digital Repository, https://doi.org/10.5061/dryad.67bc0 and at http://authors.fhcrc.org/1202/ with DOI: 10.6076/j77d2s8q.

Expanded View for this article is available online.

## Acknowledgements

We are grateful to Amy Tong, Charlie Boone, Guri Giaever, and Corey Nislow for strains, plasmids, advice, and previously unpublished information that aided construction of the whole genome deletion reporter collection, Steve Andrews, Alexander Mendenhall, and Alan Bush; to SGD staff, including Mike Cherry and Stacia Engel, for valuable discussions throughout; and to Christopher Neils (UW) and Joerg Stelling (ETHZ) and Anne Stapleton (NCSU) for valuable discussions on control theory and canalizing genes. Work was supported by R01 GM097479 to RB and PICT2013-2210 from the Argentine Agency of Research and Technology to ACL. Earlier work received support from grants R01 GM086615 to RB and RCY and from P50 HG002370 to RB.

## Author contributions

CGP designed and constructed the strain collection, performed the screen, analyzed the results, and characterized the selected mutants. DR assisted in this work. CGP and SZ designed and performed the microtubule perturbation experiments. MVR and AB designed, performed and analyzed the Ste5-YFP patch experiments. WJP analyzed the numerical data, developed statistics for describing distributional shape, system stability, and comparing datasets in Figs 6 and 7. RCY contributed to experimental design, data analysis, and earlier versions of the manuscript. RB, CGP, and AC-L directed and guided the work and its interpretation. CGP, WJP, AC-L, and RB wrote the paper and guarantee the integrity of its results.

## Conflict of interest

The authors declare that they have no conflict of interest.

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
