## [Review Process File · Molecular Systems Biology]

Single-cell profiling screen identifies microtubule-dependent reduction of variability in signaling

Pesce, C. G., Zdraljevic, S., Peria, W. J., Bush, A., Repetto, V., Rockwell, D., Yu, R. C., Colman-Lerner, A., and Brent, R.

Review timeline:

Manuscript MSB-15-6386	Submission date:	20 June 2015
	Editorial Decision:	23 July 2015
Manuscripts MSB-16-7390 & MSB-16-7391	Submission date:	15 October 2016
	Editorial Decision:	6 February 2017
Merged Manuscripts MSB-16-7390 & MSB-16-7391	Submission date:	20 November 2017
	Editorial Decision:	5 January 2018
	Revision received:	25 January 2018
	Accepted:	6 February 2018

Editor: Thomas Lemberger

Transaction Report:

MSB-15-6386

MSB-15-6386, Editorial Decision

23 July 2015

Thank you again for submitting your work to Molecular Systems Biology. We have now heard back from four referees who agreed to evaluate your manuscript. As you will see from the reports below, the referees find the topic of your study of potential interest. They raise, however, substantial concerns on your work, which should be convincingly addressed.

Without repeating all the points raised in by the reviewers, the major issues refer to the following points:

- one major limitation of the current study is the absence of in-depth mechanistic follow up of the microtubule-dependent effects. Coordination with your follow up study on these mechanistic aspects would thus be very helpful in this regard.
- the data should be presented in full; in particular the results of the microscopic measurements of the 50 mutants analyzed should be presented. These data should also be made available as well as the FACS data.
- given potential defects in chromosomal segregation, the bim1 and gim4 mutants may harbor additional genetic defects that could confound the analysis. The possibility of such an artefact should be excluded.

- some of the claims have to be softened, for example about independent effects on noise and means response. A better visual presentation of the data would also be helpful. In addition, supplying source data files that provide the individual numerical values displayed in figure 4D would be helpful as well.

- the reviewers all felt that the text need a profound rework to eliminate jargon and considerably simplify the narrative to achieve a clear and concise presentation. This concerns the Introduction and the Discussion sections but also to some extent the Results section. Redundancies between these sections should be avoided and methodological details should be provided in the Materials and Methods section.

If you feel you can satisfactorily deal with these points and those listed by the referees, you may wish to submit a revised version of your manuscript. Please attach a covering letter giving details of the way in which you have handled each of the points raised by the referees. The revised manuscript will be once again subject to review and you probably understand that we can give you no guarantee at this stage that the eventual outcome will be favorable.

REVIEWER REPORTS

Reviewer #1:

This study explores the genetic factors that influence the degree to which individual cells in a clonal population show variation in their response to an external stimulus. For this, the authors use the yeast pheromone response system (PRS), a thoroughly studied signaling system with numerous advantages relevant to the issues under consideration. The authors screen a large number (roughly 1000) of viable deletion mutants, and identify a subset (roughly 50) that affect total response and/or cell-to-cell variability, which then partition into various subgroups with distinct effects on specific phenotypic variables. Most of the remaining effort is focused on the subset of genes that alter microtubule functions. The observations suggest that genetic perturbations can have distinct effects on the strength versus variability of signaling response, and that not all microtubule-related functions alter these parameters in the same way. By the end, it is evident that signaling strength and variability can be altered by mutations in several microtubule-affiliated genes, although the causal mechanisms are not probed further.

Clearly, the authors have done a lot of work, and I have no reason to doubt the technical aspects of the experiments or the mathematical analyses. But I did have trouble determining exactly what has been learned and what new insights were revealed. This difficulty stems mainly from the fact that, while the phenotypes from microtubule-associated genes are intriguing, the possible mechanistic causes were not probed and hence remain highly speculative. In addition, there are issues with the presentation: (a) The manuscript contains such an overwhelming deluge of information, some of which seems largely dispensable, that it ends up obscuring rather than clarifying the overall goals and findings. (b) The text suffers from being bloated, jargon-filled, and rather grandiose in places (especially the Discussion); it could be substantially improved by trimming the excesses and consolidating the central points into a more coherent narrative. Collectively, these features make it difficult to offer a strong positive recommendation.

Specific points on the manuscript:

1. In the long Introduction, it is frequently unclear why all of these numerous prior points are being described in such great detail. I suspect that much of it could be deleted. If prior observations are important for this paper, their relation to the present study should be clearly stated at the outset, in order to provide context and clarify their relevance.
2. The Discussion suffers in three ways: (a) it lacks a clear and concrete summary of the key findings and whatever resulting new insights or conceptual advances they might offer; (b) it contains unnecessary repetition of methodological points that are already covered in the Intro and Results; and (c) it is filled with long musings on a range of issues that relate to phenotypic variation but do

not really seem to be directly advanced by the findings in this manuscript.

3. The Results section at times includes excessive methodological detail, often redundant with Methods and/or Figure Legends, instead of a clear description of the purpose of the experiment and the information obtained. For example, the GIM4/BIM1 (not "BIM4") section on page 9-10 includes discussion of the choice of strain, the reporters, the pheromone concentrations, time intervals, etc. But the take-home message remains rather nebulous: "despite their similarity in function and phenotype, BIM1 and GIM4 have non-redundant role [sic] in controlling signal variation" (Is that all that has been learned? What does it mean? How has our understanding been advanced?).

4. Figure 3 seems dispensable. It is mentioned only extremely briefly; once as a half-sentence (page 6, top), and once largely as an aside (page 9, bottom half). Currently, readers can only guess what the Figure represents and what is its point. If the authors think it provides some value, then this should be described; otherwise, its purpose is unclear.

5. Similar to point #4, Figures 2A and 2B are never mentioned, and Figure 4 is mentioned in only one sentence (page 6, middle). If they contain important information, then their contributions should be explained.

6. Page 10, middle: The description of results in Figure 6D does not match the plots shown. The authors state that the distributions are "significantly more long-tailed" and then continue to refer to "long tails", a "heavy tail" and "outlier cells comprising the tail". But there are no "tails" evident in these plots, and so this jargon is based on an extrapolation of the results (e.g., to a distribution curve) rather than what is actually evident in the plot shown. And the meaning of "heavy tail" is unclear.

7. Page 11, second paragraph: (a) the reference to "GPY4000 reference cells" is not only an unnecessary detail that could be left to Methods or Legends, but it seems to conflict with the Figure 7A legend (which claims to use "SGA103" cells). (b) "This result...was consistent with GIM4 controlling variation..." . This seems incorrect, as the preceding sentence refers to signal strength, and so that result cannot be said to agree with results on signal variation. (c) The choice to emphasize a seemingly minor point (about slight variation in effects of microtubule poisons) is confusing; e.g., is this level of experimental variability really notably greater than that seen in the 3 repeats of the *bim1* Δ strain?

8. Page 15, top paragraph: The last sentence ("Furthermore, our interest...") does not make sense; it is unclear what "showing that" is meant to refer to.

9. Page 19, top paragraph: The last sentence ("Thus, previous studies...") does not make sense. Specifically: "genes that reduce variation due to differences in genotype" is not coherent.

10. Page 5 bottom paragraph: "a third fluorescent protein" should explicitly mention YFP; otherwise, the later reference to measuring "mRFP and YFP signal" will not make sense to readers.

11. Page 26: The Figure 2D legend is incomplete; it does not mention the pheromone output measurements that are plotted, and it does not describe the difference between the 3 curves plotted.

12. Page 26 and Page 28: In the legends to Figures 3 and 7, the authors seem to be conflating significance levels (i.e., p-values) with confidence intervals in an unusual way. I am not sure there is a strict meaning to the phrases "0.95 confidence statistical significance of differences" and "significant at 0.95". These seem like imprecise shorthand.

13. Page 28 and Figure 6C: (a) The Figure legend should explicitly describe the distinction between the three plots shown. (E.g., "colors as in panel A".) (b) The legend should also describe what the dots denote. Presumably, these are individual cells in a clonal population, but if so then this should be stated clearly and explicitly, as otherwise it is ambiguous what is being represented in these plots. (c) Finally, the grey dots (wild type) are difficult to distinguish from the colored dots; perhaps black would be better.

Reviewer #2:

In this work, Pesce and colleagues perform an impressive high-throughput screen to find genes related to cell-to-cell variability of pheromone response in yeast. In more than 1000 single-gene deletion strains, they measure pathway variation separately from other sources of noise, notably cell-to-cell variation in total gene expression capacity. They accomplish this by including the same pheromone responsive promoter driving the expression of two different fluorescent reporters and a constitutively active promoter driving the expression of another fluorescent protein in each deletion strain. The genetic screen itself is carefully executed, with careful consideration given to cell density effects (deletion of BAR1) and the important role of cell cycle on pheromone pathway (analog sensitive Cdk). It is apparent that this screen will be an important resource for the community working on cell-to-cell variation in general and the pheromone pathway in particular.

While the genetic screen itself was carefully executed, I have some major reservations for some of the following analysis and the results that the authors find. Specifically, the two main claims of the paper, i.e., "signal variability and strength [as] independent axes of system behavior" and "microtubule-dependent mechanism that limits signal variation," are not that strongly supported by the authors' data. However, rephrasing conclusions more carefully and some reanalysis of the data can rectify these problems.

Major points

a) As the authors also show in Figure 2D, the output of the pathway is inversely correlated with pathway variation. Thus, the majority of what the authors select in Figure 4 as the 'outlier' hits is not necessarily genes that affect pathway variation independent of pathway output. The claim that signal variability and strength are "independent axes" of system behavior is only supported by a few hits that lie outside of the inverse relationship cloud in Figure 4D. Instead of choosing many genes based on 1D metrics as shown in Figures 4A-C (most of which are simply genes that affect system output and affect variation through its effect on system output), the more relevant choice would be to select only those deletions that move a cell in 'y' axis in Figure 4D without a similar move in the 'x' axis. These would be the deletions that change pathway variation independently (at least relatively) of pathway output. To help guide future work based on this screen, the authors could report on Figure 4D which dot each chosen gene corresponds to. They should also rephrase their conclusions on "system variability and strength as independent axes," because most deletions do not support this independence (and they do not show the source of this independence even for the hits that independently affect output and variation). With the given data, it would be acceptable to say that there exist genes that seem to affect pathway variation and output independently. However, it does not follow that pathway variation and output are independent axes.

b) Similarly, the level of the presented data does not match the authors' strong conclusions on "microtubule-dependent mechanism that limits signal variation." The authors do not show a "mechanism" in this paper; they find that mutations that disrupt microtubules, which, in turn, possibly disrupt many different processes, and somehow affect pheromone signal variation (in some cases, somewhat independently of pathway output). The use of the word "mechanism" is warranted only if the authors find the mechanism of this effect on signal variation, on which they speculate on Discussion. The sections that refer to "the mechanism" should be rewritten to reflect this point.

c) It is known that deletions can often pick up additional mutations and aneuploidy. In one study, 8% of strains in a deletion library have been found to have a chromosomal abnormality (Hughes et al, 2000). This is particularly important for *bim1* Δ , which is known to have a defect in chromosomal segregation (Schwartz et al., 1997). Therefore, the authors should do a whole genome sequencing of their *bim1* and *gim4* mutants to make sure that their results on microtubules do not stem from such an artefact.

d) In Figures 3, 6, 7, the more important relation is pathway output vs. pathway variation in different mutants, rather than their separate effects at different pheromone concentrations. The authors should plot pathway output vs. variation for these different strains, and show how significant this difference is in this plot for the wild-type vs. the mutants (e.g., ks-test).

Minor points

- a) The introduction contains some unsubstantiated, vague, or incorrect statements. For instance, the authors write "the importance of rapid-changing stochastic sources of variation is assumed because it is consistent with mathematical models." This sentence insinuates that the importance of such noise is incorrectly applied simply because of mathematical convenience without any citations to support this statement. Similarly, the section on the "action of persistent, slow-changing sources of variation" is vague and can be explained more succinctly and clearly. Finally, the statement that extrinsic noise was attributed to differences in lac repressor in Elowitz et al. (2002) is not accurate (LacI is a contributor to extrinsic noise, and both intrinsic and extrinsic noise increase upon changing LacI concentration).
- b) Because the disruption of microtubules may affect cell morphology, the authors should show pictures of their various microtubule mutant strains (so that the readers can judge whether different mutants have similar morphologies with or without alpha factor)
- c) The authors should better motivate their follow-up on microtubule mutants. Why did this group of deletions seem more interesting than the other groups?

Reviewer #3:

The study of Pesce et al. is a genetic screen for genes that modify noise in the yeast pheromone response pathway. The method is based on a two-color system, where one fluorescent protein reports pathway response and a second one reports constitutive gene expression capacity. Using high-throughput strain construction, culturing and flow cytometry, the authors identified 50 gene deletions showing evidence of noise alteration. They followed-up with time-lapse microscopy on a number of these genes (they say 42 but the results are not presented, see below), and precisely characterized the implication of two genes : BIM1 and GIM4, which regulate microtubule dynamics and folding. The results show that perturbations of microtubules (in the gene deletion strain, or by chemical treatment, as well as by overexpressing a dominant negative genetic perturbator) generates variability of the pathway response.

The experimental work is very abundant and the conclusion is important : linking microtubule functions to molecular pathway transduction variability opens questions relevant to all eukaryotes.

Major critics :

1. The authors used flow cytometry as the primary screening steps and identified 50 genes which deletion modified pathway variability. This list is based on cutoff thresholds. The text says that 42 were then studied by microscopy but the results obtained on these 42 strains are not provided. The Supplementary Information details only the experimental method and the gene names. How many of the 42 were validated and on what basis ? All results from the microscopy acquisitions and analysis should be presented, not just the trajectories of bim1 and gim4 strains.
2. The raw flow cytometry data should be deposited in a public repository, such as <http://flowrepository.org/>. This will ensure reproducibility of the analysis results, especially since the methods do not detail data preprocessing such as gating, compensation or background correction (if any).
3. There are conceptually two very distinct ways to generate pathway response diversity when perturbing microtubules. i) The genetic perturbation could generate abnormal cellular internal organization (e.g. failure to polarize the nucleus) in many cells and these cells could then respond more or less efficiently to the pheromone signal because of this mis-organization of their content. This is variability prior to the stimulation and differs from ii) The genetic perturbation could impact the precision with which the signal is transmitted. For example, if the active Fus3p gradient described by Knop is critical for precise signalling and if this gradient cannot be well established. This variability is posterior to the stimulation. The authors could examine the cell morphology of bim1, gim4 and related mutants from the data of Ohya et al. PNAS 2005 (<http://yeast.gi.k.u-tokyo.ac.jp>). If variability in nucleus position or actin distribution is reported in these mutants in the

absence of pheromone, this would argue for i).

4. The possible mechanisms are not fully discussed :

- The main implication of microtubules organization in response to pheromone is probably to get ready for karyogamy. The authors may have found a piece of evidence of a feedback control of proper MT architecture on the pheromone signalling pathway. Cells would benefit from such a feedback, so that MTs have time to be properly organized before cell fusion.

- In wild-type cells, the nucleus moves to the tip via oscillations whereas in bim1- cells it most likely moves continuously in one direction because pushing forces are abolished. Oscillations can homogenize molecular concentrations in viscous media (cytosol). Could it be that pulling-only causes situations where local concentrations are extremes (bad diffusion and spatial heterogeneity of concentrations in the cytosol) ?

- Could the nuclear export of Fus2p to the shmoo tip (Paterson et al. 2008 JCB) be implicated ? Paterson et al. showed that Fus2p movements take place under nocodazole treatment, but this does not rule out a change of fine properties such as cell-cell variability. Is it possible that MT dysfunctions perturb actin filaments architecture and, in turn, Fus2p localization at the tip could be altered or delayed in some cells. Would that affect their dynamics/strength of pathway activation ?

Less major points :

- First half of second paragraph page 10. The description of the distributions of the spread of trajectories is hard to follow : tails of distributions are mentioned and readers need to imagine the distributions. Better to explain clearly PSD and MPS directly, or better phrase what these tails are.

- Interpretation of symmetrical (bim1) versus below-only (gim4) crookedness. The half-life of the fluorescent reporters used should be mentioned : if fluo proteins are very stable (especially for PRM1 activity), only acceleration of production can reliably be detected, not deceleration.

- Fig6D : ACT1 is indicated in the y-axis and legend whereas text page 9 bottom says that BMH2 promoter was used. Which one was used ? Also, crookedness can be caused by irregularities in constitutive promoter signal (denominator). How do the authors rule this out for the trajectories presented on the figure ?

- The review of Molk and Bloom J. of Cell Science 2006 (119) is very helpful to imagine microtubules arrangements in the shmoo. It should be cited to guide non-specialist readers.

- The title of the study differs between the manuscript file and the SI file. « Signal variation mutants identify microtubule dependent mechanisms required for accurate cell fate decisions » is not appropriate : the cellular decision itself (fusion) was not quantified, and we already knew that MT mechanisms are essential for karyogamy. « Genetic screen identifies microtubule-dependent mechanism that limits signal variation » does correspond to the study.

- Pathway variation (CV2 of mRFP/<mRFP> - YFP/<YFP>) is sometimes notated $n^2(P)$ (Figure 5) and $n^2(P) + n^2(\gamma)$ (Figures 3, 6B ?, 7?, 8 ?). Be more precise and consistent on the quantities reported.

- page 9 subtitle. BIM4 should be BIM1

- page 10 bottom : « average » appears twice.

- page 15 : two thirds of bim1- cells lack the bridge, not one half.

- page 15 : what does « at least somewhat » mean ?

- page 20 : Wang et al. (Altschuler paper) is not in the reference list page 35.

- legend Fig6D. « see methods » but information is not in methods. See SI ?

Reviewer #4:

In this study, Pesce et al. performed a screen to identify genes that reduce signal variation in the *Saccharomyces cerevisiae* pheromone pathway. The study builds on experimental methodology they developed in a previous paper (Coleman-Lerner et al, 2005) to measure pathway specific noise.

Here they identify 50 genes that affect signal strength or variation and followed up on a specific set of genes involves in microtubule function. They further showed that proper function of the microtubule is necessary for maintaining low pathway noise in the pheromone pathway.

Although we think the data set is interesting and has potential new insights on the role of microtubules in regulating noise, we have concerns regarding the analysis and the interpretation of the data. The writing is also rather long-winded in the introduction and especially in the discussion; rewriting these sections to be more concise would help improve the clarity and focus the message.

Major comments:

1) the study claims to identify genes that alter pathway variation, however many of the deletions also affect pathway output (mean expression) (Figure 4D). The main concern is that in most cases the change in noise could be simply explained by the change in mean expression, as shown before both theoretically and experimentally (Bar-Even et al., 2006; Newman et al., 2006; Thattai and van Oudenaarden, 2001). Could the authors calculate how much of the pathway noise alteration can be explained by the change in the mean expression how much it is changed due to "pure" noise modulation?

2) In Figure 4D there do appear to be a handful of candidates under 20nM pheromone that change primarily in the noise (highest blue circles; maybe 8-15 candidates depending on where one places the statistical cutoff). Were these the ones that were focused on? Which deletions are these? Do they show increased variation under 0.6nM as well or only 20nM? This should be made clear with a new panel (of variation versus output) focusing on these candidates and labeling them.

3) Throughout the paper, in Figures 3, 6A-B, 7 and 8, noise is plotted as a function of Pheromone dose. Again, since the dose is affecting also the mean expression, and mean expression for each dose is different for each mutant, it is hard to determine from these graphs if noise is indeed modulated. To be specific, if we look a figure 3A-B, the right hand column, we can see the mean expression of the mutant is lower than the wild type in the high pheromones regime. This regime also correspond to higher noise in the mutant. To what degree the change in noise is affected by the alteration in the mean?

4) Could the authors simply add a plot of pathway noise as a function of system output for different doses and mutants? If noise indeed controlled by these genes we expect to see that the noise vs. mean curve of the mutants will be higher than the Wild Type curve.

Specific comments:

- In order to understand the experimental setting we had to go back to the authors' 2005 paper. It would be beneficial to add a diagram in Figure 1 explaining which promoter is driving which reporter. It is stated only in the caption of Figure 2A that act1 promoter is driving YFP.

- On a related note, it would greatly help focus the paper if Figure 2 were absorbed into Figure 1 so that the screen could be presented earlier.

- The necessity of figure 3 being presented in the main text as full stand alone figure was not entirely clear. This seems appropriate for the SI.

- Other studies showed that chromatin regulators also control expression noise in *Saccharomyces cerevisiae* (<http://dx.doi.org/10.1016/j.molcel.2012.05.008>). Could the authors report the effect of

these genes on the noise properties that they measure in their study? It may be interesting to compare/contrast against the other recent screen cited (Dar et al. 2014), which incidentally find that microtubule inhibitors affect expression noise.

- We could not find where Figures 2A-B were referenced in the main text.
- The introduction and discussion sections could be made more concise and to the point

MSB-16-7390 & MSB-16-7391

Authors' point-by-point response (MSB-15-6386)

15 October 2016

Responses to editorial and reviewer comments on revised Pesce et al. papers.

We appreciate the very substantial effort the reviewers put into reviewing the first manuscript. We hope that they might view favorably the person-years of effort we have put into addressing their concerns in these revisions.

Comments refer to the previously submitted manuscript, and our description of our response will refer to actions that now affect both papers here, P1, "screen", and P2, "mechanism and consequences".

Overall summary of our response.

We addressed all of the editor's comments and all the comments of the four reviewers. By doing so, we believe that the paper improved greatly. We were constrained by our inability to perform additional wet experiments, but we were not constrained in our ability to perform deeper analysis and new analysis on the trove of data from the genetic screen and our followup experiments, nor in our ability to hunt up and include additional experimental data. We're very proud of the depth that our conclusions gained from this additional work and hope that much of what we've done will define a new standard.

We note that we labor under the same constraint as the previous work, that there is a well, the performance of additional wet experiments, to which we cannot go back. We can think about the great quantities of data we already possess, but we cannot generate new data. So, again, I am also requesting that reviewers make a publication determination whether the experiments and data we present here justify the results and conclusions.

Remarks from editor, Thomas Lemberger

- one major limitation of the current study is the absence of in-depth mechanistic follow up of the microtubule-dependent effects. Coordination with your follow up study on these mechanistic aspects would thus be very helpful in this regard.

Absolutely. Coordinated by means of finishing the follow up manuscript, P2, which is enclosed.

- the data should be presented in full; in particular the results of the microscopic measurements of the 50 mutants analyzed should be presented. These data should also be made available as well as the FACS data.

We now describe the analysis in Supplementary Information for P1 and show the data in P1 Table S6.

- given potential defects in chromosomal segregation, the bim1 and gim4 mutants may harbor additional genetic defects that could confound the analysis. The possibility of such an artefact should be excluded.

General points about additional genetic defects in the mutant strains that could confound the analysis.

Actually, in the initial mutant screen, and in the initial submitted manuscript, we had gone to great lengths to avoid this possibility. Recall that we begin with an instance of the early haploid deletion collection, then mate haploid cells from populations carrying each deletion with our particular mating partner that carries all our reporters. This gives us diploid strains. Which we then sporulate, to create haploid strains, from which we then select clonal haploids that bear the needed markers. So we wash out ploidy alterations like extra chromosomes, gross chromosomal linkages,

etc. defect by passing the deletions through a diploid strain and meiosis and requiring that the spores germinate. Then, we are careful to select multiple, small colony clonal isolates, which we use in the screen. To our knowledge, neither of these steps to ensure genotype: a) passage through a round of mating/ meiosis/ sporulation/ germination of viable spores into colonies, and b) then use of clonal cultures resulting from small, viable haploid spores, has been done before in study of this type. We hope it might define a kind of "best practice".

Then (in the previous paper, and now, in P1) we remake deletions that we identified in the screen and choose to study here in a clean genetic background, and verify those by PCR. And we verify the mutant phenotypes in these clean strains.

The phenotypes we observe are reproducible and are due to the gene deletions we found in the screen.

- some of the claims have to be softened, for example about independent effects on noise and means response. A better visual presentation of the data would also be helpful. In addition, supplying source data files that provide the individual numerical values displayed in figure 4D would be helpful as well.

Softening of claims about independent effects on noise and mean response.

I'd like to say first that we thought hard about this issue, how to think about the effects of mutations on mean and noise and what a meaningful mutant would mean in this regard, in the process of the first submission. So we considered use system output or even total transmitted signal, P, instead of dose for our workhorse x axis in the first manuscript. We decided against that for reasons no longer relevant here.

We read your request and the request by three reviewers to correct for effects on mean expression as a pretty clear indication of what other people think appropriate. So we acted on it, going through the work to re-analyze and re-plot all data, and reexamine all conclusions, in this light.

To say that again, we addressed this concern directly by reanalysis of the data supporting every one of our conclusions in a way that corrected for differences in the mean (by graphing noise in transmitted signal on the y against system output O on the x, in one case, total transmitted signal P on the x), then by performing appropriate statistical analyses to show that any difference from reference on which we based any qualitative assertion was significant.

As a result of this process, some of our previous specific conclusions and results changed. However, now, for all of the results we now show in the two revised papers, by definition, all of effects we see on noise are real even when corrected for changes in mean expression, and they are in all cases backed up by appropriate statistical work.

The changed text describes this new work. In P1, it is reflected in changed Figure 4d and e (was 4d) and Figure 6. Moreover, in P1, to directly address the question for the two key mutants, $\Delta bim1$ and $\Delta gim4$ (made, as above, in clean genetic backgrounds) we added a new Figure, Figure 7, to show specifically that these genetic changes affect cell-to-cell variation in transmitted signal independently of any effect on mean expression.

We also note that much of the companion paper, P2, analyzes the effects of additional targeted perturbations on microtubule function and graphs them in the same way. In P2, Figures 4, 5, 6 and 7 reflect this new approach.

To repeat, we have addressed this issue by consistently adjusting for effects on mean output in all of our analyses and in all of our plotted data.

Better visual presentation of data.

We took this to heart. We tried for a clean design of amended figures and new figures. We carried the same set of visual/ design metaphors over to the companion paper, P2.

Source data files for the data in 4d.

We include this as new Table S2. We also made a new Figure, Figure 6, to show more about the behavior of these mutants in a rescreen.

- the reviewers all felt that the text need a profound rework to eliminate jargon and considerably simplify the narrative to achieve a clear and concise presentation. This concerns the Introduction and the Discussion sections but also to some extent the Results section. Redundancies between these sections should be avoided and methodological details should be provided in the Materials and Methods section.

We took this to heart and subjected the text to what we can honestly describe as a "profound rework". As important or more important, was hiving off some of the work in the previous version into a separate mechanistic paper, P2.

Remarks from Reviewer # 1.

This study explores the genetic factors that influence the degree to which individual cells in a clonal population show variation in their response to an external stimulus. For this, the authors use the yeast pheromone response system (PRS), a thoroughly studied signaling system with numerous advantages relevant to the issues under consideration. The authors screen a large number (roughly 1000) of viable deletion mutants, and identify a subset (roughly 50) that affect total response and/or cell-to-cell variability, which then partition into various subgroups with distinct effects on specific phenotypic variables. Most of the remaining effort is focused on the subset of genes that alter microtubule functions. The observations suggest that genetic perturbations can have distinct effects on the strength versus variability of signaling response, and that not all microtubule-related functions alter these parameters in the same way. By the end, it is evident that signaling strength and variability can be altered by mutations in several microtubule-affiliated genes, although the causal mechanisms are not probed further.

Clearly, the authors have done a lot of work, and I have no reason to doubt the technical aspects of the experiments or the mathematical analyses. But I did have trouble determining exactly what has been learned and what new insights were revealed. This difficulty stems mainly from the fact that, while the phenotypes from microtubule-associated genes are intriguing, the possible mechanistic causes were not probed and hence remain highly speculative. In addition, there are issues with the presentation: (a) The manuscript contains such an overwhelming deluge of information, some of which seems largely dispensable, that it ends up obscuring rather than clarifying the overall goals and findings. (b) The text suffers from being bloated, jargon-filled, and rather grandiose in places (especially the Discussion); it could be substantially improved by trimming the excesses and consolidating the central points into a more coherent narrative. Collectively, these features make it difficult to offer a strong positive recommendation.

What has been learned/ insights revealed/ mechanistic causes not probed.

For what it is worth, we find this observation more than fair. We have three responses to it.

First, the first paper, P1 describes making a library, setting up a high throughput screen of it, and then from it. The library is cleanly made and a useful resource, the screen is technically well done, and the things we found from it were mutants with interesting behaviors that affect different "aspects" (or even "axes") of variation in signal and response.

We don't view this kind of work as qualitatively different from any other genetic screen that sets up to look for mutants, looks for them, classifies those in some way, and then points to some particular mutants as potentially interesting for further study. This is for example what Wieschaus and N-V did for *Drosophila* maternal effect lethal genes that identifies some genes that screw up anterior-posterior development and others that screw up dorsal-ventral development.

In the revised manuscript, P1, due to comments by Reviewr 1 and the others, we've taken care to show that some effects (for example on signal variation) are independent of others (effects

on mean). So, we carried out a mutant hunt and found mutations that affect an aspect of system behavior that we consider interesting. There is no dishonor in this, and in fact, much of what we believe to be true in biology has begun with this sort of genetic work. In this view, the only new thing is that our phenotypes are quantitative, and subtle.

Second, for the mechanistic causes not probed, we have addressed this with the companion paper, P2. This describes extensive experimentation to learn causes -- and to show consequences. That paper presents 9 different sorts of mechanistic experiments in all. One set, the ectopic expression/ genetic bypass experiments to show where in the pathway the noise is introduced, represents a transposition or mapping of a classical genetic methods to this new, quantitative key. All through P2, as before for P1, we have tried to make the experimentation and analysis maximally technically well done.

Moreover, in the work described in P2, we learned a good deal about the cell biology that underlies the increase in variation and how the variation arises. We can now point to the fact that for some alterations that affect cytoplasmic plus end microtubule function, the variation depends on Fus3 acting at the signaling site, and that additional experiments point to destabilizing the formation and lifetime of the signaling site defined by membrane-localized Ste5.

From this work, we also learned about, and discuss limits. Limits to what we learned from these studies and the possible limitations of any of the numerous experimental modalities we deployed to offer complete mechanistic insight of this type into his and other questions about system function. In this case, one possible systems-biological goal, understanding and predicting the quantitative outcomes of processes involving about two dozen different molecular species and a large number of autostimulatory and cross-stimulatory interactions, remains out of reach for now.

Finally, we are publishing the full data from this screen. We view our libraries and data sets as troves, which we hope will be mined by investigators to find mutants that affect other processes in which those

Specific points on the manuscript:

Hereafter, when we violently agree with the Reviewer's point, we will simply begin by saying "yes"

1. In the long Introduction, it is frequently unclear why all of these numerous prior points are being described in such great detail. I suspect that much of it could be deleted. If prior observations are important for this paper, their relation to the present study should be clearly stated at the outset, in order to provide context and clarify their relevance.

Yes. Descriptions shortened, extraneous materials moved, etc. Some of the material wound up in a box, but most of what we suspect the reviewer was objecting to was moved to the companion paper.

2. The Discussion suffers in three ways: (a) it lacks a clear and concrete summary of the key findings and whatever resulting new insights or conceptual advances they might offer; (b) it contains unnecessary repetition of methodological points that are already covered in the Intro and Results; and (c) it is filled with long musings on a range of issues that relate to phenotypic variation but do not really seem to be directly advanced by the findings in this manuscript.

Yes. New four and one half paragraph discussion tries to address these points a, b, and c. Splitting the papers into two helped here as well.

3. The Results section at times includes excessive methodological detail, often redundant with Methods and/or Figure Legends, instead of a clear description of the purpose of the experiment and the information obtained. For example, the GIM4/BIM1 (not "BIM4") section on page 9-10 includes discussion of the choice of strain, the reporters, the pheromone concentrations, time intervals, etc. But the take-home message remains rather nebulous: "despite their similarity in function and phenotype, BIM1 and GIM4 have non-redundant role [sic] in controlling signal variation" (Is that all that has been learned? What does it mean? How has our understanding been advanced?).

Yes. Most of what we have to say in P1 about different roles now lies in the companion paper, P2, leaving P1 to describe the genetic screen. In the revised P1, we are down to one such, about relation between the phenotypes conferred by $\Delta bim1$ and $\Delta gim4$ "This is thus an epistatic interaction (as defined by Fisher (1918)) suggesting that two gene products might have independent effects on signal variation." This is an appropriate observation to make in a paper describing a genetic screen, and, speaking as sometimes-geneticists, we note (with Fischer) that when one observes this effect, suggesting that two genes might be have independent effects, it's not even a trivial observation.

4. Figure 3 seems dispensable. It is mentioned only extremely briefly; once as a half-sentence (page 6, top), and once largely as an aside (page 9, bottom half). Currently, readers can only guess what the Figure represents and what is its point. If the authors think it provides some value, then this should be described; otherwise, its purpose is unclear.

Yes. Banished from Maintext.

5. Similar to point #4, Figures 2A and 2B are never mentioned, and Figure 4 is mentioned in only one sentence (page 6, middle). If they contain important information, then their contributions should be explained.

Yes and yes. Banished and deleted yes.

6. Page 10, middle: The description of results in Figure 6D does not match the plots shown. The authors state that the distributions are "significantly more long-tailed" and then continue to refer to "long tails", a "heavy tail" and "outlier cells comprising the tail". But there are no "tails" evident in these plots, and so this jargon is based on an extrapolation of the results (e.g., to a distribution curve) rather than what is actually evident in the plot shown. And the meaning of "heavy tail" is unclear.

Yes. The use of the term "tail" in this context proved unintelligible to an overwhelming majority of text readers. We changed the wording.

7. Page 11, second paragraph: (a) the reference to "GPY4000 reference cells" is not only an unnecessary detail that could be left to Methods or Legends, but it seems to conflict with the Figure 7A legend (which claims to use "SGA103" cells).

Yes. Changed, eliminated the detail, and erroneous reference to SGA103 lost. Text now mentions GPY4000 once, as our "clean genetic background strain", and Figure 7 legend describes it and its derivatives.

(b) "This result...was consistent with GIM4 controlling variation..." . This seems incorrect, as the preceding sentence refers to signal strength, and so that result cannot be said to agree with results on signal variation.

Yes. That was badly worded in any case, and should have said "consistent with other results..." this now appears (in intelligible form in the companion paper, P2

(c) The choice to emphasize a seemingly minor point (about slight variation in effects of microtubule poisons) is confusing; e.g., is this level of experimental variability really notably greater than that seen in the 3 repeats of the $bim1\Delta$ strain?

Yes. This set of experiments is now in the companion paper, P2, and we no longer emphasize that point.

8. Page 15, top paragraph: The last sentence ("Furthermore, our interest...") does not make sense; it is unclear what "showing that" is meant to refer to.

Yes, deleted from here. Whole introduction of and discussion of possible models for perturbation of microtubule function increasing noise moved to P2.

9. Page 19, top paragraph: The last sentence ("Thus, previous studies...") does not make sense. Specifically: "genes that reduce variation due to differences in genotype" is not coherent.

Yes. This now figures in our discussion of canalization in the companion paper. We hope that that discussion is now intelligible. In point of fact there are genes, for which specific allelic forms decrease the effects on quantitative phenotypes caused by genetic differences elsewhere in the genome. Population geneticists like these genes. We discuss these issues, hopefully intelligibly, in P2.

10. Page 5 bottom paragraph: "a third fluorescent protein" should explicitly mention YFP; otherwise, the later reference to measuring "mRFP and YFP signal" will not make sense to readers.

Yes. No longer part of text.

11. Page 26: The Figure 2D legend is incomplete; it does not mention the pheromone output measurements that are plotted, and it does not describe the difference between the 3 curves plotted.

Yes. Figure 2D no longer part of paper.

12. Page 26 and Page 28: In the legends to Figures 3 and 7, the authors seem to be conflating significance levels (i.e., p-values) with confidence intervals in an unusual way. I am not sure there is a strict meaning to the phrases "0.95 confidence statistical significance of differences" and "significant at 0.95". These seem like imprecise shorthand.

Yes. The old Figures 3 and 7 are now gone from this and the companion paper. We note that for each qualitative assertion we have performed sometimes high order and we believe always appropriate statistical analysis. It's important that the words we use to describe any analysis be understood by careful readers.

13. Page 28 and Figure 6C: (a) The Figure legend should explicitly describe the distinction between the three plots shown. (E.g., "colors are as in panel A".) (b) The legend should also describe what the dots denote. Presumably, these are individual cells in a clonal population, but if so then this should be stated clearly and explicitly, as otherwise it is ambiguous what is being represented in these plots. (c) Finally, the grey dots (wild type) are difficult to distinguish from the colored dots; perhaps black would be better.

Yes. Terrible plots and graphs no longer in this paper or a companion paper. We went through this carefully to make sure that the whatever iconography a set of figures used was well described in the figure legend.

Remarks from Reviewer #2:

In this work, Pesce and colleagues perform an impressive high-throughput screen to find genes related to cell-to-cell variability of pheromone response in yeast. In more than 1000 single-gene deletion strains, they measure pathway variation separately from other sources of noise, notably cell-to-cell variation in total gene expression capacity. They accomplish this by including the same pheromone responsive promoter driving the expression of two different fluorescent reporters and a constitutively active promoter driving the expression of another fluorescent protein in each deletion strain. The genetic screen itself is carefully executed, with careful consideration given to cell density effects (deletion of BAR1) and the important role of cell cycle on pheromone pathway (analog sensitive Cdk). It is apparent that this screen will be an important resource for the community working on cell-to-cell variation in general and the pheromone pathway in particular.

While the genetic screen itself was carefully executed, I have some major reservations for some of the following analysis and the results that the authors find. Specifically, the two main claims of the paper, i.e., "signal variability and strength [as] independent axes of system behavior" and "microtubule-dependent mechanism that limits signal variation," are not that strongly supported by the authors' data. However, rephrasing conclusions more carefully and some reanalysis of the data can rectify these problems.

We thank the reviewer for a fair summary.

About the first assertion, as above for Reviewer 1, one of the most important sets of work we performed over the past year was a wholesale reanalysis of the data to see under what circumstances mutants exert an effect on variation independently from their effects on signal strength. This involved re-thinking all our qualitative conclusions as well. So we believe we addressed this directly

As above, one major change from the previous work was our inclusion of all the experiments to find out about mechanism in a separate paper, P2. We conducted 9 different sets of experiments, using different experimental approaches, to find out the mechanistic conclusions we found out. And, we qualified the language stating our conclusions appropriately-- we no longer claim to have found a microtubule dependent mechanism.

Major points

a) As the authors also show in Figure 2D, the output of the pathway is inversely correlated with pathway variation. Thus, the majority of what the authors select in Figure 4 as the 'outlier' hits is not necessarily genes that affect pathway variation independent of pathway output. The claim that signal variability and strength are "independent axes" of system behavior is only supported by a few hits that lie outside of the inverse relationship cloud in Figure 4D. Instead of choosing many genes based on 1D metrics as shown in Figures 4A-C (most of which are simply genes that affect system output and affect variation through its effect on system output), the more relevant choice would be to select only those deletions that move a cell in 'y' axis in Figure 4D without a similar move in the 'x' axis. These would be the deletions that change pathway variation independently (at least relatively) of pathway output.

Yes. We can't get round the fact that, in our screen, we identified "interesting" mutants by the metrics that we used. But in this revised work, we have addressed this issue in general by universally moving to moving to plot variation in transmitted signal versus system output as the x. So the new 4D and 4E plots show contour lines showing which mutants exert effects on variation independent of effects on output. In other words, one can see a swarm of replicates of WT (purple points) in each panel, and compare to these-- a mutation that increased relative variation merely by attenuating the output would be expected to land on roughly the same contour line as these WT points. On the other hand, a mutation that increased relative variation by disrupting the signal path would be expected to land on a higher contour, even if signal strength was reduced somewhat. And many mutations do just that!

To help guide future work based on this screen, the authors could report on Figure 4D which dot each chosen gene corresponds to.

We tried for awhile to make each dot interactive, but decided to settle for reporting all the data in a new supplementary table S2.

They should also rephrase their conclusions on "system variability and strength as independent axes," because most deletions do not support this independence (and they do not show the source of this independence even for the hits that independently affect output and variation). With the given data, it would be acceptable to say that there exist genes that seem to affect pathway variation and output independently. However, it does not follow that pathway variation and output are independent axes.

We believe our re-analysis and new plots allow all to see which genes affect variation independent of their effects on system output. Many genes do affect both, but some genes that affect one thing but not the other, we are using the term axis, which perhaps is never well defined, deliberately, to make an analogy with ambitious genetic screens of the past. Many *Drosophila* maternal effect lethal mutations screw up development of the embryo, and many screw up both anterior-posterior and dorsal-ventral morphology, but some quite predominantly affect one but not the other.

b) Similarly, the level of the presented data does not match the authors' strong conclusions on

"microtubule-dependent mechanism that limits signal variation." The authors do not show a "mechanism" in this paper; they find that mutations that disrupt microtubules, which, in turn, possibly disrupt many different processes, and somehow affect pheromone signal variation (in some cases, somewhat independently of pathway output). The use of the word "mechanism" is warranted only if the authors find the mechanism of this effect on signal variation, on which they speculate on Discussion. The sections that refer to "the mechanism" should be rewritten to reflect this point.

Yes. Quite so. This analysis now becomes part of the companion paper, P2, along with a very large number of additional experiments. We discuss what we have learned about the processes that operate normally, about the nature of the processes disrupted in mutants and by other perturbations to microtubule function, and we discuss the limits to such an analysis.

c) It is known that deletions can often pick up additional mutations and aneuploidy. In one study, 8% of strains in a deletion library have been found to have a chromosomal abnormality (Hughes et al, 2000). This is particularly important for *bim1*Δ, which is known to have a defect in chromosomal segregation (Schwartz et al., 1997). Therefore, the authors should do a whole genome sequencing of their *bim1* and *gim4* mutants to make sure that their results on microtubules do not stem from such an artefact.

In our initial paper, we went to great lengths to minimize the possibility of additional mutations and aneuploidy in the new strains we made. Remember that the strains in our haploid reporter-strain deletion library come to us (ie, we constructed them) from sporulated members of our diploid library, only one of whose haploid parents represented the original, buggy haploid deletion collection. During the process of mating to make a diploid, undergoing meiosis, sporulating, and germinating only live spores, original sins of aneuploidy were washed away. That is, the strains we used in our primary screen were born euploid).

We also selected and treated our haploid library strains in ways (including use of multiple clones) that guarded against second site suppressor mutants making it through the mating-meiosis-sporulation-germination drill. In this revised paper, we've reviewed what we believe to be most of the relevant literature on second site suppressor mutations and aneuploidy, and even ran down some of the investigators most involved in this work and carefully considered some of their unpublished data on their original collection.

Finally, we rewrote the paper (this was probably lost in the previous text) to make clear that our assertions about the phenotypes caused by *Δbim1* and *Δgim4* hold in freshly constructed strains in a different, and clean, genetic background.

This result, showing that the increased variation is a consequence of deletion of these genes, is where our manuscript on the genetic screen now ends.

Then in P2, companion paper, we go on to carry out experiments using number of cleanly targeted genetic mutations and perturbations that affect different aspects of microtubule function and show that these also increase variation.

Given the numerous ways we confirmed subsequently in different strains that microtubule mutations increase variation, and match the phenotypes we observed in the screen, we are comfortable that the phenotypes we have been studying are a consequence of the genetic alterations we describe.

The revised text in both manuscripts now mentions explicitly each brick in the edifice supporting the finding that alterations in microtubule function cause these variation phenotypes.

d) In Figures 3, 6, 7, the more important relation is pathway output vs. pathway variation in different mutants, rather than their separate effects at different pheromone concentrations. The authors should plot pathway output vs. variation for these different strains, and show how significant this difference is in this plot for the wild-type vs. the mutants (e.g., ks-test).

Yes. Addressed by changes in x axis throughout and tests of significance throughout.

Minor points

a) The introduction contains some unsubstantiated, vague, or incorrect statements. For instance, the authors write "the importance of rapid-changing stochastic sources of variation is assumed because it is consistent with mathematical models." This sentence insinuates that the importance of such noise is incorrectly applied simply because of mathematical convenience without any citations to support this statement.

We cleaned up that exposition and put it into a box. We got rid of the offending assertion. There are points to be made about differences in understand the different sources of cell-to-cell variation perhaps best discussed over a beer with the reviewers, but we do not wish to court controversy by doing so here.

We note that we do need the information in the box. We needed to take it out of the introduction to make the introduction survivable, but we need to have it in the paper in order to explain our own mutant screen. That is, we need to explain differences in P and G in order to set up a screen and a paper that will describe isolation of mutants that have increased cell-to-cell variation in P.

Finally, the statement that extrinsic noise was attributed to differences in lac repressor in Elowitz et al. (2002) is not accurate (LacI is a contributor to extrinsic noise, and both intrinsic and extrinsic noise increase upon changing LacI concentration).

That particular offending assertion was based on close reading of Elowitz et al. 2002, and we can defend it, but we are aware that this is not one of the conclusions from this work that is commonly remembered. As above, rather than court a controversy we believe unrelated to the material we need to present, we softened the language so that an example cause of extrinsic noise was cell-to-cell fluctuations in the abundance of molecules such as regulatory proteins and polymerases.

b) Because the disruption of microtubules may affect cell morphology, the authors should show pictures of their various microtubule mutant strains (so that the readers can judge whether different mutants have similar morphologies with or without alpha factor) .

There aren't visible differences in morphology in these mutants. In response to this request and a similar request from Reviewer 3 have now added a section to the supplementary information for P2 (Section 5) that discusses this issue.

c) The authors should better motivate their follow-up on microtubule mutants. Why did this group of deletions seem more interesting than the other groups?

In the revised paper, P1, we supplied as much honest motivation as we could. The awkward truth is that we were interested in a possible effect of microtubules due to our interest in a model that they actively position the nucleus within a gradient of signaling protein. Among the reasons this model was attractive to us was our belief at the time (based on Yu et al. Nature, 2008) that downstream response was aligned with receptor occupancy due to a long range negative feedback. During the intervening years, we have come to believe that the PRS and other signaling systems do not use closed loop feedback control of this type, but another, inferior mechanism (Andrews et al. Cell Systems, *in press*). More important, in experiments that comprise a large portion of the companion paper P2, we now argue that our mechanical model by which pushing and pulling by cytoplasmic microtubules on the nucleus to maintain its position within a gradient is not likely to be true. In the revised P1 here, we state the in bare facts describing the reason for our interest, and we introduce/ motivate, test, and discuss the above model at greater length in P2.

Remarks from Reviewer #3:

The study of Pesce et al. is a genetic screen for genes that modify noise in the yeast pheromone response pathway. The method is based on a two-color system, where one fluorescent protein reports pathway response and a second one reports constitutive gene expression capacity. Using high-throughput strain construction, culturing and flow cytometry, the authors identified 50 gene deletions showing evidence of noise alteration. They followed-up with time-lapse microscopy on a number of these genes (they say 42 but the results are not presented, see below), and precisely

characterized the implication of two genes : BIM1 and GIM4, which regulate microtubule dynamics and folding. The results show that perturbations of microtubules (in the gene deletion strain, or by chemical treatment, as well as by overexpressing a dominant negative genetic perturbator) generates variability of the pathway response.

The experimental work is very abundant and the conclusion is important : linking microtubule functions to molecular pathway transduction variability opens questions relevant to all eukaryotes.

Major critics :

1. The authors used flow cytometry as the primary screening steps and identified 50 genes which deletion modified pathway variability. This list is based on cutoff thresholds. The text says that 42 were then studied by microscopy but the results obtained on these 42 strains are not provided. The Supplementary Information details only the experimental method and the gene names. How many of the 42 were validated and on what basis ? All results from the microscopy acquisitions and analysis should be presented, not just the trajectories of *bim1* and *gim4* strains.

We now present all the microscopy results. We describe the secondary screens and their results in detail.

2. The raw flow cytometry data should be deposited in a public repository, such as <http://flowrepository.org/>. This will ensure reproducibility of the analysis results, especially since the methods do not detail data preprocessing such as gating, compensation or background correction (if any).

We are in the process of depositing the data at flowrepository.org. We included the raw flow cytometry data from the screen in this paper itself as Table S2, where MSB will archive it. We've also gone over the methods description to a level of detail we think people will be able to reproduce it.

3. There are conceptually two very distinct ways to generate pathway response diversity when perturbing microtubules. i) The genetic perturbation could generate abnormal cellular internal organization (e.g. failure to polarize the nucleus) in many cells and these cells could then respond more or less efficiently to the pheromone signal because of this mis-organization of their content. This is variability prior to the stimulation and differs from ii) The genetic perturbation could impact the precision with which the signal is transmitted. For example, if the active Fus3p gradient described by Knop is critical for precise signalling and if this gradient cannot be well established. This variability is posterior to the stimulation. The authors could examine the cell morphology of *bim1*, *gim4* and related mutants from the data of Ohya et al. PNAS 2005 (<http://yeast.gi.k.u-tokyo.ac.jp>). If variability in nucleus position or actin distribution is reported in these mutants in the absence of pheromone, this would argue for i).

The reviewer proposes another way to conceptualize the universe of possible models that might explain how microtubule perturbations could result in increased variation in transmitted signal. We appreciate this way of framing the problem.

Our response at least sets up the work in P2 paper and now (we hope) might influence how reviewers and readers might read that companion paper.

We need to lay out the work in the companion paper briefly here. In P2, we establish by extensive experiment that particular treatments that affect the function of plus ends of cytoplasmic microtubules lead to erratic signaling by scaffold and membrane localized Fus3 and instability of the Ste5 scaffold at the signaling site. All the experimental evidence is consistent with erratic signaling and leads toward a particular model we propose. So our work is consistent with hypothesis ii).

In response to the specific suggestions of the possibility of visible morphological differences due to overall changes in cytoskeletal organization in [any of our numerous] *Δbim1* and *Δgim4* strains do did not observe these. We now include a paragraph describing our observations and our comparison with our own observations with the Ohya et al, 2005 data as part of the Supplementary Information for P2.

4. The possible mechanisms are not fully discussed :

- The main implication of microtubules organization in response to pheromone is probably to get ready for karyogamy. The authors may have found a piece of evidence of a feedback control of proper MT architecture on the pheromone signalling pathway. Cells would benefit from such a feedback, so that MTs have time to be properly organized before cell fusion.

If we understand this point, we would say that work in P2 points to an additional role for normal cytoplasmic microtubule function in cells exposed to pheromone, which is to ensure (or not cause instability in) the smooth operation of the signaling apparatus. And we understand the emphasis that some previous investigators have placed on the function of cytoplasmic microtubules in nuclear congression. And for what it is worth we agree that the main function of cytoplasmic microtubules might well be to get the cell ready for karyogamy. We certainly have no sense about which function we should call most important.

The reviewer proposes a particular (and, to us, interesting and plausible) idea that alterations in cytoplasmic microtubule function might create a feedback that might delay cell fusion until the microtubules are properly organized. Again, we find this to be an interesting idea, and it's not inconsistent with our newly presented results in P2 Figure 8. We don't, however, know that our results bring enough new insight, or the authors enough deep knowledge to this topic, to help us evaluate or even speculate about it very well.

- In wild-type cells, the nucleus moves to the tip via oscillations whereas in bim1- cells it most likely moves continuously in one direction because pushing forces are abolished. Oscillations can homogenize molecular concentrations in viscous media (cytosol). Could it be that pulling-only causes situations where local concentrations are extremes (bad diffusion and spatial heterogeneity of concentrations in the cytosol)?

Aha. For what it is worth, we like this idea (in P2 we set out to test another mechanical model with which the above idea would be consistent). In P2, the companion paper, Figure 3, we directly examine the dynamic relationship between the nucleus and the shmoo tip. Actually, the nucleus in *Δbim1* cells is not being constantly pulled closer to the shmoo tip than in reference cells (moreover, and consistent with previous published work, we point out that in many of *Δbim1* cells, the microtubule bridge is unattached). So our data does not support the Bim1-dependent cytoplasmic mixing idea. But, we in fact have an unpublished result, from the same experiment now shown in Figure 3C. In *kar3-1* rigor mutants, in which the nucleus does not move around, variation is increased. We changed the Figure legend to reflect this result.

More broadly, however, P2 shows why we were forced to discard our mechanical model, and neither it nor the above idea is easy for us to integrate into our findings that increased η^2P requires active Fus3 and stable Ste5 residence at the membrane.

- Could the nuclear export of Fus2p to the shmoo tip (Paterson et al. 2008 JCB) be implicated? Paterson et al. showed that Fus2p movements take place under nocodazole treatment, but this does not rule out a change of fine properties such as cell-cell variability. Is it possible that MT dysfunctions perturb actin filaments architecture and, in turn, Fus2p localization at the tip could be altered or delayed in some cells. Would that affect their dynamics/strength of pathway activation?

Yes, very much so. This idea is wholly consistent with what we think is going on. The companion paper P2 implicates a whole set of proteins being recruited to and transported to the signaling site/ polarity patch and a devil's cat's cradle (or rat's nest) of auto- and cross- stimulatory interactions to amplify instabilities. We discuss this extensively in P2. For here, we agree with the reviewer; nuclear export of Fus2 to the shmoo tip (as well as lots of other things dependent on microtubule transport, actin transport, and simple recruitment by diffusion) seems to be altered, delayed, made unstable, and otherwise screwed up. The legend for Figure 1e in P2 covers this transport, and we changed it to add the reference to Fus2 transport.

Less major points :

- First half of second paragraph page 10. The description of the distributions of the spread of trajectories is hard to follow : tails of distributions are mentioned and readers need to imagine the distributions. Better to explain clearly PSD and MPS directly, or better phrase what these tails are.

Yes. See response to same issue raised by another reviewer. This is now described in P2. Our use of the word "tails" in the way we did was not an open invitation to confuse the most careful of readers. We have gotten rid of it and should never have used it in the first place. We defined PSD and MPS directly.

- Interpretation of symmetrical (bim1) versus below-only (gim4) crookedness. The half-life of the fluorescent reporters used should be mentioned : if fluo proteins are very stable (especially for PRM1 activity), only acceleration of production can reliably be detected, not deceleration.

We've thought about this, and, if we understand it correctly, we are not sure we agree with the point. That is, we believe we can detect shortfalls in new reporter synthesis as reliably as increases.

Consider. We are measuring total signal from each cell. We understand that everything we measure is smeared out in time by maturation delay. If production has been constant for a long while and then abruptly stops, then the rate of increase of total fluorescence will drop to zero exponentially, with a time constant equal to the maturation delay. That's pretty much what we did in the first of the two ways we measured those numbers, in Gordon et al. 2007. Maturation delay also inhibits to some extent our sensitivity to increases in expression rate, which we see as an exponential approach to the new rate of increase.

But for degradation, we know, again from Gordon et al. 2007 that (at least for CFP and YFP) the half lives of the mature fluorophores are longer 3 hours, thus we can ignore them here. So, at a given timepoint, we are in essence seeing all the XFP that a cell ever produced, recording those numbers, and using them to extract smeared rates of production etc. A deceleration in the rate of expression, over time, will result in a measured signal that falls below the signal expected for a constant rate of expression.

- Fig6D : ACT1 is indicated in the y-axis and legend whereas text page 9 bottom says that BMH2 promoter was used. Which one was used ? Also, crookedness can be caused by irregularities in constitutive promoter signal (denominator). How do the authors rule this out for the trajectories presented on the figure ?

Promoter used: P_{ACT1} . This error fixed, with thanks for the reviewer for catching it. Figure now becomes P2 Figure 2.

Crookedness. Glad the reviewer asked. In fact, we had spent quite a while on this very question before we sent off the first version of the paper. Here's where that stands. Variation in constitutive expression (the denominator) does contribute and its contribution is different from one strain to the next. We know this well, because, to address it, we did a multiple linear regression model of the ratio crookedness. The inputs were the numerator and denominator crookedness (we called these cn and cd), and the predictors were cn , cd , $cn*cd$, and a constant. In the analysis we performed, a linear combination of the predictors gave a prediction of the ratio crookedness, and the regression adjusted the combination so that the model fits as well as it can. For the mutants $\Delta bim1$ and $\Delta gim4$, the numerator always contributed more than the denominator (i.e. the pheromone inducible promoter more than the constitutive).

We changed the legend to P2 Figure 2 to make this point.

- The review of Molk and Bloom J. of Cell Science 2006 (119) is very helpful to imagine microtubules arrangements in the shmoo. It should be cited to guide non-specialist readers.

Thanks. Lovely and helpful review cited, along with the near contemporaneous original paper by Molk et al. that lays out cytoplasmic microtubule function in nuclear congression. Thanks for that.

- The title of the study differs between the manuscript file and the SI file. « Signal variation mutants identify microtubule dependent mechanisms required for accurate cell fate decisions » is not appropriate : the cellular decision itself (fusion) was not quantified, and we already knew that MT mechanisms are essential for karyogamy.

With thanks to the reviewer for catching this error. The second paper, P2, does examine the fate decision. In it, we look for the mechanism by which normal cytoplasmic microtubule function allows less erratic signaling, and in which we show that lesions that affect cytoplasmic microtubule function also affect choice of the "correct" mating partner in competitive mating experiments (Jackson and Hartwell, 1990).

Titles for companion paper and SI now correct and synchronized.

«Genetic screen identifies microtubule-dependent mechanism that limits signal variation » does correspond to the study.

With thanks again. Related to the split of content between the P1, the screen paper, and P2, the companion paper. P1 is now all about the screen.

P1 title now corrected so that screen "identifies genes with different effects on transmitted signal and response.

P2 title changed as well.

- Pathway variation (CV2 of mRFP/<mRFP> - YFP/<YFP>) is sometimes notated $n2(P)$ (Figure 5) and $n2(P) + n2(\gamma)$ (Figures 3, 6B ?, 7?, 8 ?). Be more precise and consistent on the quantities reported.

We now explain that we compute from the data a quantity that, strictly speaking, is equal to $n2(P) + n2(\gamma)$. We further explain that we know from past work (Colman-Lerner et al 2005), that $n2(\gamma)$ is an order of magnitude smaller than $n2(P)$, and we will thus for the sake of brevity refer to this computed quantity, in plots and legends, as $n2(P)$.

- page 9 subtitle. BIM4 should be BIM1

Eliminated in the rewrite.

- page 10 bottom : « average » appears twice.

Eliminated as above.

- page 15 : two thirds of bim1- cells lack the bridge, not one half.

We were combining our own unpublished observations with those published. Eliminated. Replaced with the more neutral "fraction"

- page 15 : what does « at least somewhat » mean ?

Eliminated. Banned forever from set of phrases permitted in scientific prose. Emperor Qin has exiled the scholars who suggested this phrase to the north to labor for their lifetimes building the Great Wall.

- page 20 : Wang et al. (Altschuler paper) is not in the reference list page 35.

Rewritten so as no longer to reference.

- legend Fig6D. « see methods » but information is not in methods. See SI ?

These are the measurement methods we worked out and published in a first methods heavy

paper-- Colman-Lerner et al., Nature, 2005 and elaborated on in a second paper devoted wholly to methods (Gordon et al. Nature Methods 2007). We now reference these in the methods section of the P2 and in Figure 2.

Reviewer #4:

In this study, Pesce et al. performed a screen to identify genes that reduce signal variation in the *Saccharomyces cerevisiae* pheromone pathway. The study builds on experimental methodology they developed in a previous paper (Coleman-Lerner et al, 2005) to measure pathway specific noise.

Here they identify 50 genes that affect signal strength or variation and followed up on a specific set of genes involves in microtubule function. They further showed that proper function of the microtubule is necessary for maintaining low pathway noise in the pheromone pathway.

Although we think the data set is interesting and has potential new insights on the role of microtubules in regulating noise, we have concerns regarding the analysis and the interpretation of the data. The writing is also rather long-winded in the introduction and especially in the discussion; rewriting these sections to be more concise would help improve the clarity and focus the message.

We hope the two papers, P1 and P2, now adequately address the reviewer's concerns about the analysis and interpretation of the data. We greatly shorted the introduction and discussion by various stratagems including drastically cleaning up the writing (finishing P2, the companion paper, and dividing the subject matter between the genetic screen and the work on mechanism was instrumental in making progress toward this goal).

Major comments:

1) the study claims to identify genes that alter pathway variation, however many of the deletions also affect pathway output (mean expression) (Figure 4D). The main concern is that in most cases the change in noise could be simply explained by the change in mean expression, as shown before both theoretically and experimentally (Bar-Even et al., 2006; Newman et al., 2006; Thattai and van Oudenaarden, 2001). Could the authors calculate how much of the pathway noise alteration can be explained by the change in the mean expression how much it is changed due to "pure" noise modulation?

Yes. In response to this concern, shared by other reviewers, we re-analyzed all relevant data in both papers and replotted all of it, throughout, to compensate for this effect. We also changed all qualitative assertions so that they were consistent with this means to assess the degree to which mutants affected variation in transmitted signal. In P1, note that Figure 4 (d and e) and Figure 7 even have contour lines to allow the reader to make her/his own determination as to how much a given mutant affects noise *per se*. We analyzed all data in P2 in this same way.

2) In Figure 4D there do appear to be a handful of candidates under 20nM pheromone that change primarily in the noise (highest blue circles; maybe 8-15 candidates depending on where one places the statistical cutoff). Were these the ones that were focused on? Which deletions are these? Do they show increased variation under 0.6nM as well or only 20nM? This should be made clear with a new panel (of variation versus output) focusing on these candidates and labeling them.

Yes, done, as above. Figure 4 (d and e) and Figure 6 now show the deletions we examined, plotted in this way, and we provide data for all the strains in P1 Table S2.

3) Throughout the paper, in Figures 3, 6A-B, 7 and 8, noise is plotted as a function of Pheromone dose. Again, since the dose is affecting also the mean expression, and mean expression for each dose is different for each mutant, it is hard to determine from these graphs if noise is indeed modulated. To be specific, if we look a figure 3A-B, the right hand column, we can see the mean expression of the mutant is lower than the wild type in the high pheromones regime. This regime also correspond to higher noise in the mutant. To what degree the change in noise is affected by the alteration in the mean?

Yes, done, by re-analysing and replotting everything in both papers as above.

4) Could the authors simply add a plot of pathway noise as a function of system output for different doses and mutants? If noise indeed controlled by these genes we expect to see that the noise vs. mean curve of the mutants will be higher than the Wild Type curve.

Yes, done. In this paper, P1, We added a Figure 7 that does exactly this for the two mutants, and we do that for all the various genetic perturbations used in P2.

Specific comments:

- In order to understand the experimental setting we had to go back to the authors' 2005 paper. It would be beneficial to add a diagram in Figure 1 explaining which promoter is driving which reporter. It is stated only in the caption of Figure 2A that act1 promoter is driving YFP.

We appreciate the fact that the reviewer carried out that work. Our big move to make this clearer was to get the description of how the measurements were done out of the introduction. Legend to 1B and 1C say this in words and we added a big Figure 2B that shows the reporters with all the markers. The maintext of P1 now describes the setup in a box, and we'd be glad to include a diagrammatic figure in that box if the editor would let us.

- On a related note, it would greatly help focus the paper if Figure 2 were absorbed into Figure 1 so that the screen could be presented earlier.

We believe (unfortunately) that we need to describe some aspects of the pheromone response system first, in the introduction, or the reader may be lost. That requires, given that the methods section for articles in MSB comes at the end, that our Figure 2, about the screen, can only come as the very next figure. That is the very first part of the results. If the eyes of some quantitative biologists glaze over at seeing that diagram of a signaling system as Figure 1, they have my sympathy.

- The necessity of figure 3 being presented in the main text as full stand alone figure was not entirely clear. This seems appropriate for the SI.

Removed from Maintext. Not important for paper.

- Other studies showed that chromatin regulators also control expression noise in *Saccharomyces cerevisiae* (<http://dx.doi.org/10.1016/j.molcel.2012.05.008>). Could the authors report the effect of these genes on the noise properties that they measure in their study? It may be interesting to compare/contrast against the other recent screen cited (Dar et al. 2014), which incidentally find that microtubule inhibitors affect expression noise.

Yes. Yet another new thing about the study in P1 is that we searched for, and found, non-essential genes that affected not noise in gene expression, but cell-to-cell variation in transmitted signal. We took from this comment the realization that other readers might not appreciate this distinction either, so we made it explicit in the text. The revised paper P1 contains a sentence making this point and citing this other work.

Over the next month, we will also be examining, carefully, the gene list from Weinberger et al. 2012 to look for possible overlap in genes that affect these different kinds of variation. Also, for now only for the reviewers, we note that Dar et al. did find that inhibitors of microtubule function such as docitaxel affected expression noise from the HIV-1 LTR. In our hands in P2, Figure 6, we note that expression of the plus end inhibitor Tub1-828 protein increases η 2P when the system is activated by overexpression of the wholly nuclear protein Ste12. We have always tended to attribute this effect of Tub1-828 expression to the fact that it inhibits the action of cytoplasmic and nuclear microtubules both. If we can think of anything to say about this that adds to the discussion, we may raise this similarity in a revised discussion in P2.

- We could not find where Figures 2A-B were referenced in the main text.

We've removed these.

- The introduction and discussion sections could be made more concise and to the point

As above, I believed we've done so, by splitting the papers, by moving in each paper a class of needed introductory material to boxes, and by tightening up the language. The discussion for the genetic screen is now down to just over four paragraphs. The discussion for the companion paper, P2, is longer, but needs to cover more ground, from canalization genes that affect variation, to positive feedback.

References for cover letter.

Elowitz, M.B., Levine, A.J., Siggia, E.D., and Swain, P.S. (2002). Stochastic gene expression in a single cell. *Science* 297, 1183-1186.

Andrews, S. A, Peria, W, J., Yu, R. C., Colman-Lerner, A. and Brent, R. Push-pull and feedback mechanisms can align signaling system outputs with inputs, *Cell Systems*, in press, 2016.

Colman-Lerner, A., Gordon, A., Serra, E., Chin, T., Resnekov, O., Endy, D., Pesce, C.G., and Brent, R. (2005). Regulated cell-to-cell variation in a cell-fate decision system. *Nature* 437, 699-706.

Dar, R.D., Hosmane, N.N., Arkin, M.R., Siliciano, R.F., and Weinberger, L.S. (2014). Screening for noise in gene expression identifies drug synergies. *Science* 344, 1392-1396.

Gordon, A., Colman-Lerner, A., Chin, T.E., Benjamin, K.R., Yu, R.C., and Brent, R. (2007). Single-cell quantification of molecules and rates using open-source microscope-based cytometry. *Nat Methods* 4, 175-181.

Molk, J. N., and Bloom, K. (2006) Microtubule dynamics in the budding yeast mating pathway. *J. Cell Sci.* 119(Pt 17): 3485-3490.

Ohya, Y., J. Sese, M. Yukawa, F. Sano, Y. Nakatani, T. L. Saito, A. Saka, T. Fukuda, S. Ishihara, S. Oka, G. Suzuki, M. Watanabe, A. Hirata, M. Ohtani, H. Sawai, N. Fraysse, J.-P. Latgé, J. M. François, M. Aebi, S. Tanaka, S. Muramatsu, H. Araki, K. Sonoike, S. Nogami, and S. Morishita, High-dimensional and large-scale phenotyping of yeast mutants, *PNAS* 2005 102 (52) 19015-19020

Weinberger, L., Voichek, Y., Tirosh, I., Hornung, G., Amit, I and Barkai, N. (2012). Expression Noise and Acetylation Profiles Distinguish HDAC Functions. *Molecular Cell*, 47, 193-202

Yu, R.C., Pesce, C.G., Colman-Lerner, A., Lok, L., Pincus, D., Serra, E., Holl, M., Benjamin, K., Gordon, A., and Brent, R. (2008). Negative feedback that improves information transmission in yeast signalling. *Nature* 456, 755-761.

MSB-16-7390 & MSB-16-7391, Editorial Decision

6 February 2017

Thank you again for submitting your work to *Molecular Systems Biology* and I apologize again for the long process. We have now finally heard back from the referees who agreed to evaluate your two manuscripts. Reviewer #1 in paper 1 (MSB-16-7390) and paper 2 (MSB-16-7391) was involved in the previous round (as Reviewer #1 of MSB-15-6386). Reviewer #2 in paper 1 was involved in the previous round as well (as Reviewer #3 of MSB-15-6386). Reviewer #2 in paper 2 was new to this round. As you will see from the reports below, the referees find the topic of your study of potential interest. They remain however globally lukewarm and they still raise significant concerns, which, I am afraid to say, preclude publication of the manuscripts in their present form.

With regard to the new configuration of having 2 separate papers, I am afraid that the reviewers were not supportive. Thus reviewer #1 of paper 1 notes "I don't think splitting the paper in two was an improvement; as suggested previously, it would have been better to do the reverse" and reviewer

#2 of paper 1 recommends us "to ask for a single, streamlined and coherent article, that goes from a screen to mechanistic conclusions, rather than two partial studies". Reviewer #2 of paper 2 feels that "the biological insights are already included in the accompanying paper describing the genetic screen, and thus this manuscript [paper 2] does not provide sufficient insight how microtubule dynamics contributes to signaling fidelity".

In terms of presentation and clarity, I am afraid that the reviewers were also not convinced and all of them raised numerous issues. For example reviewer #1 states "the presentation issues persist, and I can't say that this aspect has been improved; there is still a lot of bloat and redundancy", reviewer #2 feels that "results in P[aper 2] could be presented in a more concise way, and many long sentences in both manuscripts tend to dilute the key messages".

After re-reading the two manuscripts again, we do agree with these evaluations and we feel that the quality of the presentation still needs major improvements and the overall length of the text should be drastically condensed to improve focus and clarity. Based on these clear recommendations, we feel therefore that it would not be possible to publish the two manuscripts as separate standalone papers in Molecular Systems Biology.

Having said that, if we make abstraction of presentation issues and of the splitting of the study into two manuscripts, we agree with reviewer #1 that the collective scientific message of the combined papers remains of interest. Even if we realize that the ultimate mechanistic details remain to be elucidated, as pointed out by reviewer #2 of paper 2, we nevertheless feel that a combined study that presents the global analysis of the large scale screen together with the follow up genetic analysis and cell biology characterization of the bim1 and gem1 mutant would provide sufficient insights into interesting regulatory mechanisms of cell-to-cell signaling variability.

Given that the reviewers did not support publication of paper 2 (MSB-16-7391) as a stand alone paper and given that we had initially asked for a major revision of MSB-15-6386, we would therefore ask you to carefully prepare a revision of paper 1 (MSB-16-7390) by including the key data (see below for details) of paper 2 (MSB-16-7391) such that paper 1 becomes a complete study as requested by the reviewers. I am afraid however that we cannot offer to publish paper 2 (MSB-16-7391) as an independent manuscript in Molecular Systems Biology (for simplicity, I include the same decision letter for MSB-16-7391).

Such a single combined revised manuscript should address the points raised by the current reviewers and should focus on the key aspects of the study so that it delivers a coherent and complete message. We would thus kindly ask that only the following figures should be included in the main paper:

Fig 1 (Paper 1 Fig 1): overview of the PRS, noise def & high throughput design
Fig 2 (Paper 1 Fig 4): distribution of signal variation over >100 strains screened
Fig 3 (Paper 1 Fig 5): Clustering of 50 genes influencing variation
Fig 4 (Paper 1 Fig 7 + Paper 1 Fig 2): bim1 and gim1 phenotype on signal variation and single cell trajectories.
Fig 5 (Paper 2 Fig 5): Effect of microtubule end function on signal variation
Fig 6 (Paper 2 Fig 6): Bypass genetic analysis
Fig 7 (Paper 2 Fig 7): Effect of Fus3
Fig 8 (Paper 2 Fig 8): Impact on Ste5 recruitment to signaling patches.
(optional: Fig 9 (Paper 2 Fig 1c or perhaps 1e): Summary diagram of the interplay between microtubule end functions and PRS/Ste5)

Since we provide the possibility to include "Expanded View" figures that can be collapsed/expanded online by the reader (see <http://msb.embopress.org/authorguide#expandedview>), we would also suggest to include the following Expanded View figures:

Figure EV1 (Paper 1 Supp Fig S1)
Figure EV2 (Paper 2 Fig 4): Nocodazole experiments

We appreciate that you provide also a nice analysis of the hypothesis that variation in nucleus position could be linked to signal variation. While this analysis is interesting, the hypothesis is eventually rejected. In a study that is already complex and in view of the presentation issues, we

would suggest to mention briefly the outcome of this exploration in the main text but present the details in the Appendix (our MSB nomenclature for 'Supplementary information'), where interested readers could dig into those results:

Appendix

Figure S1 (Paper 2 Fig 3): Lack of correlation between nucleus position and signal output

With regard to the experiments shown in Fig 9 of paper 2 that are testing the impact of bim and gim on mating discrimination, the results obtained are complex and difficult to interpret in the light of the effect of these 2 genes on signal variation. Even if we recognize that a functional test of this nature would have been interesting, we would suggest forgoing inclusions of these data, since this aspect of the study seems to need additional experimentation and analysis to lead to robust conclusions.

With regard to the text, the reviewers, and reviewer #1 in particular, provide a series of crucial indications. Without repeating all the points, the following aspects would need to be very seriously addressed:

- The data should be made publicly available as requested by the reviewers and deposition should be made *_before_* resubmission. Availability of the datasets and the respective accession number or links should be listed in a Data availability section at the end of Materials & Methods.
- All "hyperboles" and "grandiosity" should be removed and the study should be condensed to its "essential core".
- Digressions and extensive historical accounts should be removed such that the presentation is simplified, presented in a compact manner and the main message of the study does not become diluted.
- All redundancies and double presentations (for example, avoid the use of "Restated, ...") should be removed.
- The data should be presented in a reasonably uniform and coherent way (for example using the $\eta^2(P)$ vs P plots).

Please note that we allow in principle only a single round of major revision. We would therefore strongly suggest that you send us an initial draft *_before_* formal submission such that we can potentially comment on it and possibly provide some help or editorial guidance. You can do this informally by sending us your revised combined draft by email to msb@embo.org.

When you resubmit your manuscript, please download our CHECKLIST (http://embopress.org/sites/default/files/Resources/EP_Author_Checklist_Master.xlsx) and include the completed form in your submission. **Please note** that the Author Checklist will be published alongside the paper as part of the transparent process <http://msb.embopress.org/authorguide#transparentprocess>.

If you feel you can satisfactorily deal with these points and those listed by the referees, you may wish to submit a revised version of your manuscript MSB-16-7390. Please attach a covering letter giving details of the way in which you have handled each of the points raised by the referees.

REVIEWER REPORTS FOR MSB-16-7390

Reviewer #1:

An earlier manuscript has been revised by splitting it into two separate manuscripts. This one describes a genetic screen and the categories of phenotypes obtained, while the companion manuscript delves into more mechanistic analysis of some mutant phenotypes. Here, the overall goal is to explore the genetic factors that influence how individual cells in a clonal population vary in their response to a stimulus, using the yeast pheromone response system. The authors screen a large number of viable deletion mutants, and identify a subset (roughly 50) that affect response magnitude and/or variability; these then partition into subgroups with distinct effects on specific phenotypic variables. The observations suggest that individual quantitative parameters (e.g., output and variability) are separate "axes" of system behavior, because they are under distinct genetic control.

The original manuscript was credited for thorough experimentation and convincing results. It was faulted for a lack of insightful mechanistic follow-up of the microtubule-dependent phenotypes, and for a presentation whose excesses (long-winded, jargon-filled, grandiose...) often obscured rather than clarified the central points. The mechanistic follow-up now resides in the companion manuscript. For what remains here, the presentation issues persist, and I can't say that this aspect has been improved; there is still a lot of bloat and redundancy (compounded now by unnecessary extra filler in Figures 1-3), and the manuscript contains a remarkable number of errors (often so conspicuous that I wonder how they went undetected by the numerous co-authors). So, I wouldn't say the revised text meets the editor's recommendation to "considerably simplify the narrative to achieve a clear and concise presentation".

Otherwise, the study does provide valuable contributions: its design is logical, the execution is competent, and the overall conclusions are valid and convincing. They clearly indicate that there is genetic control of signal variability, distinct from control of signal strength. A relevant question is whether this "part I" manuscript can "stand alone", separate from the part that has been split off. I suppose it can, although the repeated emphasis on the microtubule-associated factors is a constant reminder that it is a first half. Frankly, I don't think splitting the paper in two was an improvement; as suggested previously, it would have been better to do the reverse, by stripping away the grandiosity and hyperbole (e.g., comparing their screen to those of Nusslein-Volhard) to distill the story down to its essential core. Nevertheless, aside from presentation issues, the overall findings here are solid.

Major points

1. Regarding Figure 7. The results are clear but the description and interpretation are puzzling in several respects.
 - (a.) On page 9, the authors state "This result is in our contrast to our results in the original flow cytometric screen, in which Δ gim4 only increased [variability] at low doses." There are two problems here: First, Figure 5 says gim4 Δ increases variability at both high and low doses, not only at low doses. Did the authors mean to cite bim1 Δ here, rather than gim4 Δ ? Second, the authors offer no explanation for this "contrast"; it needs to be discussed and clarified. (E.g., does it reflect different strains used, or different assays, etc.?)
 - (b.) In describing the synergistic phenotype in the bim1 gim4 double mutant, the authors make the following puzzling statement (page 9): "This is thus an epistatic interaction suggesting that the two gene products might have independent effects...". This use of "epistatic" is the opposite of standard terminology. There would be epistasis if the double mutant was indistinguishable from one of the single mutants (e.g., typically, if the phenotype of mutA is "masked" in the mutA mutB double mutant, then mutB is said to be epistatic to mutA). Here, the double mutant shows synergy, not epistasis. (By convention, these mutants would be said to be in different "epistasis groups", or act independently, because they DON'T show epistasis; mutants that DO show epistasis are said to be the same epistasis group, interpreted as acting in a common pathway.)
 - (c.) The reference cited in support of the mixed-up assertion, "Fisher (1918)" is absent from the bibliography. Regardless, it is most important to get the concept straight.

Minor points

2. I previously commented that the Introduction was filled with excessive detail that was of unclear relevance and not placed in context. In their response, the authors appeared to agree, and suggest this has been fixed: "Descriptions shortened, extraneous materials moved, etc. Some ... wound up in a box, but most ... was moved to the companion paper." But this claim is manifestly untrue. The original nine paragraphs of Introduction all remain largely intact, with the exception that some of them have been segregated into a Box. As with much of the text, the Box material is excessive, and its deletion would probably improve readability.
3. There are new non-data figures added that were not in the original manuscript: Figures 1B, 1C, 2 A-B, and 3 A-B. These don't seem necessary. Their absence from the previous version posed no problem. They are largely redundant with the text and with similar figures in prior papers.

Specifically, Figure 1 and its extensive "legend" are near copies of Figure 1 from Colman-Lerner et al, 2005; Figures 2 and 3 cover methodological issues that are not especially novel or difficult to understand in text form. So, these seem to be unnecessary "filler".

4. On page 4, the text in the Box that describes Figure 1 B-C conflicts with the figure panels themselves. In describing Figure 1B, the Box text claims that "alpha-factor-responsive promoters driving the YFP and CFP reporter genes...", but the figure shows both reporters driven by the constitutive ACT1 promoter, not by regulated promoters. In describing panel 1C, the Box text says "...a pheromone responsive promoter driving YFP...", but the figure shows RFP instead. (How did this escape review by co-authors?)
5. In Figure 5, how is it possible that the same gene deletion, *ckb1Δ*, appears in two distinct clusters (II and V)? If it is a mistake, then obviously it requires fixing. If it is correct, then it requires some explanation.
6. As far as I can determine, no part of Figure S2 is ever cited or discussed. So, what is its purpose?
7. The legend to Figure 7 states that cells were "stimulated for 3h with the indicated pheromone doses". But there are no pheromone doses indicated.
8. Throughout the text, many citations are incomplete, due to missing years. (How did this escape the notice of co-authors?)
9. Page 7, middle: as was noted to be problematic with the original version, in this manuscript there are many methodological details that are redundant with the Methods section. On Page 7 of the Results this is especially true, in that it is redundant with Methods (pg 13), and some parts are also included in Fig 3A so they are triply redundant.
10. Sizable parts of the Discussion also repeat methodological details that are redundant with Results and/or Methods, including the two sub-sections on page 10. This issue was pointed out previously; despite author claims to have mitigated this, the problem persists.
11. Page 10, lower half: "This use of multiple freshly generated clonal cultures...is novel...". Frankly, this claim of novelty strikes me as an exaggeration. It is a common genetic strategy to use multiple independent clonal isolates, or multiple genetically identical spores, to ensure that phenotypes are truly the product of a particular genotype and not due to genetic changes during outgrowth. So, this is not "novel", it is standard practice.
12. Page 11, bottom: "Our work showed that two genes involved in microtubule function: GIM4 and PAC10/GIM2...". The latter gene, PAC10/GIM2, was never mentioned in the Results or anywhere else previously in the manuscript. Here, they appear out of the blue, and nothing was previously "showed" about them.
13. Typos and grammar, etc.:
 - (a) Page 2, second sentence of Intro: "Although" and "however" are redundant.
 - (b) Page 2, second paragraph, first sentence: "Yet cells, ...". Re-write to fix grammar.
 - (c) Page 2, bottom: it is unlikely that "Novick and Weiner, 1957" is relevant to the point about the pheromone response system being discussed.
 - (d) Page 10, middle: "PHDDVC collection" is repetitive; the "C" stands for "collection".

Reviewer #2:

My task here was to examine if the revised version of manuscript 77387 (called P1 in the response letter from the authors) covering the screen, if published back-to-back with another study covering mechanistic investigations (called P2), correctly addressed the points I had raised from the initial submission, and to give an opinion on whether both studies (P1 and P2) stand alone.

Manuscript P2 is difficult to read. Text/Legends/Figures often do not match (e.g. p7 text related to

Htb2-YFP « Figure 3a shows the results » and then 3A legend corresponds to Spc42-GFP. Similarly legend 3C corresponds to a panel labelled D on the figure). To quote a correspondance from the authors to the editor: «It's as if the bell rang and these --reviewers/authors-- had no more time ». My task here was not to review P2 but to search for information in it, which was not easy.

My major critics on the initial submission of P1 were :

1. Availability of microscopy images of the 42 « validated » genes.

Answer from the authors is « We now present all the microscopy results». Specifying where in the two papers these details are provided would have been helpful to me.

What is actually presented is :

-in P1 : Nothing in main manuscript. SI provides information about why and how the microscopy analysis was done, and on which strains. It also writes p16 that «Table S6 contains all data from the microscope cytometry screens ». Unfortunately, there was no Table S6 in the archive I received. All the microscopy results initially presented in P1 have now been transferred to P2.

-in P2 : This ms. is focused on bim1+gim4, and does not mention microscopy data for 42 genes.

So, this point has not been addressed.

2. Availability of the raw flow cytometry data.

Answer from the authors : « We are in the process of depositing the data at flowrepository.org ».

So, this point has not (yet) been addressed.

3a. Discuss possible distinction between i) variability of predisposition before pathway induction and ii) variability of induction.

The authors' response is that the Fus3 and Ste5 related results of P2 support an erratic signalling which « is consistent with ii) ». I would argue that the results of P2 also seem consistent with i) : If initial concentration or intra-cellular distribution of Fus3p is crucial for bim1 effect, then the results on kss1delta and fus3delta mutants would also make sense. Anyway, I had not asked for an answer to this distinction, but for discussing it, which is done neither in P1 nor in P2.

So, this point has not been addressed.

3b. Examine cell morphology of bim1 and gim4 mutants and discuss possible implication.

Authors answer : « ...do did not observe this », which I guess stands for « we did not observe this » and :

« we now include a paragraph describing our observations and our comparison with our own observations with Ohya 2005 as part of SI for P2 ». This is grammatically difficult to follow. There is no such paragraph in P2 main text, and I did not receive SI of P2 in the documents.

So, this point has possibly been addressed, but I can't say.

4. Better discuss possible mechanisms, which now concerns manuscript P2. I had asked to discuss :

- A possible feedback of proper MT organization on signalling (preparation for karyogamy).

Authors found this point interesting (Thank you! Am I among the 3/4 « good » reviewers mentioned in correspondance to editor?) but do not discuss it in P2.

So, this point has not been addressed.

- A possible effect of improper oscillations of the nucleus on local cytosolic concentrations.

Authors discuss the possible link between nucleus movements and signalling in P2. They prefer not speculating on the cytosolic homogenization possibility and I understand.

So, this point has been partly and satisfyingly addressed.

- Nuclear export of Fus2p to the tip.

This export (and ref) is now mentioned in legend 1e of P2. It is not specifically discussed but I understand that not all possibilities can.

So, this point has been addressed.

In conclusion, very few of the major points I had raised have been addressed.

Regarding the fullness of each study : The main interest of initially-submitted P1 was that the screen led to interesting biological observations, especially the differences in traces, which has now been transferred to P2 (Fig 2 there). As it is now, P1 describes a screen and a list of genes « validated », which does not correspond to the type of articles I am used to reading in MSB. Also, results in P2 could be presented in a more concise way, and many long sentences in both manuscripts tend to dilute the key messages. My recommendation to the editor is therefore to ask for a single, streamlined and coherent article, that goes from a screen to mechanistic conclusions, rather than two partial studies.

REVIEWER REPORTS FOR MSB-16-7391

Reviewer #1:

A companion manuscript describes a genetic screen for mutants that affect the variability of signaling in the yeast pheromone response system. This manuscript pursues mechanistic analysis of those mutants that affect microtubule functions. The experiments analyze an array of microtubule perturbations to describe those that do and do not affect signaling variability, and test some specific models. Using signaling bypass methods and deletion of distinct MAP kinases, the authors constrain which step(s) in the pathway are susceptible to microtubule-controlled variation. They further show that these microtubule perturbations disrupt the polarized membrane localization behavior of the signaling scaffold protein, Ste5.

In general, the experiments and data here are extensive, and the logic behind the approaches is sound. Ultimately, however, it remains unclear why some microtubule perturbations (and not others) affect signaling variability, and hence the responsible mechanisms remain enigmatic. The paper is admirably incisive in testing the "nuclear position" model, which is shown to be insufficient to explain the results. It is less incisive and more speculative in addressing alternative models. For example, the type of analysis that is so effective in disproving the "nuclear position" model (i.e., poor correlation of output with nucleus-to-tip distance; Fig 3D), is not applied to later tests of Ste5 patch polarization in Fig 8 (i.e., do the effects on Ste5 polarization correlate with signaling variation?). Consequently, while there are many relevant experiments, the mechanistic insights remain somewhat limited. The findings serve to constrain viable models without really nailing down a specific model. Perhaps that is as good as one can expect at this point, though this ambiguity is not really explicitly acknowledged in the paper. Instead, the Discussion is long-winded and philosophical about issues that are not concretely advanced here. The manuscript also contains numerous errors.

Major points

1. Regarding studies of Ste5 polarization in Figure 8:

(a) It is unclear if the effects of mutations on Ste5 polarization are due to increases in signaling variability or to reductions in signal strength. One way this could be clarified is to test those perturbations that do NOT increase variability: namely, nocodazole treatment, and the *kar1-15* mutant (e.g., Figs 4-5). But such tests were not shown. If these were tested, they would be valuable to include here; if not, then the authors should at least provide a clear discussion of this caveat and the alternative interpretations.

(b) As was so useful in Figure 3D, here it also could be very useful to know whether the signaling output of individual cells correlates with their time of polarization initiation or duration (i.e., do cells with weakest signaling show the slowest initiation and/or shortest duration?). This seems worth testing; or, if technically prohibitive, at least discuss the issue explicitly. It seems that such information would illuminate the issue of the physiological relevance of variability.

2. Regarding the bypass activator experiments in Figure 6:

(a) It would help to plot the response of the same strains to pheromone, as a top row, in order to clearly compare effects on pheromone signaling versus bypass signaling. The Results text makes several references to the normal pheromone response, but the data are unavailable; the companion paper includes no plot of dose response for *bim1* and *gim4* mutants, and no plots of the sort used in panel 6a.

(b) To a non-trivial extent, the utility of the bottom two rows of data is limited and redundant with later experiments with *fus3Δ* mutants. That is, *Fus3* is not activated in experiments involving either *Ste11-4-Ste7* (because they use *ste5Δ* cells) or *Ste12* (because there is no kinase signaling). Since variation is suppressed by removing *Fus3*, then the suppression of variation by these bottom two rows can be explained for the same reason, and hence they don't provide further information relevant to the position in the pathway where signaling is initiated.

3. It is not clear why related analyses in Figures 4, 5, and 6 are plotted differently: i.e., variability is alternately plotted as a function of signal $\langle p \rangle$ (Fig 4), or output $[O]$ (Fig 5), or a ratio $[P_{\text{mutant}} / \max(P_{\text{reference}})]$ (Fig 6). A priori, I would think that plotting these all in the same way would facilitate comparison. If there is a sensible reason for plotting them differently, then it should be explained briefly. Currently, it seems as if the approach changes randomly.

4. In the Abstract and Discussion, the authors repeatedly suggest that their findings imply a role for microtubules in "reducing" or "suppressing" signaling variability (e.g., pg 2, Abstract; pg 15, bottom; pg 17, middle, etc.). It's not clear to me that this is correct, because depolymerization of microtubules by nocodazole (Fig 4) does not increase variability. Thus, rather than microtubules being required to suppress variability, it seems more accurate to say that variability can be induced or increased when microtubule function is disrupted (and only in very specific ways).

Minor points

5. Page 7, top half: The description of experiments in Figure 3 is confusing. It introduces the set-up, and then states that "Figure 3a shows the results". Only after extensive confusion did it become evident that this section is actually describing Figure 3b instead. Not only is the citation erroneous, but it is unclear why the authors choose to describe Figure 3b first and 3a second.

6. Figure 3: Panel C lacks a legend. (The current legend for "C" is actually for panel D.) Also, panel C has mysterious characters that seem to have been mistranslated (" $7m$ " and " $\#10^{-3}$ "). Finally, because panel C lacks a legend, the Results text in page 8 (middle) is confusing, as it is unclear how the rate of change is measured or expressed.

7. The strain list in Table S1 is severely mangled. Individual genotypes are split among as many as four (non-contiguous) pages. Plus, some "names" and "descriptions" are masked by adjacent columns. It is difficult to imagine how this could go undetected if the manuscript materials were examined carefully by co-authors prior to submission.

8. Page 12: Bypass results should be interpreted with care. The best language is that used at the end of the top paragraph: "...at or upstream of the recruitment of *Ste5* to the membrane". Later language, in the third paragraph, is too strong: "...the reduction...caused by *Tub1-828* expression mapped to the *Ste5* recruitment step." It is more accurate to say it maps "at or upstream" of this step. (E.g., if *Tub1-828* affected the level or function of the α subunit, this would affect the bypass by *Ste4* but not by *Ste5-CTM*; such possibilities are obscured by claiming the effect "maps to *Ste5* recruitment".)

9. Page 18, end of top paragraph: "This series of experiments thus extends and generalizes the scope of a classical genetic method first used in the 1930s...". This strikes me as self-aggrandizing

hyperbole, as does a companion claim in the correspondence ("...a triumphant extension of classical genetic epistasis ala Ephrussi and Beadle..."). In fact, the bypass epistasis tests performed here are a rather standard. Moreover, they use reagents that have been used for similar purposes in prior studies.

10. Page 2, bottom: "S to P stimulation" is undefined.

11. Page 2, bottom: "Moreover, each arm stimulates the other's activity". This claim is questionable, and no citation is provided to support it. It is contradicted by evidence in the literature; e.g., in Nern & Arkowitz 1998 (PMID: 9428768), disrupting the "polarity arm" does not affect signaling.

12. Page 5, bottom: "P_PRM1 promoter" is redundant, as is "P_ACT1 promoter", because the "P" stands for promoter.

13. Page 18, bottom: "...including Cdc42, Cdc20, and Ste5...". "Cdc20" is incorrect. Presumably either Cdc24 or Ste20 was intended here.

14. Page 19, top: The sentence beginning with "This picture is consistent..." is incomplete; the idea here is never finished.

15. Page 19, middle: The sentence beginning "Moreover, in some cases, ..." is incoherent.

16. Page 20: The "Box" is dispensible. It would suffice to cite the literature, without rehashing.

17. Figure 1A: "a-factor" should say "alpha-factor".

18. Figure 1D: In "step 1", Cdc42 is shown as being phosphorylated. Surely the authors know the difference between phosphorylation and GTP exchange. Did any of them examine this figure?

19. Figure 6: The pathway cartoons on the left-hand side of each panel are so small that they will never be visible when published. Probably best to just delete them.

20. Figure 8B: In the bar graphs at left, the error bars are undefined. In the table at right, there is no mention in the legend of what statistical test is used, or what is N (number of trials).

21. Figure 9:

(a) Panel A says "matalpha1". This is incorrect. It should be "mfalpha1".

(b) The legend for panel B says 3 biological replicates for each strain, but the second and fourth charts have only 2 symbols visible for each strain.

Reviewer #2:

In this revised study, Pesce et al. characterize the effects of two genes involved in microtubule dynamics on the *S. cerevisiae* pheromone response pathway. Overall, the authors claim that cells lacking the microtubule end binding protein Bim1 or the microtubule biogenesis modulator Gim4 exhibit increased noise in signaling output of the pheromone response pathway.

Major comments:

In my opinion, the manuscript does not meet the expectations required for publication in Molecular Systems Biology. Firstly, the biological insights are already included in the accompanying paper describing the genetic screen, and thus this manuscript does not provide sufficient insight how microtubule dynamics contributes to signaling fidelity. Secondly, the authors use sophisticated quantitative readouts and statistical analysis to document rather minor differences that are hard to interpret. For example, I am confused about sentences like "showing at least marginally significant large reductions". Finally, a number of claims are overstated. For example, the authors perform some experiments using various dominant active mutants, but fail to show convincing evidence that lack of microtubule dynamics interferes with pheromone signaling. Nevertheless, they draw

relatively far reaching conclusions based on the data. Similarly, the claim that the increased noise is dependent on Fus3 is not warranted based on the data shown in Figure 7.

Minor comment:

The legend of Figure 4 is missing some information, "strains in right column were and perturbed by Tub1-828 expression".

MSB-16-7390 & MSB-16-7391 Merged

MSB-16-7390 & MSB-16-7391 Merged, 1st Submission - authors' response

20 November 2017

Here is a revised version of MSB-16-7390, now titled "Single-cell profiling screen identifies microtubule-dependent reduction of variation in cell signaling". It has benefitted immensely from the extensive work we performed in response to the extensive work and comments carried out by you and the reviewers. We believe it is now ready for publication.

We detail our specific responses below.

With thanks for your consideration and hard work.

MSB-16-7390, Pesce et al. "Single-cell profiling screen identifies microtubule-dependent reduction of variation in cell signaling"

Thank you again for submitting your work to Molecular Systems Biology and I apologize again for the long process. We have now finally heard back from the referees who agreed to evaluate your two manuscripts. Reviewer #1 in paper 1 (MSB-16-7390) and paper 2 (MSB-16-7391) was involved in the previous round (as Reviewer #1 of MSB-15-6386). Reviewer #2 in paper 1 was involved in the previous round as well (as Reviewer #3 of MSB-15-6386). Reviewer #2 in paper 2 was new to this round. As you will see from the reports below, the referees find the topic of your study of potential interest. They remain however globally lukewarm and they still raise significant concerns, which, I am afraid to say, preclude publication of the manuscripts in their present form.

With regard to the new configuration of having 2 separate papers, I am afraid that the reviewers were not supportive. Thus reviewer #1 of paper 1 notes "I don't think splitting the paper in two was an improvement; as suggested previously, it would have been better to do the reverse" and reviewer #2 of paper 1 recommends us "to ask for a single, streamlined and coherent article, that goes from a screen to mechanistic conclusions, rather than two partial studies". Reviewer #2 of paper 2 feels that "the biological insights are already included in the accompanying paper describing the genetic screen, and thus this manuscript [paper 2] does not provide sufficient insight how microtubule dynamics contributes to signaling fidelity".

In terms of presentation and clarity, I am afraid that the reviewers were also not convinced and all of them raised numerous issues. For example reviewer #1 states "the presentation issues persist, and I can't say that this aspect has been improved; there is still a lot of bloat and redundancy", reviewer #2 feels that "results in P[aper]2 could be presented in a more concise way, and many long sentences in both manuscripts tend to dilute the key messages".

After re-reading the two manuscripts again, we do agree with these evaluations and we feel that the quality of the presentation still needs major improvements and the overall length of the text should be drastically condensed to improve focus and clarity. Based on these clear recommendations, we feel therefore that it would not be possible to publish the two manuscripts as separate standalone papers in Molecular Systems Biology.

Having said that, if we make abstraction of presentation issues and of the splitting of the study into two manuscripts, we agree with reviewer #1 that the collective scientific message of the combined papers remains of interest. Even if we realize that the ultimate mechanistic details remain to be elucidated, as pointed out by reviewer #2 of paper 2, we nevertheless feel that a combined study that presents the global analysis of the large scale screen together with the follow up genetic analysis and cell biology characterization of the bim1 and gem1 mutant would provide sufficient insights into interesting regulatory mechanisms of cell-to-cell signaling variability.

Given that the reviewers did not support publication of paper 2 (MSB-16-7391) as a stand alone paper and given that we had initially asked for a major revision of MSB-15-6386, we would therefore ask you to carefully prepare a revision of paper 1 (MSB-16-7390) by including the key data (see below for details) of paper 2 (MSB-16-7391) such that paper 1 becomes a complete study as requested by the reviewers. I am afraid however that we cannot offer to publish paper 2 (MSB-16-7391) as an independent manuscript in Molecular Systems Biology (for simplicity, I include the same decision letter for MSB-16-7391).

Understood.

Such a single combined revised manuscript should address the points raised by the current reviewers and should focus on the key aspects of the study so that it delivers a coherent and complete message. We would thus kindly ask that only the following figures should be included in the main paper:

*Fig 1 (Paper 1 Fig 1): overview of the PRS, noise def & high throughput design
 Fig 2 (Paper 1 Fig 4): distribution of signal variation over >100 strains screened
 Fig 3 (Paper 1 Fig 5): Clustering of 50 genes influencing variation
 Fig 4 (Paper 1 Fig 7 + Paper 1 Fig 2): bim1 and gim1 phenotype on signal variation and single cell trajectories.
 Fig 5 (Paper 2 Fig 5): Effect of microtubule end function on signal variation
 Fig 6 (Paper 2 Fig 6): Bypass genetic analysis
 Fig 7 (Paper 2 Fig 7): Effect of Fus3
 Fig 8 (Paper 2 Fig 8): Impact on Ste5 recruitment to signaling patches.
 (optional: Fig 9 (Paper 2 Fig 1c or perhaps 1e): Summary diagram of the interplay between microtubule end functions and PRS/Ste5)*

Since we provide the possibility to include "Expanded View" figures that can be collapsed/expanded online by the reader (see <http://msb.embopress.org/authorguide#expandedview>), we would also suggest to include the following Expanded View figures:

*Figure EV1 (Paper 1 Supp Fig S1)
 Figure EV2 (Paper 2 Fig 4): Nocodazole experiments*

Done. The EV figures are actually ideal for results like these. We describe how we carry these out in the Appendix.

We appreciate that you provide also a nice analysis of the hypothesis that variation in nucleus position could be linked to signal variation. While this analysis is interesting, the hypothesis is eventually rejected. In a study that is already complex and in view of the presentation issues, we would suggest to mention briefly the outcome of this exploration in the main text but present the details in the Appendix (our MSB nomenclature for 'Supplementary information'), where interested readers could dig into those results:

Appendix

Figure S1 (Paper 2 Fig 3): Lack of correlation between nucleus position and signal output.

Understood. Done. The lack of correlation now appears on pages 40-42 of the Appendix.

With regard to the experiments shown in Fig 9 of paper 2 that are testing the impact of bim and gim on mating discrimination, the results obtained are complex and difficult to interpret in the light of the effect of these 2 genes on signal variation. Even if we recognize that a functional test of this nature would have been interesting, we would suggest forgoing inclusions of these data, since this aspect of the study seems to need additional experimentation and analysis to lead to robust conclusions.

Suggestion adopted; these experiments deleted.

With regard to the text, the reviewers, and reviewer #1 in particular, provide a series of crucial indications. Without repeating all the points, the following aspects would need to be very seriously addressed:

- The data should be made publicly available as requested by the reviewers and deposition should be made before resubmission. Availability of the datasets and the respective accession number or links should be listed in a Data availability section at the end of Materials & Methods.

Data are now archived in the "Dryad" repository doi:10.5061/dryad.67bc0 and at <http://authors.fhcrc.org/1202/> with doi:10.6076/J77D2S8Q. These links and numbers in the data availability section at the end of Materials and Methods.

- All "hyperboles" and "grandiosity" should be removed and the study should be condensed to its "essential core".

Done

- Digressions and extensive historical accounts should be removed such that the presentation is simplified, presented in a compact manner and the main message of the study does not become diluted.

Done

- All redundancies and double presentations (for example, avoid the use of "Restated, ...") should be removed.

Done. And writing immensely improved by so doing.

- The data should be presented in a reasonably uniform and coherent way (for example using the $\eta^2(P)$ vs P plots).

Done. In some cases we added additional information about P . For example in Figure 6, when we were turning on the pathway at downstream steps using ectopic expression of proteins induced by estradiol, and where the meaning of "transmitted signal", P , is less clear, to help the reader, we plotted $\eta^2(P)$ vs P and also then plotted P vs estradiol dose.

Please note that we allow in principle only a single round of major revision. We would therefore strongly suggest that you send us an initial draft before formal submission such that we can potentially comment on it and possibly provide some help or editorial guidance. You can do this informally by sending us your revised combined draft by email to msb@embo.org.

We sent the revised combined draft to editor, Thomas Lemberger, before submission, and have based this resubmission on his overview.

*When you resubmit your manuscript, please download our CHECKLIST (http://embopress.org/sites/default/files/Resources/EP_Author_Checklist_Master.xlsx) and include the completed form in your submission. *Please note* that the Author Checklist will be published alongside the paper as part of the transparent process <http://msb.embopress.org/authorguide#transparentprocess>.*

Done, checklist filled out.

If you feel you can satisfactorily deal with these points and those listed by the referees, you may wish to submit a revised version of your manuscript MSB-16-7390. Please attach a covering letter giving details of the way in which you have handled each of the points raised by the referees.

Done, actions detailed below.

[Instructions to authors on submitted manuscript]

Reviewer #1:

An earlier manuscript has been revised by splitting it into two separate manuscripts. This one describes a genetic screen and the categories of phenotypes obtained, while the companion manuscript delves into more mechanistic analysis of some mutant phenotypes. Here, the overall goal is to explore the genetic factors that influence how individual cells in a clonal population vary in their response to a stimulus, using the yeast pheromone response system. The authors screen a large number of viable deletion mutants, and identify a subset (roughly 50) that affect response magnitude and/or variability; these then partition into subgroups with distinct effects on specific phenotypic variables. The observations suggest that individual quantitative parameters (e.g., output and variability) are separate "axes" of system behavior, because they are under distinct genetic control.

Concur with summary.

The original manuscript was credited for thorough experimentation and convincing results. It was faulted for a lack of insightful mechanistic follow-up of the microtubule-dependent phenotypes, and for a presentation whose excesses (long-winded, jargon-filled, grandiose...) often obscured rather than clarified the central points. The mechanistic follow-up now resides in the companion manuscript. For what remains here, the presentation issues persist, and I can't say that this aspect has been improved; there is still a lot of bloat and redundancy (compounded now by unnecessary extra filler in Figures 1-3), and the manuscript contains a remarkable number of errors (often so conspicuous that I wonder how they went undetected by the numerous co-authors). So, I wouldn't say the revised text meets the editor's recommendation to "considerably simplify the narrative to achieve a clear and concise presentation".

Concur with the reviewer's evaluation. We believe we have fixed the issues that the reviewer raises.

Otherwise, the study does provide valuable contributions: its design is logical, the execution is competent, and the overall conclusions are valid and convincing. They clearly indicate that there is genetic control of signal variability, distinct from control of signal strength. A relevant question is whether this "part I" manuscript can "stand alone", separate from the part that has been split off. I suppose it can, although the repeated emphasis on the microtubule-associated factors is a constant reminder that it is a first half. Frankly, I don't think splitting the paper in two was an improvement; as suggested previously, it would have been better to do the reverse, by stripping away the grandiosity and hyperbole (e.g., comparing their screen to those of Nusslein-Volhard) to distill the story down to its essential core. Nevertheless, aside from presentation issues, the overall findings here are solid.

Done. We prepared a single paper in which we removed the grandiosity and hyperbole. The comparison in the previous response with N-V's work was not meant to convey that this screen was as wonderful or enlightening as their screen, but rather that their screen and later work (I am thinking of Kathryn Anderson's work on the D-V axis) revealed and to some extent validated the existence of distinct axes as an aspect of systems that genetic analyses can sometimes reveal.

Major points

1. Regarding Figure 7. The results are clear but the description and interpretation are puzzling in several respects.

(a.) On page 9, the authors state "This result is in our contrast to our results in the original flow cytometric screen, in which $\Delta gim4$ only increased [variability] at low doses." There are two problems here: First, Figure 5 says $gim4\Delta$ increases variability at both high and low doses, not only at low doses. Did the authors mean to cite $bim1\Delta$ here, rather than $gim4\Delta$? Second, the authors offer no explanation for this "contrast"; it needs to be discussed and clarified. (E.g., does it reflect different strains used, or different assays, etc.?)

We are very grateful to the reviewer for this point. As a consequence, we reexamined quantitative data from our many assays in the screen. The results revealed that the screen worked, but also showed the extent to which multiple assays were required in order to establish "truth". Fixing this required careful rewording of more conclusions than the above one.

Revised text now reads:

"With a few exceptions (for example $\Delta ckb1$, $\Delta his1$, and $\Delta sky1$) either all or all but one of the independent segregants bearing each gene deletion grouped in the same subcluster. Taken together with the results of the tertiary screen, these results show that differences in variability in strains with different gene deletions were due to the mutations. Since all 19 cultures of the SGA85 reference cells were isogenic, that 5 of these cultures grouped into different clusters highlights high throughput flow cytometric assays have some inconsistent performance. Similarly, since our independent haploid segregants came from crosses with an otherwise isogenic MAT α strain, we believe that the observed infrequent grouping of any single deletion's isolates into multiple clusters most likely reflects measurement anomalies rather than uncharacterized genetic differences between the MAT α and MAT α parents of the strains." [...]

"Although deletions of both *BIM1* and *GIM4* caused elevated $\eta^2(P)$ in the primary screen at both low and high doses, *Abim1* did not show elevated $\eta^2(P)$ at low doses in the secondary screen, but showed elevation at both doses in the tertiary screen. We again took these differences in measured $\eta^2(P)$ values as likely indicating the limitations of such measurements via the relatively high throughput culture in multiwell plate/ flow cytometry assays rather than arising from otherwise cryptic genetic variability among isolates.

However, to address the above possibility, and get around possible effect of uncharacterized genetic heterogeneity among independently haploid deletion strains resulting from independent meioses, we remade these strains without meiosis, in a clean genetic background, an independently constructed BY4741 derivative, equivalent to the reference strain (see Appendix). From this strain (GPY4000), we...."

(b.) In describing the synergistic phenotype in the bim1 gim4 double mutant, the authors make the following puzzling statement (page 9): "This is thus an epistatic interaction suggesting that the two gene products might have independent effects...". This use of "epistatic" is the opposite of standard terminology. There would be epistasis if the double mutant was indistinguishable from one of the single mutants (e.g., typically, if the phenotype of mutA is "masked" in the mutA mutB double mutant, then mutB is said to be epistatic to mutA). Here, the double mutant shows synergy, not epistasis. (By convention, these mutants would be said to be in different "epistasis groups", or act independently, because they DON'T show epistasis; mutants that DO show epistasis are said to be the same epistasis group, interpreted as acting in a common pathway.)

(c.) The reference cited in support of the mixed-up assertion, "Fisher (1918)" is absent from the bibliography. Regardless, it is most important to get the concept straight.

Since about 100 years ago, there have been two distinct meanings of the term "epistasis" used by geneticists, and both are in common use. The reviewer mentions what we would call "Bateson epistasis"

"used in 1909 to describe a masking effect whereby a variant or allele at one locus (denoted at that time as an 'allelomorphic pair') prevents the variant at another locus from manifesting its effect."

While we were referring to the second meaning of the term ("Fisher epistasis"). See below.

"The situation has been confused further by the fact that in quantitative genetics, following a paper by Fisher in 1918, the term 'epistatic' has been generally used in yet another different sense from its original usage. In Fisher's 1918 definition, epistasis refers to a deviation from additivity in the effect of alleles at different loci with respect to their contribution to a quantitative phenotype. This definition is not equivalent to Bateson's 1909 definition, as was pointed out in the initial review of Fisher's 1918 paper by R.C. Punnett."

(both of the above quotes are from a review by Heather Cordell (2002). "Epistasis: what it means, what it doesn't mean, and statistical methods to detect it in humans". Human Molecular Genetics, 2002, Vol. 11, 2463–2468)

This second, Fisher, meaning of the word "epistasis" is in common, daily use among yeast geneticists who study gene-gene interactions (eg Boone and Andrews in Toronto, and Krogan and others in San Francisco).

We apologize for introducing confusion. In the revised manuscript, we try to fix the issue by losing the term altogether.

New passage reads....

"In $\Delta bim1 \Delta gim4$ cells, the increase in signal variation $\eta^2(P)$ was more than twice as large as the measured effect of the two individual deletions. This synergistic genetic interaction suggested that the two gene products acted through distinct mechanisms to affect $\eta^2(P)$ (see for example Fisher 1918, Boone et al. 2007)."

The references are to a 10 year old paper by Boone and Andrews that discusses the different meanings of the term "epistasis", and to the original paper by Fisher. Bibliography references both, correctly.

We apologize for introducing a source of confusion when we didn't need to, and we hope these actions straighten it out.

Minor Points.

2. I previously commented that the Introduction was filled with excessive detail that was of unclear relevance and not placed in context. In their response, the authors appeared to agree, and suggest this has been fixed: "Descriptions shortened, extraneous materials moved, etc. Some ... wound up in a box, but most ... was moved to the companion paper." But this claim is manifestly untrue. The original nine paragraphs of Introduction all remain largely intact, with the exception that some of them have been segregated into a Box. As with much of the text, the Box material is excessive, and its deletion would probably improve readability.

Well, we might not agree with the assertion that we had removed extraneous material from the introduction was "untrue". In fact, in the previous paper, we had exiled 4 out of the 9 of the offending paragraphs to a box and toned down the rest.

However here, in this revised paper, the introduction is wholly different, now 3 paragraphs, and it has no box.

We do believe that, for some or most readers, inclusion of material carefully describing cell-cell variation is helpful, even critical for any informed understanding of quantitative study of it. For this reason, we further exiled this information, including that in the former Box, to the Appendix. We genuinely believe that making this available will help at least some readers better understand and assess this work.

3. There are new non-data figures added that were not in the original manuscript: Figures 1B, 1C, 2 A-B, and 3 A-B. These don't seem necessary. Their absence from the previous version posed no problem. They are largely redundant with the text and with similar figures in prior papers. Specifically, Figure 1 and its extensive "legend" are near copies of Figure 1 from Colman-Lerner et al, 2005; Figures 2 and 3 cover methodological issues that are not especially novel or difficult to understand in text form. So, these seem to be unnecessary "filler".

All of the figures mentioned are now exiled to the Appendix. The former 1B and 1C are explanatory, largely cover previously published work and were previously part of the Box. We agree 2A was not novel or difficult to understand in text form. However, we respectfully submit that the former Figure 2B, the exact layout of the genetic elements including the complex setup of the terminators in the reporter gene, is novel in some key respects and, in the absence of a figure, *extraordinarily* difficult to understand, even from our own text.

Similarly, the former 3A is not difficult but we submit that former 3B, the layout of lasers in the flow cytometer, was critical and highly non-intuitive-- we call the reader's attention to our dichroic

mirror setup to be able to use a 532nm wavelength laser to collect RFP emission but also emissions at 550nm from YFP.

Again, we exiled all this material to the Appendix, where we hope that it might help some readers.

4. On page 4, the text in the Box that describes Figure 1 B-C conflicts with the figure panels themselves. In describing Figure 1B, the Box text claims that "alpha-factor-responsive promoters driving the YFP and CFP reporter genes...", but the figure shows both reporters driven by the constitutive ACT1 promoter, not by regulated promoters. In describing panel 1C, the Box text says "...a pheromone responsive promoter driving YFP...", but the figure shows RFP instead. (How did this escape review by co-authors?)

In these Figures, now in the Appendix, we fixed these errors. We thank the reviewer and apologize for the mistake.

5. In Figure 5, how is it possible that the same gene deletion, ckb1Δ, appears in two distinct clusters (II and V)? If it is a mistake, then obviously it requires fixing. If it is correct, then it requires some explanation.

We are grateful for this observation. Together with her/his Major Point 1a, this criticism forced us to describe in some detail the fact that these assays, conducted as carefully as we could, carried with them a substantial background of inconsistent measurements (eg. 5 out of 19 reference strain cultures appearing in subclusters not in the main subcluster). This was an important enough point that we included it in the results, prominently, as detailed in our response to Major Point 1a.

6. As far as I can determine, no part of Figure S2 is ever cited or discussed. So, what is its purpose?

With thanks to the reviewer. Gone from Main Text.

7. The legend to Figure 7 states that cells were "stimulated for 3h with the indicated pheromone doses". But there are no pheromone doses indicated.

We thank the reviewer for this observation. Dose is so important for this argument, and subsequent arguments, that we added a cognate Figure 7B that shows P, cumulative transmitted signal as function of different specific doses

8. Throughout the text, many citations are incomplete, due to missing years. (How did this escape the notice of co-authors?)

We apologize for the omissions.

9. Page 7, middle: as was noted to be problematic with the original version, in this manuscript there are many methodological details that are redundant with the Methods section. On Page 7 of the Results this is especially true, in that it is redundant with Methods (pg 13), and some parts are also included in Fig 3A so they are triply redundant.

We believe we have eliminated these unnecessary repetitions.

10. Sizable parts of the Discussion also repeat methodological details that are redundant with Results and/or Methods, including the two sub-sections on page 10. This issue was pointed out previously; despite author claims to have mitigated this, the problem persists.

We did leave methodological details in the previous discussion in cases where we thought these constituted valid discussion points. We now have eliminated these from the discussion and moved them, together with their claims of novelty, to the Appendix.

11. Page 10, lower half: "This use of multiple freshly generated clonal cultures....is novel...". Frankly, this claim of novelty strikes me as an exaggeration. It is a common genetic strategy to use multiple independent clonal isolates, or multiple genetically identical spores, to ensure that

phenotypes are truly the product of a particular genotype and not due to genetic changes during outgrowth. So, this is not "novel", it is standard practice.

Of course the reviewer is correct that the use of independent clones is standard practice. However, it is novel *in screens of this type*-- which either use cultures derived from generations of single haploid cells, which have picked up second site suppressors and aneuploidies, or from patches of cells, products of many spores mixed together, from colonies that had grown together on the haploid selection plates.

12. Page 11, bottom: "Our work showed that two genes involved in microtubule function: GIM4 and PAC10/GIM2...". The latter gene, PAC10/GIM2, was never mentioned in the Results or anywhere else previously in the manuscript. Here, they appear out of the blue, and nothing was previously "showed" about them.

Indeed. With thanks to the reviewer for catching this.

First appearance in the results now reads:

Three genes that affected $\eta^2(P)$ were known to affect microtubule function. These were *GIM4* and *PAC10/GIM2*, whose products form part of a six-protein prefoldin complex needed for tubulin supply, and *BIMI*, whose product mediates attachment of cytoplasmic microtubule plus-ends to the signaling site. We selected two of these genes, *BIMI* (2 out of 3 in Cluster IIIa) and *GIM4* (*GIM4* 2 out of 3 in cluster IIa) as candidate genes to explore a possible relationship between microtubule function and signal variability."

Note that, although this is precisely what we did, we are left with describing an arbitrary choice.

13. Typos and grammar, etc.:

(a) Page 2, second sentence of Intro: "Although" and "however" are redundant.

Gone from introduction

(b) Page 2, second paragraph, first sentence: "Yet cells, ...". Re-write to fix grammar.

Gone from introduction.

(c) Page 2, bottom: it is unlikely that "Novick and Weiner, 1957" is relevant to the point about the pheromone response system being discussed.

Indeed and with thanks. Gone from introduction

(d) Page 10, middle: "PHDDVC collection" is repetitive; the "C" stands for "collection".

Since we are no longer bragging about the selection, this is no longer in the main text. And we lost the C from the collection of comments.

Summary response to Reviewer 1. We are very grateful for the fact that the reviewer spent so much effort on her/ his response, and in particular for their care cataloging so many specific mistakes we were able to address. We hope we have corrected the deficiencies identified.

Reviewer #2:

My task here was to examine if the revised version of manuscript 77387 (called P1 in the response letter from the authors) covering the screen, if published back-to-back with another study covering mechanistic investigations (called P2), correctly addressed the points I had raised from the initial submission, and to give an opinion on whether both studies (P1 and P2) stand alone.

Manuscript P2 is difficult to read. Text/Legends/Figures often do not match (e.g. p7 text related to Htb2-YFP « Figure 3a shows the results » and then 3A legend corresponds to Spc42-GFP. Similarly legend 3C corresponds to a panel labelled D on the figure). To quote a correspondance from the

authors to the editor: «It's as if the bell rang and these --reviewers/authors-- had no more time ». My task here was not to review P2 but to search for information in it, which was not easy.

We understand the difficulty of finding information in previous versions of these manuscripts. To some extent this reflects the unusual complexity of this body of work. We've worked hard to make the information more accessible, and hope we have succeeded.

My major critics on the initial submission of P1 were :

1. Availability of microscopy images of the 42 « validated » genes.

Answer from the authors is « We now present all the microscopy results ». Specifying where in the two papers these details are provided would have been helpful to me.

What is actually presented is :

-in P1 : Nothing in main manuscript. SI provides information about why and how the microscopy analysis was done, and on which strains. It also writes p16 that «Table S6 contains all data from the microscope cytometry screens ». Unfortunately, there was no Table S6 in the archive I received. All the microscopy results initially presented in P1 have now been transferred to P2.

-in P2 : This ms. is focused on bim1+gim4, and does not mention microscopy data for 42 genes.

So, this point has not been addressed.

We have dealt with this point with the data we have. We do not have the raw image data for the strains in the tertiary screen, only the raw data from different fluorescence channels for each cell. We included a table of these values as an Excel file in our previous submission, but the conversion to pdf at the journal either lost it or rendered it non-human readable. We present this data, all of it, in Appendix Table S6. We now mention our examination of morphology of $\Delta bim1$ and $\Delta gim4$ cells in the Appendix on page 39.

Maintext now makes explicit the fact that we did not assess effects on morphology. It now reads

"Although we did not seek to gain biological insight from observation of effects of these gene deletions on cell morphology, this microscope based quantification of fluorescence signal had two advantages. First, it allowed us to rule out the possibility that putative single cell values were actually derived from clumps of several cells."

3b. Examine cell morphology of bim1 and gim4 mutants and discuss possible implication.

Authors answer : « ...do did not observe this », which I guess stands for « we did not observe this » and :

« we now include a paragraph describing our observations and our comparison with our own observations with Ohya 2005 as part of SI for P2 ». This is grammatically difficult to follow. There is no such paragraph in P2 main text, and I did not receive SI of P2 in the documents.

So, this point has possibly been addressed, but I can't say.

We thank the reviewer for bearing with our grammar and syntax.

We are not sure why SI for P2 did not reach the reviewer.

As above, we now mention our examination of morphology of $\Delta bim1$ and $\Delta gim4$ cells in the Appendix on page 39.

This text now reads

"Consideration of morphology of $\Delta bim1$ and $\Delta gim4$ strains

We verified by light microscopic observation that, compared with reference cells, all $\Delta bim1$ and $\Delta gim4$ strains, including those in the arrayed collection and the deletions made freshly in the GPY4000 background, showed no morphological differences, during normal growth or after treatment with stimulating pheromone concentrations. We assessed the morphology of these mutants in microscopic assays of cell to cell variation. In addition, when we performed the Ste5

signaling patch formation experiments in main text Figure 8, we again verified by eye that those strains showed no morphological differences to the reference strains, as previously documented. At the suggestion of a reviewer, we also examined a published repository of images of cells from the original haploid deletion collection (Ohya et al., 2005, <http://yeast.gi.k.u-tokyo.ac.jp>) and confirmed that *Δbim1* and *Δgim4* cells from the original deletion collection did not show morphological differences from the reference B4741 strain."

2. Availability of the raw flow cytometry data.

Answer from the authors : « We are in the process of depositing the data at flowrepository.org ».

So, this point has not (yet) been addressed.

This data set now appears in full in Appendix Table S2. We thank the reviewer for pushing on this, as it turned out that flowrepository.org would not accept our raw data as formatted. We therefore deposited it, for a fee, in a repository called Dryad that I am not sure will be around for the next ten years. Therefore, as a backup to Dryad, we arranged that the table be published to the Hutchinson Center digital archive, with DOI number, which is backed up by the University of California Digital Archive, which we expect to persist. The relevant DOIs and links are doi:10.5061/dryad.67bc0 and <http://authors.fhrc.org/1202/> with doi:10.6076/J77D2S8Q.

4. Better discuss possible mechanisms, which now concerns manuscript P2. I had asked to discuss :

- A possible feedback of proper MT organization on signalling (preparation for karyogamy). Authors found this point interesting (Thank you! Am I among the 3/4 « good » reviewers mentioned in correspondance to editor?) but do not discuss it in P2.

So, this point has not been addressed.

We detail our previous exchange with the reviewer below

- The main implication of microtubules organization in response to pheromone is probably to get ready for karyogamy. The authors may have found a piece of evidence of a feedback control of proper MT architecture on the pheromone signalling pathway. Cells would benefit from such a feedback, so that MTs have time to be properly organized before cell fusion.

"[From previous response] If we understand this point, we would say that work in P2 points to an additional role for normal cytoplasmic microtubule function in cells exposed to pheromone, which is to ensure (or not cause instability in) the smooth operation of the signaling apparatus. And we understand the emphasis that some previous investigators have placed on the function of cytoplasmic microtubules in nuclear congression. And for what it is worth we agree that the main function of cytoplasmic microtubules might well be to get the cell ready for karyogamy. We certainly have no sense about which function we should call most important.

The reviewer proposes a particular (and, to us, interesting and plausible) idea that alterations in cytoplasmic microtubule function might create a feedback that might delay cell fusion until the microtubules are properly organized. Again, we find this to be an interesting idea, and it's not inconsistent with our newly presented results in P2 Figure 8. We don't, however, know that our results bring enough new insight, or the authors enough deep knowledge to this topic, to help us evaluate or even speculate about it very well."

Today, we still find this an interesting idea, and we have added the speculation. New discussion sentence reads.

"..., a weaker transmitted signal in the absence of microtubules might enable cells to delay cell fusion until the nucleus is properly tethered to the membrane fusion site, a required step for zygotic nuclear fusion."

We thank the reviewer for this idea.

- A possible effect of improper oscillations of the nucleus on local cytosolic concentrations. Authors discuss the possible link between nucleus movements and signalling in P2. They prefer not speculating on the cytosolic homogenization possibility and I understand.

So, this point has been partly and satisfyingly addressed.

Actually, although we do not use the word "homogenize", we do mention this in the discussion with the word "stir". Discussion text now reads

"For example, the frequent forays of the unattached nucleus in $\Delta bim1$ (Maddox et al. 2003) might stir the cytosol and disrupt a gradient."

So we hope that point is addressed.

- Nuclear export of Fus2p to the tip. This export (and ref) is now mentioned in legend 1e of P2. It is not specifically discussed but I understand that not all possibilities can.

We mention Fus2 in the legend to what is now Figure 9.

So, this point has been addressed.

In conclusion, very few of the major points I had raised have been addressed.

I hope we have now addressed the reviewer's major points satisfactorily.

Regarding the fullness of each study : The main interest of initially-submitted P1 was that the screen led to interesting biological observations, especially the differences in traces, which has now been transferred to P2 (Fig 2 there). As it is now, P1 describes a screen and a list of genes « validated », which does not correspond to the type of articles I am used to reading in MSB. Also, results in P2 could be presented in a more concise way, and many long sentences in both manuscripts tend to dilute the key messages. My recommendation to the editor is therefore to ask for a single, streamlined and coherent article, that goes from a screen to mechanistic conclusions, rather than two partial studies.

We have now so consolidated the two papers. About the reviewer's last point, we must acknowledge the limitations to the mechanistic conclusions we can draw. We believe the manuscript appropriately qualifies its results.

Reviews of second manuscript

Reviewer #1:

A companion manuscript describes a genetic screen for mutants that affect the variability of signaling in the yeast pheromone response system. This manuscript pursues mechanistic analysis of those mutants that affect microtubule functions. The experiments analyze an array of microtubule perturbations to describe those that do and do not affect signaling variability, and test some specific models. Using signaling bypass methods and deletion of distinct MAP kinases, the authors constrain which step(s) in the pathway are susceptible to microtubule-controlled variation. They further show that these microtubule perturbations disrupt the polarized membrane localization behavior of the signaling scaffold protein, Ste5.

In general, the experiments and data here are extensive, and the logic behind the approaches is sound. Ultimately, however, it remains unclear why some microtubule perturbations (and not others) affect signaling variability, and hence the responsible mechanisms remain enigmatic. The paper is admirably incisive in testing the "nuclear position" model, which is shown to be insufficient to explain the results. It is less incisive and more speculative in addressing alternative models. For example, the type of analysis that is so effective in disproving the "nuclear position" model (i.e., poor correlation of output with nucleus-to-tip distance; Fig 3D), is not applied to later tests of Ste5 patch polarization in Fig 8 (i.e., do the effects on Ste5 polarization correlate with signaling

variation?). Consequently, while there are many relevant experiments, the mechanistic insights remain somewhat limited. The findings serve to constrain viable models without really nailing down a specific model

For what it is worth, we concur completely with this overall assessment of limitations.

Perhaps that is as good as one can expect at this point, though this ambiguity is not really explicitly acknowledged in the paper. Instead, the Discussion is long-winded and philosophical about issues that are not concretely advanced here. The manuscript also contains numerous errors.

This study surely contributes to a sense of humility at the complexity of the living system. We include a graf that addresses the "as good one can expect at this point".

“Too much remains unknown. At the signaling site, there are too many different proteins operating, too many positive cross-regulatory interactions, too many simultaneously occurring mechanical processes like cargo delivery and membrane fusion that are now insufficiently understood. It is as if we had tried to understand the smooth function of an electric motor by monitoring frequency and timing of sounds it made after disrupting the operation of particular bearings, bushings, and shafts. In this light, analysis of microtubule effects on signal transmission is a classic "inverse problem", for which inferences from doable experiments are limited, and insufficient to fully describe the system under investigation (Brenner, 2010). Within these limits, however, our genetics-powered quantitative physiological experimentation enabled us to identify the proteins involved, and this in turn helped us constrain models for their function. Moreover, as in the motor analogy above, different kinds of noises may well identify aspects or axes of system function dependent on different proteins and molecular events, and so perhaps contribute to future insight.”

Major points

1. Regarding studies of Ste5 polarization in Figure 8:

(a) It is unclear if the effects of mutations on Ste5 polarization are due to increases in signaling variability or to reductions in signal strength. One way this could be clarified is to test those perturbations that do NOT increase variability: namely, nocodazole treatment, and the kar1-15 mutant (e.g., Figs 4-5). But such tests were not shown. If these were tested, they would be valuable to include here; if not, then the authors should at least provide a clear discussion of this caveat and the alternative interpretations.

We added to the discussion the idea that the effects on Ste5 might be a consequence of reduction in signal strength, and then this sentence.

"This hypothesis could be tested: it predicts that the Ste5 patches should behave normally under perturbations that eliminate the microtubule bridge (*kar1-Δ15* and microtubule depolymerizing drugs)."

(b) As was so useful in Figure 3D, here it also could be very useful to know whether the signaling output of individual cells correlates with their time of polarization initiation or duration (i.e., do cells with weakest signaling show the slowest initiation and/or shortest duration?). This seems worth testing; or, if technically prohibitive, at least discuss the issue explicitly. It seems that such information would illuminate the issue of the physiological relevance of variability.

We thank the reviewer for pushing on this point. The revised discussion now reads.

"Future experiments using faster maturing and shorter lived fluorescent protein derivatives might enable temporal correlation of transmitted signal increase and decrease with formation and loss of Ste5 patches. In the absence of such data, our results showed that perturbations that increased pathway variability impaired Ste5 accumulation at the signaling site and that their effects on variability were suppressed by artificial recruitment of Ste5 to the membrane."

2. Regarding the bypass activator experiments in Figure 6:

(a) It would help to plot the response of the same strains to pheromone, as a top row, in order to clearly compare effects on pheromone signaling versus bypass signaling. The Results text makes

several references to the normal pheromone response, but the data are unavailable; the companion paper includes no plot of dose response for *bim1* and *gim4* mutants, and no plots of the sort used in panel 6a.

Actually, we didn't reference the normal pheromone response in the strains used in this experiment, because the strains don't respond to pheromone. That is, with the exception of the Ste4 Pryciak strain, for the rest of the strains, to prevent basal PRS activation, we deleted their native Ste5 gene. So no PRS inducibility. In the paper we now say:

"Figure 6 shows results of experiments using two artificial activators: native Ste4, whose expression mimics dissociated (i.e., active) G β γ dimers, and Ste5-CTM, a fusion of the Ste5 scaffold with a transmembrane domain, whose expression mimics membrane-recruited (i.e. activated) Ste5 (Pryciak and Huntress, 1998). To prevent interference from basal activation of the native PRS, this last strain was $\Delta ste5$."

What we've done here, to address the reviewer's question, is to include the dose response in response to pheromone for the closest (most isogenic) non-Pryciak-ized, *STE5+ $\Delta bim1$* and *$\Delta gim4$* reference strains. Figure 4B now shows these dose responses. Legend to Figure 6 now includes a line.

"Figure 4b shows corresponding reduced response of $\Delta bim1$ and $\Delta gim4$ mutants to normal pheromone induction."

*(b) To a non-trivial extent, the utility of the bottom two rows of data is limited and redundant with later experiments with *fus3 Δ* mutants. That is, *Fus3* is not activated in experiments involving either *Ste11-4-Ste7* (because they use *ste5 Δ* cells) or *Ste12* (because there is no kinase signaling). Since variation is suppressed by removing *Fus3*, then the suppression of variation by these bottom two rows can be explained for the same reason, and hence they don't provide further information relevant to the position in the pathway where signaling is initiated.*

Done. We removed from the results the downstream activators Ste11-4-7 and Ste12 from the Main text.

3. It is not clear why related analyses in Figures 4, 5, and 6 are plotted differently: i.e., variability is alternately plotted as a function of signal (Fig 4), or output [O] (Fig 5), or a ratio [P_mutant / max(P_reference)] (Fig 6). A priori, I would think that plotting these all in the same way would facilitate comparison. If there is a sensible reason for plotting them differently, then it should be explained briefly. Currently, it seems as if the approach changes randomly.

Done. With one exception, we replotted all data to show $\eta^2(P)$ vs P.

That single exception is Figure 6. In these experiments, the microtubule perturbations caused changes in P, which complicated the interpretation of straight $\eta^2(P)$ v P plots, making them to be substantially different from one another. Accordingly, Figure 6A shows $\eta^2(P)$ on the y axis vs. normalized P [P_mutant / max(P_reference)] on the x axis. Having done this we now include in 6B P as a function of estradiol dose, so that the readers are aware of the differences in P and can see why we did use normalized P in panel A. Doing so allowed us to address the reviewer's desire to see "normal" PRS dose response for these strains akin to those in Figure 4.

4. In the Abstract and Discussion, the authors repeatedly suggest that their findings imply a role for microtubules in "reducing" or "suppressing" signaling variability (e.g., pg 2, Abstract; pg 15, bottom; pg 17, middle, etc.). It's not clear to me that this is correct, because depolymerization of microtubules by nocodazole (Fig 4) does not increase variability. Thus, rather than microtubules being required to suppress variability, it seems more accurate to say that variability can be induced or increased when microtubule function is disrupted (and only in very specific ways).

To state our current interpretation of the results, when a microtubule bridge is present, normally functioning microtubules are needed for signal variation to be low. However, when a microtubule bridge is absent (as in nocodazole or in *kar1- $\Delta 15$* mutant cells), signal variation is low.

To address the reviewer's concern, throughout the manuscript, we eliminated blanket references to microtubules "reducing" and "suppressing" signal variation. In all places, that is, except the title (but even the running title uses the words "microtubule dependent stabilization"). We follow up the title immediately in the abstract (4th sentence) which tells the story in all its qualifications, to wit.

"We used genetic and chemical perturbations to show that, without microtubules, PRS output is reduced but variability is unaffected, while, when microtubules are present but their function is perturbed, output is sometimes lowered, but its variability is always high."

Minor points 5. Page 7, top half: The description of experiments in Figure 3 is confusing. It introduces the set-up, and then states that "Figure 3a shows the results". Only after extensive confusion did it become evident that this section is actually describing Figure 3b instead. Not only is the citation erroneous, but it is unclear why the authors choose to describe Figure 3b first and 3a second.

We apologize for the confusion. Nuclear position experiments now moved to the Appendix (pages 40-42) and the references to the Appendix Figures in the Appendix text and Figure legends are correct. All Figures and figure legends are correct and all references to them in the Appendix text are accurate.

6. Figure 3: Panel C lacks a legend. (The current legend for "C" is actually for panel D.) Also, panel C has mysterious characters that seem to have been mistranslated ("7m" and "#10⁻³"). Finally, because panel C lacks a legend, the Results text in page 8 (middle) is confusing, as it is unclear how the rate of change is measured or expressed. 7.

The previous and confusing panel C deleted. We have made sure that all figure legends and figures are correct.

The strain list in Table S1 is severely mangled. Individual genotypes are split among as many as four (non-contiguous) pages. Plus, some "names" and "descriptions" are masked by adjacent columns. It is difficult to imagine how this could go undetected if the manuscript materials were examined carefully by co-authors prior to submission.

We apologize for confronting the reviewer with a mangled table. We believe that that particular calamity occurred during conversion of the table to PDF at the journal. We spent extensive time making a MS word, human readable strain table.

8. Page 12: Bypass results should be interpreted with care. The best language is that used at the end of the top paragraph: "...at or upstream of the recruitment of Ste5 to the membrane". Later language, in the third paragraph, is too strong: "...the reduction...caused by Tub1-828 expression mapped to the Ste5 recruitment step." It is more accurate to say it maps "at or upstream" of this step. (E.g., if Tub1-828 affected the level or function of the Galpha subunit, this would affect the bypass by Ste4 but not by Ste5-CTM; such possibilities are obscured by claiming the effect "maps to Ste5 recruitment".)

Language changed, "at or upstream of" is the language now used.

9. Page 18, end of top paragraph: "This series of experiments thus extends and generalizes the scope of a classical genetic method first used in the 1930s...". This strikes me as self-aggrandizing hyperbole, as does a companion claim in the correspondence ("...a triumphant extension of classical genetic epistasis ala Ephrussi and Beadle..."). In fact, the bypass epistasis tests performed here are a rather standard. Moreover, they use reagents that have been used for similar purposes in prior studies.

Here, in this consolidated manuscript, we clearly acknowledge our debt to Takahashi and Pryciak and have changed the language.

10. Page 2, bottom: "S to P stimulation" is undefined. We no longer use this terminology.

11. Page 2, bottom: "Moreover, each arm stimulates the other's activity". This claim is questionable, and no citation is provided to support it. It is contradicted by evidence in the literature; e.g., in Nern & Arkowitz 1998 (PMID: 9428768), disrupting the "polarity arm" does not affect signaling.

We have removed this sentence from the consolidated manuscript.

12. Page 5, bottom: "*P_PRM1 promoter*" is redundant, as is "*P_ACT1 promoter*", because the "*P*" stands for promoter.

13. Page 18, bottom: "...including *Cdc42*, *Cdc20*, and *Ste5*...". "*Cdc20*" is incorrect. Presumably either *Cdc24* or *Ste20* was intended here.

14. Page 19, top: The sentence beginning with "This picture is consistent..." is incomplete; the idea here is never finished.

All this corrected, with thanks.

15. Page 19, middle: The sentence beginning "Moreover, in some cases, ..." is incoherent.

With respect, there is a considerable difference between "incoherence" and authors using complex syntax when attempting to articulate thoughts that are new (at least to them). We freely admit to having been reproached in similar circumstances with such assessments before. For decades. We also admit here to continuing difficulty (in recent years) in translating control theory into English.

We disagree with the appellation "incoherent" but acknowledge complex syntax. In the revised manuscript, we dropped the points about concepts from control theory.

Revised sentence now reads.

"Within these limits, however, this sort of genetics-powered quantitative physiological experimentation does identify proteins needed in various processes, and constrains models for their function."

16. Page 20: The "Box" is dispensible. It would suffice to cite the literature, without rehashing.

As above, the term "noise" gets thrown around quite a lot, papers use the concepts differently, and often imprecisely (Justman, 2016). We do not want to have a paper that is largely about $\eta^2(P)$ without placing, in the Appendix, the reasoning that established this concept. We believe it will likely be of use to some readers.

17. Figure 1A: "a-factor" should say "alpha-factor".

Corrected, with thanks.

18. Figure 1D: In "step 1", *Cdc42* is shown as being phosphorylated. Surely the authors know the difference between phosphorylation and GTP exchange. Did any of them examine this figure?

Biologist-physicist communication error. Corrected, with thanks.

19. Figure 6: The pathway cartoons on the left-hand side of each panel are so small that they will never be visible when published. Probably best to just delete them.

Done, with thanks for the push.

20. Figure 8B: In the bar graphs at left, the error bars are undefined. In the table at right, there is no mention in the legend of what statistical test is used, or what is *N* (number of trials).

Corrected, with thanks. Revised Figure legend now reads.

“**B.** Bar graph plots show qualitative defects observed: cells with no polarization, cells with gaps between 1st and 2nd polarizations and cells with overlapping (coexisting) polarizations. Error bars represent the standard error as calculated by bootstrapping (10⁴ resamples) and asterisks indicate significant difference from WT as calculated by Fisher's exact test for count data. Table shows quantitative defects. Data corresponds to the mean +/- SEM, the standard deviation +/- SE and the coefficient of variation (standard deviation divided by the mean) +/- SE. Standard errors were calculated by bootstrapping (10⁴ resamples), and significant differences from WT were calculated by permutation tests (10⁴ permutations). Values for the probability p of the observed data under the null hypothesis that each mutant strain is no different from WT are shown by asterisks: * = p<0.05; ** = p<0.01; *** = p<0.001. Experiments were done in three biological replicates (N=3). As no important differences were observed among replicates, cells were pooled for the analysis. At least 80 cells of each strain were quantified (N>80).”

21. Figure 9: (a) Panel A says "matalpha1". This is incorrect. It should be "mfalpha1". (b) The legend for panel B says 3 biological replicates for each strain, but the second and fourth charts have only 2 symbols visible for each strain.

With thanks for the observations. We are no longer including this figure in the paper.

Reviewer #2: In this revised study, Pesce et al. characterize the effects of two genes involved in microtubule dynamics on the S. cerevisiae pheromone response pathway. Overall, the authors claim that cells lacking the microtubule end binding protein Bim1 or the microtubule biogenesis modulator Gim4 exhibit increased noise in signaling output of the pheromone response pathway.

Major comments: In my opinion, the manuscript does not meet the expectations required for publication in Molecular Systems Biology.

Firstly, the biological insights are already included in the accompanying paper describing the genetic screen, and thus this manuscript does not provide sufficient insight how microtubule dynamics contributes to signaling fidelity.

We now merged both papers into a single one, that we hope does meet the reviewer's expectations for publication in MSB.

Secondly, the authors use sophisticated quantitative readouts and statistical analysis to document rather minor differences that are hard to interpret. For example, I am confused about sentences like "showing at least marginally significant large reductions".

Finally, a number of claims are overstated. For example, the authors perform some experiments using various dominant active mutants, but fail to show convincing evidence that lack of microtubule dynamics interferes with pheromone signaling.

We assert that this work does establish that interference with some aspects of cytoplasmic microtubule function increases cell to cell differences in signal transmission. We are not able to connect these functional differences with *specific* mechanisms involved in microtubule dynamics, although our results point to processes that depend on the cytoplasmic microtubule plus ends in the vicinity of the plasma membrane.

Nevertheless, they draw relatively far reaching conclusions based on the data. Similarly, the claim that the increased noise is dependent on Fus3 is not warranted based on the data shown in Figure 7.

Our assertions arise from careful analysis of quantitative data, and we believe them to be temperate and warranted. In particular, we assert that the experiments in the current merged manuscript Figure 7 constitute what is probably the strongest mechanistic conclusion in this work. Consider only effects of the $\Delta bim1$ mutation. Figure 7A top left clearly shows that Fus3 is needed for the increased variation displayed by $\Delta bim1$ yeast and Figure 7A bottom left shows that, when Kss1 is absent and signaling proceeds only via Fus3, the increased variation displayed by the $\Delta bim1$ single mutant is further exacerbated. The same conclusions apply for $\Delta gim4$.

Minor comment: The legend of Figure 4 is missing some information, "strains in right column were and perturbed by *Tub1-828* expression".

With thanks to the reviewer. *tub1-828-expression*, replotted, is now in Figure 5, the nocodazole/benomyl experiments that were part of this figure are now Extended View figures, with the methods described in the Appendix, and we fixed both Maintext and Extended View Figure legends so that they are no longer problematic.

References used in this response.

Avery, L. and Wasserman, S. (1992). Ordering gene function: the interpretation of epistasis in regulatory hierarchies. *Trends in Genetics* 8, 312-316

Cordell, H. (2002). Epistasis: what it means, what it doesn't mean, and statistical methods to detect it in humans. *Human Molecular Genetics*, 2002, Vol. 11, 2463–2468

Justman, Q. A. (2016) An explicit source for extrinsic noise. *Cell Syst.* (5), 308-309.

Figures mentioned in this response, now, in the Appendix showing methodological details we believe to be non-obvious.

Genetic elements in screening strain. Non-obvious elements include the additional *URA3* terminator downstream of *SNL1*, and the promoters and terminator controlling expression of *nat1* and *hph*.

Laser setup. The idea of that one can measure YFP signal by stealing orange excitatory light will not be obvious to many or most readers.

Thank you again for submitting your work to Molecular Systems Biology and I apologize for the delay in getting back to you. We are now globally satisfied with the modifications made and the combined manuscript is now much clearer. I am please to inform you that we will be able to accept your paper for publication in Molecular Systems Biology pending the following minor modifications:

Main manuscript

- On p. 3 it was not immediately clear what is meant by "This work asserted analytically that...". Would the following simplification work? "In the current work, we make the assumption that..."
- In Table 2, it would be useful to add a column with a short description/code that summarizes the behavior of the respective mutants (low or high $\eta^2(P)$, high v_a at high dose etc...)
- The last 2 words of the discussion are "therapeutic intervention". While speculations are interesting, it might not be completely necessary to relate your basic research results to the hope of potential future therapeutic intervention. If it is important for you, it is fine to leave it, but if it does not strengthen the work, maybe it is better to leave out.
- Please add a Conflict of Interest statement.
- FYI: The tables in the main manuscript will be typeset in black and white.
- Please update the reference list to 20 authors + et al instead of 10 authors + et al, to match the MSB reference style.

Appendix

- All Appendix items should be in the same file. Please include the Appendix figures in the Appendix file.
- Please include the legends of Tables S1-4, S6 as a separate worksheet in the Excel file or, alternatively, zip together a README text file with the table.

Callouts

- Fig. 4 has been called out before fig. 3. It is mandatory to number the figures by order of appearance in the text. Figure 3 should be cited in the section presenting the results of the clustering analysis.
- All panels in the main figures should be called out explicitly. Please add a callout to fig 7A.

HTML version of the paper

Please supply the following:

- three to four 'bullet points' highlighting the main findings of your study
- a short 'blurb' text summarizing in two sentences the study (max. 250 characters)
- a 'thumbnail image' (width=211 x height=157 pixels, Illustrator, PowerPoint, OmniGraffle or jpeg format), which can be used as 'visual title' for the synopsis section of your paper.

Detailed response:

Main manuscript

- *On p. 3 it was not immediately clear what is meant by "This work asserted analytically that...". Would the following simplification work? "In the current work, we make the assumption that..."*

We had intended to discuss an important aspect of our 2005 work (an assumption we made), rather than to say anything about the current work. We changed this confusing sentence to read:

“In this previous work, we made the assumption that cell-to-cell differences in (P) were composed of $\eta^2(L)$, (differences in L, the capacity component of the signal transmission subsystem at the start of the experiment) and $\eta^2(\lambda)$, (rapid acting changes in signal during the measurement) but we could not separate $\eta^2(L)$ and $\eta^2(\lambda)$ experimentally.”

- In Table 2, it would be useful to add a column with a short description/code that summarizes the behavior of the respective mutants (low or high $\eta^2(P)$, high ν at high dose etc...)

With thanks for this suggestion, which significantly improves the table. “Screen” is now labeled “Screen, Criteria”. The criteria field is filled with a list of between one and three numbers, with values between 1 and 5. The numbers indicate which of our selection criteria were met by each strain. We added the following text to the Table 2 legend:

“**Selection criteria codes: (see Figure 2A-C)**

- (1) $O(0.6 \text{ nM}) < 3.39$
- (2) $O(0.6 \text{ nM}) > 7.72$
- (3) $\eta^2(P(0.6 \text{ nM})) < 0.027$
- (4) $\eta^2(P(0.6 \text{ nM})) > 0.054$
- (5) $\eta^2(P(20 \text{ nM})) > 0.019$ ”

- The last 2 words of the discussion are "therapeutic intervention". While speculations are interesting, it might not be completely necessary to relate your basic research results to the hope of potential future therapeutic intervention. If it is important for you, it is fine to leave it, but if it does not strengthen the work, maybe it is better to leave out.

We changed it. The original closing sentence was:

“The existence of genes that affect pathway variability independently of transmitted signal strength suggests that manipulation of pathway variability by genetic manipulation of plants and animals should be possible, and that some identified genes might one day define targets for therapeutic intervention.”

New closing sentence reads:

“In this light, the existence of alleles affecting pathway variability may help motivate development of genetic and pharmacological interventions aimed at reducing it.”

- Please add a Conflict of Interest statement.

We did so, between Author Contributions and Acknowledgments.

This statement reads: "The authors affirm that they have no conflicts of interest."

- FYI: The tables in the main manuscript will be typeset in black and white.

We removed extraneous shading from Table 2.

- Please update the reference list to 20 authors + et al instead of 10 authors + et al, to match the MSB reference style.

We edited our Endnote style file and reformatted the references to match MSB style

Appendix

- All Appendix items should be in the same file. Please include the Appendix figures in the Appendix file.

We inserted figures into the Appendix, each one adjacent to its legend.

- Please include the legends of Tables S1-4, S6 as a separate worksheet in the Excel file or, alternatively, zip together a README text file with the table.

We included these legends as separate worksheets within the Excel files, and labeled them as such.

Callouts

- Fig. 4 has been called out before fig. 3. It is mandatory to number the figures by order of appearance in the text. Figure 3 should be cited in the section presenting the results of the clustering analysis.

We apologize for this error, and thank you for catching it. We now call out Figure 3, immediately before the callout for Table S2, near the bottom of page 5.

Revised text now reads "...we grouped the mutant strains in the secondary screen using a hierarchical clustering approach based on the 5 variables we measured by flow cytometry, at low and high pheromone dose (Figure 3 and Table S2)."

- All panels in the main figures should be called out explicitly. Please add a callout to fig 7A.

We added a callout for Figure 7A, near the top of page 12, during the discussion of its data, together with references to other panels in the figure.

Revised text now reads

"Thus, we measured $\eta^2(P)$ in $\Delta fus3$ or $\Delta kss1$ cells. Notably, in $\Delta bim1$ and $\Delta gim4$ strains, the $\Delta fus3$ deletion suppressed, at all doses, the increased $\eta^2(P)$ (Figure 7A, top panels), while the $\Delta kss1$ deletion exacerbated the defect of $\Delta bim1$ (Figure 7A, bottom panels). The fact that deletion of *FUS3* eliminated the increase in pathway variability caused by the $\Delta gim4$ and $\Delta bim1$ mutations showed that the increased variability was not a secondary consequence of a generalized increase in variability in cells with disrupted microtubule function, but rather reflected an effect of these mutations on the operation of the PRS. These results demonstrate that microtubule perturbations increase pathway variability $\eta^2(P)$ by specifically impacting signaling by the MAP kinase Fus3.

In its effects on P, deletion of *Afus3* was additive to the diminution caused by *Abim1*. By contrast, the deletion of *kss1* counteracted the diminution caused by *Abim1*, so that P for the double mutant *Δkss1-Δbim1* was above the reference at all but the highest doses (Figure 7B, left panels). The reduction of signal strength due to $\Delta gim4$ (Figure 7B, top right) was smaller in cells that were also $\Delta fus3$, whose reduction in P was the same as in $\Delta fus3$ *GIM4+* cells. By contrast, in $\Delta gim4$ cells, the additional deletion *Δkss1* does not enhance signal strength, but in fact reduces it (Figure 7B, bottom right). These results further support the idea that the mechanism(s) that affect pathway variability is (are) distinct from those affecting signal transmission strength."

HTML version of the paper

Please supply the following:

- three to four 'bullet points' highlighting the main findings of your study

We inserted these into the main text file, between the abstract and the keywords. These are:

- 1) Screen identifies yeast microtubule mutants with increased pheromone signaling variability
- 2) Mutations and other perturbations affecting cytoplasmic microtubule plus end function cause erratic signaling
- 3) Signal variability affects gene expression, cell polarization

4) Variation due to effect on Fus3 MAP Kinase at cell membrane signaling site

- a short 'blurb' text summarizing in two sentences the study (max. 250 characters)

We inserted this into the main text file, immediately before the bullet points.

Blurb reads: "A genetic screen for pheromone signal variation mutants and other experiments showed that cytoplasmic microtubule plus end function stabilizes signaling. Perturbing + end function impacted signal transmission by Fus3 MAPK and cell polarization."

- a 'thumbnail image' (width=211 x height=157 pixels, Illustrator, PowerPoint, OmniGraffle or jpeg format), which can be used as 'visual title' for the synopsis section of your paper.

After looking through other thumbnail images for MSB articles, we took the time to try to make a tiny image that captured the ideas of the work.

Corresponding Author Name: Gustavo Pesce

Manuscript Number: MSB-16-7390